# ALIGNING DISTRIBUTIONALLY ROBUST OPTIMIZATION WITH PRACTICAL DEEP LEARNING NEEDS

## ABSTRACT

Deep learning (DL) models often struggle with real-world data heterogeneity, such as class imbalance or varied data sources, as standard training methods treat all samples equally. Distributionally Robust Optimization (DRO) offers a principled approach by optimizing for a worst-case data distribution. However, a significant gap exists between DRO and current DL practices. DRO methods often lack adaptive parameter updates (like Adam), struggle with the non-convexity of neural networks, and are difficult to integrate with group-based weighting in standard mini-batch training pipelines. This paper aims to bridge this gap by introducing ALSO – Adaptive Loss Scaling Optimizer – a novel optimizer that integrates an adaptive, Adam-like update for the model parameters with an efficient, principled mechanism for learning worst-case data weights. Crucially, it supports stochastic updates for both model parameters and data weights, making it fully compatible with group-based weighting and standard Deep Learning training pipelines. We prove the convergence of our proposed algorithm for non-convex objectives, which is the typical case for DL models. Empirical evaluation across diverse Deep Learning tasks characterized by different types of data heterogeneity demonstrates that ALSO outperforms both traditional DL approaches and existing DRO methods.

## 1 INTRODUCTION

Deep Learning (DL) has long been centered around the empirical risk minimization (ERM) problem:

$$\min_{\theta \in \mathbb{R}^d} \left\{ \frac{1}{n} \sum_{i=1}^n f_i(\theta) + \frac{\tau}{2} \|\theta\|_2^2 \right\}, \tag{1}$$

where $\theta$ are the parameters of the DL model, $f_i(\theta)$ is the loss function on the $i$-th element $(\mathbf{x}_i, \mathbf{y}_i) \in \mathbf{X} \times \mathbf{Y}$ of the training data, $n$ is the number of training samples and $\frac{\tau}{2}\|\theta\|_2^2$ is a regularization term used to avoid overfitting (Ying, 2019). The standard ERM framework, and the optimizers designed for it like SGD and Adam (Kingma, 2014), implicitly assume that all training samples are of equal importance. However, this assumption rarely holds in real-world applications, which are often characterized by significant data heterogeneity. Datasets may suffer from severe class imbalance or be composed of data from different sources with varying distributions. In these common scenarios, treating all samples equally can lead to models with suboptimal performance and poor generalization.

A principled approach to address this challenge is Distributionally Robust Optimization (DRO) (Delage & Ye, 2010; Lin et al., 2022; Wiesemann, 2024). Instead of minimizing loss over a fixed, uniform data distribution, DRO seeks to optimize the model performance against a "worst-case" distribution. While DRO is a broad area (Wiesemann, 2024), one of the common formulations of this idea leads to the following minimax problem:

$$\min_{\theta \in \mathbb{R}^d} \left\{ L_{DRO}(\theta) := \max_{\pi \in \Delta_{n-1} \cap U} \sum_{i=1}^n \pi_i f_i(\theta) + \frac{\tau}{2} \|\theta\|_2^2 \right\}, \tag{2}$$

where $U$ is an uncertainty set, i.e. constraint on $\pi$. For example, one can use KL-divergence ball to prevent significant deviations from some prior distribution $\hat{\pi}$: $U = \{\pi \in \Delta_{n-1} : \text{KL}[\pi \| \hat{\pi}] \leq r\}$. Although DRO has successful applications in specific DL fields (Lotidis et al., 2023; Kallus et al., 2022; Liu et al., 2022; Blanchet & Kang, 2020), we identify several challenges in applying existing methods for general DL:

- **Lack of adaptive $\theta$ update**. Most general DRO methods either use simple SGD updates (Carmon & Hausler, 2022), Normalized SGD (Jin et al., 2021) or apply Variance Reduction (VR) techniques (Mehta et al., 2024; 2023; Levy et al., 2020; Qi et al., 2021), while the most successful DL optimizers are adaptive (Kingma, 2014; Choi et al., 2019). Although some adaptive DRO methods exist (Guo & Yang, 2024), they are often impractical, introducing instability and overfitting risks (see Section 2), since they solve the problem (3) instead of the problem (4).
- **Gap Between Theory and Practice**. Despite the success of the existing DRO methods in the convex domain (e.g. logistic regression) (Mehta et al., 2024; 2023; Levy et al., 2020), neural networks are inherently non-convex, presenting additional challenges. Several attempts have been made to develop DRO methods specifically for Deep Learning, but they are either heuristic (Liu et al., 2021; Sagawa et al., 2019), or pose instability and overfitting risks (Qi et al., 2021; Jin & Sidford, 2020; Guo & Yang, 2024).
- **Challenges with Batching and Grouping.** For practical needs one often wants to assign weights to samples based on their specific properties such as class (He & Garcia, 2009; Lin et al., 2017) or worker identification (Mohri et al., 2019), rather than assigning unique weights to individual objects. The problem (2) can deal with this requirement if one uses $i$ as group id, not object, i.e. $f_i$ is the total loss of the group (and $n$ is number of groups). However, most DRO methods assume that $f_i$ is deterministic, which is impractical for group-based weighting in cases the presence of large groups, since calculating the full $f_i$ requires a pass over the entire group. Additionally, the requirement of full $f_i$ computation complicates algorithm integration into the standard DL training pipelines with batching.

This paper aims to bridge this critical gap by introducing `ALSO` – Adaptive Loss Scaling Optimizer – an optimizer designed to align DRO with the needs of practical Deep Learning. `ALSO` is designed from the ground up to be practical: it employs an adaptive, Adam-like step for the model parameters, deals with stochastic updates for both $\theta$ and $\pi$ (allowing standard batching during training with group-based $\pi$), and effectively solves the distribution-finding subproblem for the group weights. We provide a rigorous theoretical analysis, proving `ALSO`'s convergence for non-convex objectives typical in Deep Learning.

The key contributions of this work are:

- **Deep Learning DRO optimizer.** We present `ALSO` – a novel algorithm designed to solve the problem (4) in Deep Learning contexts (see Algorithm 1).
- **Theory.** We establish a convergence of `ALSO` in the stochastic, non-convex, $L$-smooth case.
- **Experiments**. We experimentally demonstrate that `ALSO` outperforms classical DL approaches and DRO algorithms in a diverse set of DL tasks characterized by data heterogeneity: learning from unbalanced data, tabular DL, robust training against adversarial attacks, distributed training with gradient compression, and split learning. Our code is available at `https://anonymous.4open.science/r/ALSO-B4DA/`.

## 2 BACKGROUND

Distributionally Robust Optimization has emerged as a powerful framework for decision-making under uncertainty (Delage & Ye, 2010; Lin et al., 2022; Wiesemann, 2024). DRO has successful applications in separate DL fields such as Reinforcement Learning (Lotidis et al., 2023; Kallus et al., 2022; Liu et al., 2022), Semi-Supervised Learning (Blanchet & Kang, 2020), Sparse Neural Network training (Sapkota et al., 2023). However, none of these methods are for general-purpose use.

Most general DRO methods use simple SGD updates (Carmon & Hausler, 2022) or apply Variance Reduction (VR) techniques (Mehta et al., 2024; 2023; Levy et al., 2020). The main goal of such methods is to reduce the complexity of the optimization process for convex functions and to have a step cost independent of data size. Despite their success in the convex domain, neural networks are inherently non-convex, presenting additional challenges. VR methods are usually ineffective in DL (Defazio & Bottou, 2019), because of data augmentation and batch normalization, which disrupt finite-sum structure. However, recently proposed `MARS` optimizer (Yuan et al., 2024) achieves good performance in language modeling tasks – the field, in which none of the techniques mentioned above are used. Additionally, `MARS` utilizes STORM (Cutkosky & Orabona, 2019) for variance reduction which is closer to `ALSO` negative momentum (see Algorithm 1), rather than to classical VR methods

like SAGA (Defazio et al., 2014) or SVRG (Johnson & Zhang, 2013) used in most DRO methods. Furthermore, SAGA based methods like state-of-the-art DRO methods (Mehta et al., 2024) require storing a table of size $n \times d$ – a significant limitation for large Deep Learning models with millions of parameters. It is also important to note that these methods heavily depend on deterministic $f_i$, i.e. require either assigning unique weights to individual objects or the loss computation for the whole group, and usually have limited experimental validation in neural network training scenarios.

From another perspective, several attempts have been made to develop DRO methods specifically for Deep Learning. For instance, in (Liu et al., 2021) the authors propose a heuristic algorithm without theoretical guarantees that requires two separate training phases to produce the final model. An alternative approach is presented in (Sagawa et al., 2019), where the authors propose an algorithm with convergence guarantees for the convex case and apply it to neural network training. However, this work implements a simple gradient step for $\theta$ update, while the most successful DL optimizers are adaptive (Kingma, 2014; Choi et al., 2019). Another approach is proposed by (Qi et al., 2021; Jin et al., 2021; Guo & Yang, 2024), where authors address the non-convex scenario. They solve the inner maximization problem exactly, resulting in the following formulation:

$$\min_{\theta \in \mathbb{R}^d} \left\{ \log \left( \frac{1}{n} \sum_{i=1}^{n} \exp \left[ \tau^{-1} f_i(\theta) \right] + \frac{\tau}{2} \|\theta\|_2^2 \right) \right\}. \tag{3}$$

While reformulation (3) eliminates the need to store and update $\pi$, it has several important limitations. First, since the problem (3) involves expressions of the form $\exp[\tau^{-1} \cdot]$, small values of $\tau$ can lead to extremely large values that may be computationally intractable. Second, modern deep neural network training methods are iterative, with initial weight vectors $\theta^k$ typically far from optimal. However, in (3), we immediately compute the optimal vector $\pi^*$, which can be problematic in early training stages when higher errors on some samples may simply reflect undertraining rather than inherent difficulty. Furthermore, using the exact value of $\pi^*$ may lead to overfitting to outliers in the training set, despite the regularization term in the problem (4). Finally, the approach in (3) fundamentally assumes that each data point has its own weight. When multiple objects share a weight, $f_i$ becomes the sum of losses for these objects. This constraint limits batching strategies (if we use a subset of objects with the same weight to compute the stochastic gradient of (3), we obtain a biased estimation of gradient), making the proposed approach less practical for Deep Learning applications. Although the problem (3) has significant limitations, Deep Learning methods for solving it exist. For instance, (Jin et al., 2021) utilize Normalized SGD. While this method can be applied to the DL, it usually provides worse performance than Adam. Another relevant work by (Guo & Yang, 2024) introduced an adaptive method for the problem (3) in the Federated Learning context, where the main goal is to minimize number of communications. Still, the main limitation of these methods in DL context is the problem they solve, which has significant limitations discussed above and performs worse in our experiments (see Section 5).

## 3 PROBLEM STATEMENT

As discussed in the introduction, it is a common requirement to assign the same weight to several samples based on their properties such as class (He & Garcia, 2009; Lin et al., 2017) or worker identification (Mohri et al., 2019). The straightforward idea is to retain the problem (2), but use $f_i$ as the mean loss on the objects of the group $i$. However, this objective hides the structure of the problem (i.e. $f_i$ is the sum), resulting in methods that require deterministic $f_i$ and implies that we need to compute the whole $f_i$ to make step, which aligns poorly with model training pipeline, where we use mini-batches to make a step (i.e. the whole $f_i$ is unavailable). To make this structure more precise, we use the following modified objective:

$$\min_{\theta \in \mathbb{R}^d} \max_{\pi \in \Delta_{c-1} \cap U} \left\{ h(\theta, \pi) := \sum_{i=1}^{c} \pi_i \left( \frac{c}{n} \sum_{j=1}^{n_i} f_{i,j}(\theta) \right) + \frac{\tau}{2} \|\theta\|_2^2 - \lambda \mathrm{KL} \left[ \pi \| \hat{\pi} \right] \right\}. \tag{4}$$

In the problem (4) weights $\pi_i$ are assigned to each of $c$ object groups. The group $i$ contains $n_i$ samples with loss functions $f_{i,j}, j \in \overline{1, n_i}$. These groups can be built based on sample class, worker ID, or each sample can compose its own group ($c = n$, $n_i = 1 \ \forall i$ in such scenario), making problems (4) and (2) almost equal. To additionally restrict deviation from the starting distribution, we use

a regularization technique, where $\lambda > 0$ is the regularization parameter. The KL-divergence term $\text{KL}\left[\pi \| \hat{\pi}\right]$ is introduced specifically to avoid degenerate solutions in which $\pi$ collapses onto a single group. Compared to the Euclidean norm, KL-divergence is a more natural choice for probability distributions on the simplex: it better respects the underlying geometry and penalizes shifts in high-probability components more strongly, thereby stabilizing the updates. As a baseline, we can set $\hat{\pi} = \mathcal{U}\left(\overline{1, c}\right) \in \Delta_{c-1}$ as the uniform discrete distribution; however, sometimes it is better to define it in a different way (for such example see Subsection 5.1). It is worth highlighting that we choose constant multiplier $\frac{c}{n}$ so that if we substitute $\pi = \hat{\pi} = \mathcal{U}\left(\overline{1, c}\right)$ into (4), the resulting equation is exactly the same as ERM.

## 4    ALSO – ADAPTIVE LOSS SCALING OPTIMIZER

The development of our algorithm is motivated by the evolution of optimization methods for saddle point problems. The easiest option to obtain methods for saddle point problems is to adapt gradient schemes from minimization tasks. In this way, it is possible to obtain the Stochastic Gradient Descent Ascent (SGDA) method. However, this scheme is inadequate from both the theoretical and practical perspective even for the simplest problems (Goodfellow, 2016; Beznosikov et al., 2023). Therefore, it is suggested to use more advanced algorithms such as Extragradient (Korpelevich, 1976). For our non-Euclidean geometry in the problem (4), it makes sense to consider an appropriate modification of Extragradient – Mirror-Prox (Nemirovski, 2004):

$$
\theta^{k+\frac{1}{2}} = \theta^k - \eta \cdot \left(\nabla_\theta h(\theta^k, \pi^k) + \tau \theta^k\right)
$$

$$
\pi^{k+\frac{1}{2}} = SM\left[\log \pi^k - \eta \cdot \left(\nabla_\pi h(\theta^k, \pi^k) + \lambda \log \frac{\pi^k}{\hat{\pi}}\right)\right]
$$

$$
\theta^{k+1} = \theta^k - \eta \cdot \left(\nabla_\theta h(\theta^{k+\frac{1}{2}}, \pi^{k+\frac{1}{2}}) + \tau \theta^k\right)
$$

$$
\pi^{k+1} = SM\left[\log \pi^k - \eta \cdot \left(\nabla_\pi h(\theta^{k+\frac{1}{2}}, \pi^{k+\frac{1}{2}}) + \lambda \log \frac{\pi^k}{\hat{\pi}}\right)\right]
$$

Here $SM$ denotes softmax operator and $\eta$ is learning rate. However, both Extragradient and Mirror-Prox require two gradient computations per iteration. To address this, so-called Optimistic version of these algorithms can be applied (Popov, 1980), which requires only one gradient call per iteration:

$$
\theta^{k+1} = \theta^k - \eta \cdot \left((1+\alpha)\nabla_\theta h(\theta^k, \pi^k) - \alpha \nabla_\theta h(\theta^{k-1}, \pi^{k-1}) + \tau \theta^k\right)
$$

$$
\pi^{k+1} = SM\left[\log \pi^k - \left((1+\alpha)\nabla_\pi h(\theta^k, \pi^k) - \alpha \nabla_\pi h(\theta^{k-1}, \pi^{k-1}) + \lambda \log \frac{\pi^k}{\hat{\pi}}\right)\right] \tag{5}
$$

It turns out that the Extragradient and Optimistic updates outperform SGDA not only in the theory, but also in DL practice, particularly in training GANs (Daskalakis et al., 2017; Gidel et al., 2018; Mertikopoulos et al., 2018; Chavdarova et al., 2019; Liang & Stokes, 2019; Peng et al., 2020). In practice, nearly all works that employ Optimistic method for DL do not use its theoretical version, but rather an adaptive variant (typically with Adam-style stepsizes). This substitution is often justified as a standard procedure in DL. However, we question this approach, as establishing theoretical guarantees for adaptive methods is a nontrivial and technically demanding task.

Building upon the foundation of Optimistic Mirror-Prox, we introduce `ALSO` (Algorithm 1) – `Adaptive Loss Scaling Optimizer` – which effectively addresses our requirements. Optimistic Mirror-Prox utilizes GD-like step over $\theta$ and uses full gradient for both $\theta$ and $\pi$. To enhance adaptivity, we replace this GD step with Adam (Kingma, 2014) and leave the same step over $\pi$ as before; to allow batching we replace full gradients with stochastic ones, resulting in our proposed `ALSO` algorithm for solving the problem (4). In this work, we do not follow the standard simplified route; instead, we provide a rigorous analysis of the adaptive method (see Theorem 4.5).

---

**Algorithm 1** ALSO

---

1: **Parameters:** $\gamma_\theta, \gamma_\pi$ – stepsize for $\theta$ and $\pi$; $\beta_1, \beta_2, \varepsilon$ for Adam; momentum $\alpha$; $\lambda, \tau$ – regularization parameters for $\pi$ and $\theta$; number of iterations $T$; $\hat\pi$ – regularization distribution.
2: **Initialization:** $m^0 = g^0 = p^0 = \mathbf{0}$, $v_0 = 0$, $\pi^0 = \hat\pi$, $\hat\gamma_\pi = \gamma_\pi/(1 + \gamma_\pi \lambda)$
3: **for** $k = 0, 1, 2, \ldots, T$ **do**
4: $\quad$ Sample $B$ objects: $\{(c_1^k, i_1^k), ..., (c_B^k, i_B^k)\}$ – pairs (group, index)
5: $\quad$ $g^{k+1} = \frac{c}{B} \sum_{j=1}^{B} \pi_{c_j^k} \nabla_\theta f_{c_j^k, i_j^k}(\theta^k)$
6: $\quad$ $\hat{g}^{k+1} = (1 + \alpha)g^{k+1} - \alpha g^k + \tau \theta^k$
7: $\quad$ $p^{k+1} = \frac{c}{B} \sum_{j=1}^{B} e_{c_j^k} \cdot f_{c_j^k, i_j^k}(\theta^k)$, where $e_i$ is vector with 1 in $i$-th position and zeros in others
8: $\quad$ $\hat{p}^{k+1} = (1 + \alpha)p^{k+1} - \alpha p^k$
9: $\quad$ $\theta^{k+1} = \theta^k - \gamma_\theta \cdot \text{Adam}(\hat{g}^{k+1}, \beta_1, \beta_2, \varepsilon)$
10: $\quad$ Option I: $\pi^{k+1} = \text{SM}\left[\log \pi^k - \hat\gamma_\pi(\hat{p}^{k+1} + \lambda \log(\pi^k/\hat\pi))\right]$
11: $\quad$ Option II: $\pi^{k+1} = \arg\min_{\pi \in U \cap \Delta_{c-1}} \left\{\hat\gamma_\pi \langle \hat{p}^{k+1} + \lambda \log \frac{\pi}{\hat\pi}, \pi\rangle + \text{KL}[\pi \| \pi^k]\right\}$
12: **end for**

---

**Discussion.** In contrast to (5), where full gradients are used, in Lines 5, 7 of Algorithm 1, we use gradients obtained by a straightforward sampling strategy: we unite all objects into a single group and then sample from it. This approach is mathematically equivalent to combining the two sums in the equation (4) and selecting $B$ terms from the unified sum. This method allows for seamless integration of ALSO into standard Deep Learning training pipelines. Nevertheless, alternative sampling strategies are viable. For instance, one might first sample groups and subsequently sample objects within each selected group, if it is suitable for specific applications (see Appendix A). In Lines 6, 8 we utilize negative momentum – a common technique used to prevent too sharp steps and obtain convergence. While introduction of this term is inspired by Optimistic Mirror-Prox, the similar term is used in MARS (Yuan et al., 2024) to reduce the variance. This observation further confirms this design choice. We additionally ablate it in Appendix D. In Line 9 we utilize Adam step to update $\theta$. It is worth noting that Adam itself can be seen as a particular case of ALSO: if we set the hyperparameters so that $\pi$ remains constant and equal to $1/c$, the procedure reduces to Adam. In Lines 10, 11 we present two options for the $\pi$ update. Option I employs $U = \Delta_{c-1}$ for simplicity and is used in practical implementation. Option II provides a more general formulation and is particularly valuable for theoretical analysis. The step over $\pi$ has time complexity $O(c)$, which in theory can be costly. However, in many applications $c$ is relatively small (see Section 5). Furthermore, for most DL models, gradient computations consume the majority of training time (Jiang et al., 2021). Based on these observations, we determined that updating $\pi$ using simple arithmetic operations with $O(c)$ complexity satisfies practical requirements. We validate this assessment experimentally in Appendix D.1.

## 4.1 Convergence of ALSO

We now present assumptions for the convergence analysis.

**Assumption 4.1.** The admissible domain $\mathcal{D}_\pi := \Delta_{c-1} \cap U$ is nonempty, closed, and convex. Moreover, regularizer $\hat\pi \in \text{Int}(\mathcal{D}_\pi)$

**Assumption 4.2.** For all $(i, j)$ the functions $f_{i,j}$ from (4) are $K_{i,j}$-Lipschitz continuous and $L_{i,j}$-smooth on $\Theta$ with respect to the Euclidean norm $\|\cdot\|_2$, i.e., for any $\theta^1, \theta^2 \in \Theta$ the following inequality holds:

$$\|\nabla f_{i,j}(\theta^1) - \nabla f_{i,j}(\theta^2)\| \le L_{i,j}\|\theta^1 - \theta^2\|_2 \text{ and } |f_{i,j}(\theta^1) - f_{i,j}(\theta^2)|_2 \le K_{i,j}\|\theta^1 - \theta^2\|_2.$$

**Assumption 4.3.** At each iteration of Algorithm 1 we have access to oracles $g = g(\theta, \pi)$ and $p = p(\theta, \pi)$, which provide unbiased estimates of the gradients for the problem (4) using batch size $B$. Moreover,

$$\mathbb{E}\|g(\theta, \pi) - \nabla_\theta h(\theta, \pi)\|_2^2 \le \frac{\sigma^2}{B}, \qquad \mathbb{E}\|p(\theta, \pi) - \nabla_\pi h(\theta, \pi)\|_2^2 \le \frac{\sigma^2}{B}.$$

For example, if one uses straightforward sampling (Lines (5), (7)) without any other source of stochasticity (e.g., no dropout, augmentations, etc.), then $\sigma^2 = \mathcal{O}\left(K^2 \cdot \max\left\{c^2, \frac{c^3 \sum_{i=1}^{c} n_i^2}{n^2}\right\}\right)$, where $K = \max_{(i,j)} K_{i,j}$. See Appendix A for derivation.

**Definition 4.4** (Stationary point, cf. (Lin et al., 2020))**.** A point $\theta$ is called an $\varepsilon$-stationary point ($\varepsilon \geq 0$) of a differentiable function $\Phi$ if $\|\nabla\Phi(\theta)\| \leq \varepsilon$. If $\varepsilon = 0$, then $\theta$ is a stationary point.

In our setting, the primal objective is $\Phi(\theta) := \max_{\pi \in \mathcal{D}_\pi} h(\theta, \pi)$, which is differentiable since $h(\theta, \pi)$ is smooth with respect to $\theta$ and the maximization is over a compact convex set. Therefore, following (Lin et al., 2020), it is sufficient to measure convergence of Algorithm 1 by the gradient norm $\|\nabla\Phi(\theta)\|$, as small gradients certify approximate stationarity of the original min–max problem (4). Moreover, due to stochasticity in the updates, it is natural to adopt the criterion $\mathbb{E}\|\nabla\Phi(\theta)\|^2 \leq \varepsilon^2$.

Now we are ready to present the following main theorem, which establishes the complexity bounds of Algorithm 1.

**Theorem 4.5.** *Under Assumptions 4.1, 4.2, 4.3, the required number of iterations to achieve $\varepsilon$-stationarity 4.4 ($\mathbb{E}\|\nabla\Phi(\theta)\|^2 \leq \varepsilon^2$) for the problem* (4) *by* ALSO *(Algorithm 1) with* $\gamma_\theta = \mathcal{O}(\frac{\lambda^4}{L^4})$, $\gamma_\pi = \frac{\lambda}{8L^2}$, $\beta_1 = \mathcal{O}(\frac{\varepsilon\lambda^2}{L^2})$, $\beta_2 = 1 - \mathcal{O}(\varepsilon^2)$, $B = \mathcal{O}(\frac{\sigma^2}{\varepsilon^2})$ *is*

$$T = \mathcal{O}\left( \frac{L^4}{\lambda^4 \varepsilon^2} \cdot \max\{\Delta_\Phi \cdot (K + \sigma),\ D_0\} \right),$$

*where* $\Delta_\Phi = \Phi(\theta^0) - \min_{\theta \in \mathbb{R}^d} \Phi(\theta)$, $D_0 = KL(\pi^*(\theta^0)\|\pi^0)$, $\pi^*(\theta) = \arg\max_{\pi \in \mathcal{D}_\pi} h(\theta, \pi)$ *and* $L^2 = \mathcal{O}\left( \left( \frac{c}{n} \max_i \sum_{j=1}^{n_i} L_{i,j} + \tau + \frac{c}{n} \max_i \sum_{j=1}^{n_i} K_{i,j} \right)^2 + \lambda^2 \right)$, $K = \frac{c}{n} \max_i \sum_{j=1}^{n_i} K_{i,j}$.

Appendix F provides a detailed derivation and discusses parameter selection.

**Discussion.** This convergence result matches the guarantees of the standard SGDA method (Lin et al., 2020) in terms of both iteration complexity $O(\frac{1}{\varepsilon^2})$ and batch size in the stochastic regime $O(\frac{1}{\varepsilon^2})$, resulting in total computational complexity $O(\frac{1}{\varepsilon^4})$. Furthermore, our rate matches lower-bound from (Li et al., 2021). In contrast to (Lin et al., 2020), we incorporate Adam-type updates on the $\theta$-side and provide a dedicated analysis of the Adam estimator to obtain such bounds. Moreover, unlike SGDA, ALSO leverages a non-Euclidean geometry, instead of Euclidean projection used in (Lin et al., 2020). Our work also contrasts with other recent analyses of adaptive methods for saddle-point problems. For instance, while (Guo et al., 2025) also analyze Adam-based method, their approach relies on euclidean geometry, which is less suitable for the problem (4). Moreover, convergence analysis in (Guo et al., 2025) relies on Lipschitz continuous gradient for both variables, which is violated in the problem (4) due to the KL-divergence term. The work (Yang et al., 2022) not only shares the same issues as (Guo et al., 2025), but use AdaGrad as an adaptive method. Additionally, as we discuss in Section 6 and ablate in Appendix D.3, DRO benefits from an optimistic update, making our proposed ALSO more practical for this problem than the non-optimistic methods used in (Yang et al., 2022; Guo et al., 2025). In summary, previous works (relying on Euclidean projection and a non-optimistic step) are less suitable for our problem, and more importantly, their analysis does not cover our more challenging case involving non-Euclidean geometry with KL-divergence.

## 5 EXPERIMENTS

We evaluate ALSO in several setups characterized by significant data heterogeneity. Specifically:

- **Learning from Unbalanced Data** (Section 5.1). We evaluate ALSO in an extremely class-imbalanced setup. Here, we assign weights to individual objects (i.e., no grouping is used).
- **Tabular DL** (Section 5.2). Tabular data is central to many real-world industrial problems and is often characterized by complex data heterogeneity, such as heavy-tailed and non-symmetric targets, extreme distributional shifts, and class imbalance (see Table 3 for details). In this setup, we assign weights to individual objects (i.e., no grouping is used).
- **Robust Training to Adversarial Attacks** (Section 5.3). The considered attacks vary in strength, which makes some easier to defend against than others. In this task, we assign weights to the attacks rather than to individual objects (i.e., grouping is used).
- **Distributed Training** (Section 5.4). Data heterogeneity is a well-known problem in distributed training, making it a natural setting to evaluate ALSO. Here, we assign weights to the workers instead of the individual objects (i.e., grouping is used).

- **Split Learning** (Section 5.5). The heterogeneity arises from split learning formulation: model with shared encoder trains on different tasks. In this experiment, we assign weights to each class, not to individual objects (i.e., grouping is used).

We compare `ALSO` with standard DL baselines, including vanilla `SGD` with momentum (Amari, 1993) and `AdamW`. We also consider several DRO methods that tackle problems similar to ours. We use both classical DRO methods like `Spectral Risk` (Mehta et al., 2023), and state-of-the-art methods such as `DRAGO` (Mehta et al., 2024) (noted for fast convergence), `FastDRO` (Levy et al., 2020) (a scalable method), `RECOVER` (Qi et al., 2021) (a non-convex method). In addition, we include standard imbalance-mitigation training schemes: `Upsampling` (Kahn & Marshall, 1953), `Static Weights` (He & Garcia, 2009), `Focal Loss` Lin (2017), and `Class-Balanced Loss` Cui et al. (2019) (see Appendix C.1 for details). All the methods are discussed in Section 1. Baselines were implemented using official code when suitable, or based on the paper otherwise. Details on hyperparameter tuning can be found in Appendix C. In short, all methods were tuned for the same number of iterations using either the Optuna package (Akiba et al., 2019) or a grid search. To reduce the hyperparameter search space, we fix $\alpha = 1$. This decision is supported by theory (see (Popov, 1980)) and prior empirical studies, which have shown that setting $\alpha$ near 1 is an effective choice (Mertikopoulos et al., 2018). We use uniform regularization $\hat{\pi}$ in all experiments, except Section 5.1. We provide reccomendations on $\hat{\pi}$ and hyperparameters selection in Appendix B.

## 5.1 LEARNING FROM UNBALANCED DATA

The purpose of this experiment is to demonstrate that `ALSO` can perform effectively in scenarios where the training dataset suffers from class imbalance. We consider a classification task on the CIFAR-10 dataset (Krizhevsky et al., 2009) using the ResNet-18 model (He et al., 2016). To simulate class imbalance, the ten original classes in the dataset are grouped into two based on the parity of its class. Subsequently, a proportion of samples from the second class is removed from both the training and validation datasets. Importantly, the test dataset, used to compute performance metrics, remains balanced. To quantify the class imbalance, we introduce the unbalanced coefficient (uc), which specifies the ratio of samples between the first and second classes as: $\#\,1\,\text{class}/\#\,2\,\text{class} = \text{uc}$, where $\#$ is the number of samples in the corresponding classes. For this experiment, we consider the values $\text{uc} \in \{1, 2, 5, 10, 20, 30, 40, 50\}$. The results of the experiment are presented in Figure 1. We observe that the proposed method `ALSO` outperforms all the compared baselines. The performance difference is particularly noticeable for large values of the unbalanced coefficient ($\geq 30$), where one class significantly outweighs the other.

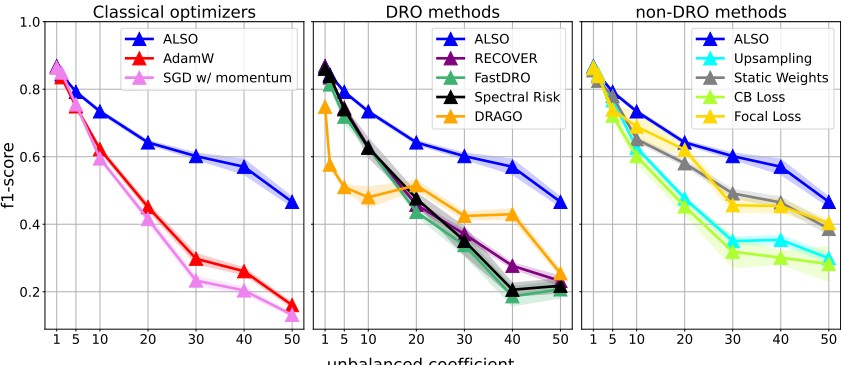

Figure 1: Performance comparison of optimization techniques designed for training in the presence of class imbalance The final f1-score was averaged over 20 runs, see Appendix C.2 for details.

## 5.2 TABULAR DEEP LEARNING

We evaluate the training procedure over 14 tabular datasets from (Gorishniy et al., 2024b; Rubachev et al., 2024) and 15 runs on them. Notably, the selected datasets possess characteristics particularly relevant for DRO methods: significant distribution shift between train-test splits, class imbalance, or

heavy-tailed target distributions in regression tasks. As a model, we choose MLP-PLR (Gorishniy et al., 2022) as it is a strong baseline in the tabular DL field. Detailed dataset characteristics, hyperparameter tuning procedures, and training specifications can be found in Appendix C.3. The results of the algorithms comparison are presented in Table 1. ALSO demonstrates the best performance on the most datasets and can be considered as an alternative to both conventional DL methods and specialized DRO methods.

Table 1: Performance comparison of ALSO, AdamW with uniform weights and *static weights* and Distributionally Robust Optimization methods – DRAGO, Spectral Risk, FastDRO, RECOVER on tabular Deep Learning datasets. Bold entries represent the best method on each dataset according to mean, underlined entries represent methods, which performance is best with standard deviations over 15 runs. Metric is written near dataset name, ↑ means that higher values indicate better performance, ↓ means otherwise.

| Dataset | ALSO | AdamW | DRAGO | Spectral Risk | FastDRO | RECOVER | Static Weights |
|---|---|---|---|---|---|---|---|
| Weather (RMSE ↓) | **1.4928 ± 0.0042** | 1.5208 ± 0.0037 | 1.5803 ± 0.0103 | 1.5189 ± 0.0047 | 1.5184 ± 0.0041 | 1.5547 ± 0.0034 | 1.5161 ± 0.0046 |
| Ecom Offers (ROC-AUC ↑) | 0.5976 ± 0.0020 | 0.5810 ± 0.0039 | **0.5983 ± 0.0019** | 0.5796 ± 0.0034 | 0.5900 ± 0.0126 | 0.5859 ± 0.0031 | 0.5803 ± 0.0033 |
| Cooking Time (RMSE ↓) | **0.4806 ± 0.0003** | 0.4813 ± 0.0003 | 0.4843 ± 0.0008 | 0.4810 ± 0.0004 | 0.4809 ± 0.0004 | 0.4813 ± 0.0006 | 0.4818 ± 0.0006 |
| Maps Routing (RMSE ↓) | **0.1612 ± 0.0001** | 0.1618 ± 0.0002 | 0.1651 ± 0.0005 | 0.1619 ± 0.0003 | 0.1620 ± 0.0003 | 0.1621 ± 0.0003 | 0.1617 ± 0.0002 |
| Homesite Insurance (ROC-AUC ↑) | **0.9632 ± 0.0003** | 0.9621 ± 0.0005 | 0.9536 ± 0.0018 | 0.9609 ± 0.0005 | 0.9614 ± 0.0008 | 0.9612 ± 0.0005 | 0.9619 ± 0.0003 |
| Delivery ETA (RMSE ↓) | **0.5513 ± 0.0020** | 0.5519 ± 0.0017 | 0.5555 ± 0.0016 | 0.5528 ± 0.0013 | 0.5528 ± 0.0017 | 0.5551 ± 0.0035 | 0.5555 ± 0.0031 |
| Homecredit Default (ROC-AUC ↑) | **0.8585 ± 0.0012** | 0.8579 ± 0.0012 | 0.8463 ± 0.0013 | 0.8575 ± 0.0012 | 0.8579 ± 0.0014 | 0.8576 ± 0.0011 | 0.8557 ± 0.0012 |
| Sberbank Housing (RMSE ↓) | **0.2424 ± 0.0024** | 0.2434 ± 0.0027 | 0.2694 ± 0.0070 | 0.2453 ± 0.0036 | 0.2458 ± 0.0044 | 0.2589 ± 0.0093 | 0.2465 ± 0.0080 |
| Black Friday (RMSE ↓) | **0.6842 ± 0.0004** | 0.6864 ± 0.0005 | 0.7011 ± 0.0040 | 0.6861 ± 0.0004 | 0.6861 ± 0.0003 | 0.6963 ± 0.0012 | 0.6870 ± 0.0008 |
| Microsoft (RMSE ↓) | **0.7437 ± 0.0004** | 0.7442 ± 0.0003 | 0.7496 ± 0.0010 | 0.7441 ± 0.0003 | 0.7448 ± 0.0004 | 0.7486 ± 0.0002 | 0.7467 ± 0.0004 |
| California Housing (RMSE ↓) | 0.4495 ± 0.0046 | **0.4602 ± 0.0042** | 0.6326 ± 0.2073 | 0.4681 ± 0.0050 | 0.4639 ± 0.0024 | 0.4787 ± 0.0042 | 0.4651 ± 0.0040 |
| Churn Modeling (ROC-AUC ↑) | **0.8666 ± 0.0027** | 0.8616 ± 0.0015 | 0.7960 ± 0.0010 | 0.8626 ± 0.0003 | 0.8622 ± 0.0020 | 0.8604 ± 0.0033 | 0.8249 ± 0.0073 |
| Adult (ROC-AUC ↑) | 0.8699 ± 0.0001 | 0.8688 ± 0.0012 | 0.7640 ± 0.0014 | 0.8687 ± 0.0009 | **0.8702 ± 0.0009** | 0.8683 ± 0.0013 | 0.8498 ± 0.0051 |
| Higgs Small (ROC-AUC ↑) | 0.7280 ± 0.0009 | 0.7274 ± 0.0017 | 0.6263 ± 0.0573 | **0.7282 ± 0.0021** | 0.7282 ± 0.0009 | 0.7267 ± 0.0013 | 0.7222 ± 0.0022 |

## 5.3 ROBUST TRAINING TO ADVERSARIAL ATTACKS

In this section, we compare ALSO with baselines on the task of robust training of DL model (Madry et al., 2017). At the first stage, a small CNN (LeCun et al., 1998) is trained with AdamW for 1 epoch on the MNIST dataset (LeCun et al., 2010). Then this pretrained model is trained with adversarial attacks (various transformations from torchvision (Marcel & Rodriguez, 2010), and the FGSM attack (Musa et al., 2021)) to obtain a more robust model. As a criterion for the quality of the models we use: MeanAccuracy $= \frac{1}{m} \sum_{i=1}^{m}$ Accuracy(Attack$_i$) and MinAccuracy $= \min_{i=1}^{m}$ Accuracy(Attack$_i$), where Attack$_i$ denotes the quality on the test dataset with the $i$-th attack. The first metric effectively captures overall model robustness, while the second one measures worst-case model performance. In this section, we slightly change the pipeline of the DRO algorithms; namely, at each iteration $k$ we sample the index $i \sim \text{Cat}(\pi^k)$, that corresponds $i$-th attack. During AdamW training, we sample $i$ from a uniform distribution. Experimental results (see Figure 2) demonstrate that ALSO outperforms both AdamW and DRO baselines.

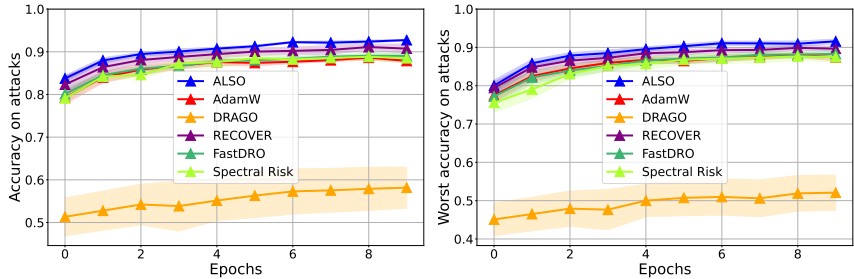

Figure 2: Comparison of mean accuracy (left) and min accuracy (right) over attacks of ALSO with other baselines on the test dataset. See details in Appendix C.4

## 5.4 Distributed Training

In this experiment, we consider the problem (1) as a distributed optimization problem, where $n$ workers have their own local data on the device. We focus on the case where gradient updates are compressed before being sent to the server. We consider the formulation (4), in which $\pi_i$ is no longer the weight of object $i$, but the weight of worker $i$, and accordingly, the larger $\pi_i$ is, the more worker $i$ will transmit information to the server. We return to ResNet-18 (He et al., 2016) on the CIFAR-10 dataset (Krizhevsky et al., 2009), where Perm-K (Szlendak et al., 2021) is chosen as the compressor. In all DRO methods, each worker transmits a personalized fraction $\pi_i$ of gradient coordinates to the server, which generalizes the Perm-K approach. As

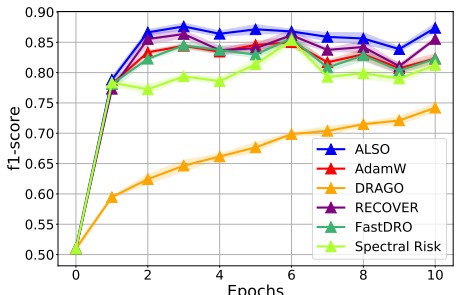

Figure 3: Comparison of f1-score of `ALSO` with other baselines on the distributed problem. See details in Appendix C.5

shown in Figure 3, applying the `ALSO` algorithm in the distributed setup demonstrates superiority over all baselines.

## 5.5 Split Learning

In this section, we compare `ALSO` with baselines in the Split Learning task (Vepakomma et al., 2018). The idea of split learning is to train a shared encoder across multiple tasks distributed over different workers, while maintaining independent heads for each task's predictions (Thapa et al., 2021; Kim et al., 2020). We use the ResNet-18 (He et al., 2016) without pretrained weights and simulate a scenario where a new worker joins the training process with the Flowers102 dataset (Nilsback & Zisserman, 2008), while training is already started on the Food101 dataset (Bossard et al., 2014). To enhance the performance of the worker that joins the training process at a later stage, we assign class-specific weights for both datasets. We compare `ALSO` optimizer and baselines by measuring Accuracy@5 on both datasets (see Figure 4). The results show that `ALSO` outperforms all other methods in terms of faster and more stable convergence, as well as better final metrics. Additional details are provided in Appendix C.6.

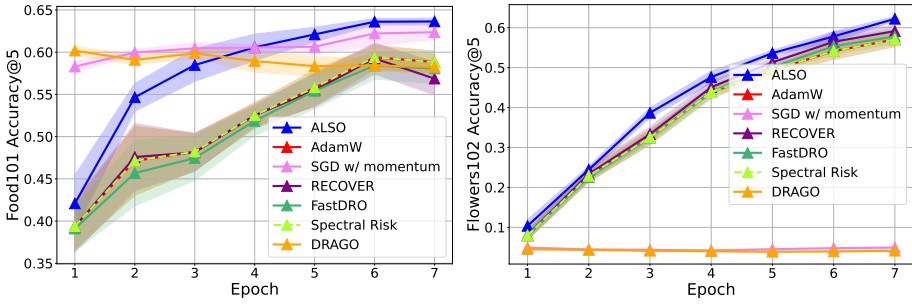

Figure 4: Metrics comparison for models trained with `ALSO`, `AdamW`, `SGD` and Distributionally Robust Optimization methods: `DRAGO`, `Spectral Risk`, `FastDRO`, `RECOVER` on Flowers102 and Food101 datasets. C.6.

## 6 Ablation Study Summary

Due to space constraints, our full ablation studies are presented in Appendix D. Key findings:

- **Computational Overhead** (Section D.1). `ALSO`'s overhead is insignificant compared to training with `AdamW`.
- **Hyperparameter Sensitivity** (Section D.2). `ALSO` is stable across a wide range of hyperparameters $(\gamma_\pi, \lambda)$, indicating it requires minimal tuning.
- **Design Choices** (Section D.3). We validate that our design, including momentum $(\alpha)$ and a non-adaptive $\pi$ update (Alg. 1), is a robust and effective design choice.

- **Tuning Comparison** (Section D.4). We show `ALSO`'s performance gain is not due to better hyperparameter tuning by running `AdamW` with its parameters.

## 7    LIMITATIONS

The main limitation of our approach is the problem we tackle. If one has data heterogeneity or needs distributional robustness, `ALSO` application is reasonable, otherwise it seems redundant to apply any DRO method. The method's effectiveness relies on meaningful data grouping; if groups are formed arbitrarily, `ALSO` is unlikely to provide gains. Additionally, usage of DRO methods raises such societal risks as bias and fairness (distribution robustness could inadvertently amplifying biases present in the data), ethical trade-offs (balancing groups' interests involves ethical judgments that must be made transparently). While DRO methods in general and `ALSO` in particular offer significant potential to the DL community, their integration requires careful consideration of these risks.

## 8    RELATED WORK

**Adaptive methods.** Adaptive optimization is central to modern Deep Learning, where methods such as Adagrad (Streeter & McMahan, 2010; Duchi et al., 2011), RMSProp (Tieleman, 2012), and Adam (Kingma, 2014) improve training by adjusting learning rates based on gradient history. Numerous variants extend Adam, e.g., NAdam (Dozat, 2016), AMSGrad (Reddi et al., 2019), AdamW (Loshchilov, 2017). However, these methods target minimization, while many DL problems are more naturally expressed as saddle-point formulations, which require different techniques (Browder, 1966; Nemirovski, 2004; Korpelevich, 1976; Popov, 1980). Recent works (Daskalakis et al., 2017; Gidel et al., 2018; Mertikopoulos et al., 2018; Chavdarova et al., 2019; Liang & Stokes, 2019; Peng et al., 2020) adapted Adam-like schemes to these settings, demonstrating strong empirical results but relying on limited theory. More rigorous studies have since appeared, e.g. AdaGrad variants (Liu et al., 2019), Adam-type analyses (Dou & Li, 2021), and scaled adaptive methods (Beznosikov et al., 2022). Nonetheless, existing results largely focus on the convex-concave Euclidean case and are insufficient for addressing non-convex, distributionally robust objectives such as our formulation (4).

**Weighting in Deep Learning.** The idea of weighting each training example has been well studied in the literature (Byrd & Lipton, 2019). Basic examples of these techniques are the classical method in statistics – importance sampling (Kahn & Marshall, 1953) – and AdaBoost (Freund & Schapire, 1997), where harder examples are selected to train subsequent classifiers. The main applications of loss weighting are learning from unbalanced data (He & Garcia, 2009; Lin, 2017), continual learning, which often involves re-weighting past and current samples to ensure that earlier knowledge is not forgotten (Aljundi et al., 2019). Another application is making the training process more stable and robust (Pang et al., 2019; Bi et al., 2022; Kendall & Gal, 2017; Ren et al., 2018). There are different approaches for weights assignment: based on specific tasks (Pang et al., 2019; Bi et al., 2022), use heuristics for weighting (Lin, 2017; Dong et al., 2017), employ a meta-learning approach (Ren et al., 2018; Jiang et al., 2018). Another aspect is non-uniform sampling, which selects examples with varying probabilities to improve optimization. For instance, it has improved convergence in randomized Kaczmarz methods (Needell et al., 2015), enhanced stochastic optimization in prox-SMD/SDCA algorithms (Zhao & Zhang, 2015), and been used in SGD variants based on individual example loss (Loshchilov & Hutter, 2015).

## 9    CONCLUSION

This paper introduces `ALSO`, an adaptive optimizer designed to bridge the gap between Distributionally Robust Optimization (DRO) and practical Deep Learning. By incorporating adaptive updates and support for standard batching even with group-based weighting, `ALSO` effectively addresses the common need to handle data heterogeneity. We provide theoretical convergence guarantees in a stochastic, non-convex setting and demonstrate through extensive experiments across diverse tasks with different challenging types of heterogeneity that `ALSO` consistently outperforms both standard DL and existing DRO methods. Our work establishes `ALSO` as a powerful and practical tool for improving the robustness and performance of Deep Learning models in challenging, heterogeneous scenarios.

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

## A  SAMPLING VARIANTS

In this section, we explore several object sampling strategies for ALSO and demonstrate that each produces unbiased estimates of the gradients for problem (4). For clarity in our analysis of unbiasedness, we consider a batch size $B = 1$ (since batch elements are sampled independently), and we omit iteration indices, using notation $(i, j)$ to represent object indices.

**Uniform Sampling Across All Objects**. We first examine the sampling approach presented in Lines 4, 5, 7 of Algorithm 1. Here, a pair $(i, j)$ is sampled with probability $\frac{1}{n}$. This yields:

$$\mathbb{E}g = \mathbb{E}\left(c\pi_i \nabla f_{i,j}(\theta)\right) = \sum_{(i,j)} \frac{c}{n} \pi_i \nabla f_{i,j}(\theta) = \sum_{i=1}^{c} \pi_i \left(\frac{c}{n} \sum_{j=1}^{n_i} \nabla f_{i,j}(\theta)\right)$$

$$\mathbb{E}p = \mathbb{E}\left(ce_i \cdot f_{i,j}(\theta)\right) = \sum_{(i,j)} \frac{c}{n} e_i \cdot f_{i,j}(\theta) = \sum_{i=1}^{c} e_i \left(\frac{c}{n} \sum_{j=1}^{n_i} f_{i,j}(\theta)\right)$$

Now let us compute the variance bound:

$$\mathbb{E}_{k,l} \| c\pi_k \nabla f_{k,l}(\theta) - \sum_{i=1}^{c} \pi_i \left(\frac{c}{n} \sum_{j=1}^{n_i} \nabla f_{i,j}(\theta)\right) \|^2 =$$

$$= \sum_{(k,l)} \frac{1}{n} \| c\pi_k \nabla f_{k,l}(\theta) - \sum_{i=1}^{c} \pi_i \left(\frac{c}{n} \sum_{j=1}^{n_i} \nabla f_{i,j}(\theta)\right) \|^2 =$$

$$= \sum_{(k,l)} \frac{1}{n} \| \frac{c}{n} \sum_{i=1}^{c} \sum_{j=1}^{n_i} \left(\pi_i \nabla f_{i,j}(\theta) - \pi_k \nabla f_{k,l}(\theta)\right) \|^2 \leq$$

$$\leq \sum_{(k,l)} \frac{c^2}{n^3} \left(\sum_{i=1}^{c} \sum_{j=1}^{n_i} \| \pi_i \nabla f_{i,j}(\theta) - \pi_k \nabla f_{k,l}(\theta) \|\right)^2 \leq$$

$$\leq \sum_{(k,l)} \frac{c^2}{n^3} \left(\sum_{i=1}^{c} \sum_{j=1}^{n_i} \left(\pi_i \| \nabla f_{i,j}(\theta) \| + \pi_k \| \nabla f_{k,l}(\theta) \|\right)\right)^2$$

Since $\|\nabla f_{i,j}(\theta)\| \leq K_{i,j} \leq \max_{i,j} K_{i,j} =: K$:

$$\mathbb{E}_{k,l} \| c\pi_k \nabla f_{k,l}(\theta) - \sum_{i=1}^{c} \pi_i \left(\frac{c}{n} \sum_{j=1}^{n_i} \nabla f_{i,j}(\theta)\right) \|^2 \leq \sum_{(k,l)} \frac{c^2}{n^3} \left(\sum_{i=1}^{c} \sum_{j=1}^{n_i} \left(\pi_i + \pi_k\right) 2K\right)^2 =$$

$$= \sum_{(k,l)} \frac{c^2}{n^3} \left( \sum_{i=1}^{c} (\pi_i + \pi_k) \, 2 n_i K \right)^2$$

Using Cauchy-Schwarz inequality:

$$\sum_{(k,l)} \frac{c^2}{n^3} \left( \sum_{i=1}^{c} (\pi_i + \pi_k) \, 2 n_i K \right)^2 \leq \sum_{(k,l)} \frac{4 c^2 K^2}{n^3} \left( \sum_{i=1}^{c} (\pi_i + \pi_k)^2 \sum_{i=1}^{c} n_i^2 \right)$$

Since $(a+b)^2 \leq 2a^2 + 2b^2$:

$$\sum_{(k,l)} \frac{4 c^2 K^2}{n^3} \left( \sum_{i=1}^{c} (\pi_i + \pi_k)^2 \sum_{i=1}^{c} n_i^2 \right) \leq \sum_{(k,l)} \frac{8 c^2 K^2 \sum_{i=1}^{c} n_i^2}{n^3} \sum_{i=1}^{c} \left( \pi_i^2 + \pi_k^2 \right) =$$

$$= \sum_{(k,l)} \pi_k^2 \frac{8 c^3 K^2 \sum_{i=1}^{c} n_i^2}{n^3} \sum_{i=1}^{c} \pi_i^2$$

Since $p_i \in \Delta_{c-1} \Rightarrow \sum_{i=1}^{c} \pi_i^2 \leq 1$:

$$\sum_{(k,l)} \pi_k^2 \frac{8 c^3 K^2 \sum_{i=1}^{c} n_i^2}{n^3} \sum_{i=1}^{c} \pi_i^2 \leq \sum_{k=1}^{c} \sum_{l=1}^{n_k} \pi_k^2 \frac{8 c^3 K^2 \sum_{i=1}^{c} n_i^2}{n^3} \leq \frac{8 c^3 K^2 \sum_{i=1}^{c} n_i^2}{n^3} \sum_{k=1}^{c} n_k \pi_k^2 \leq$$

$$\leq \frac{8 c^3 K^2 \sum_{i=1}^{c} n_i^2}{n^3} \sum_{k=1}^{c} n_k = \frac{8 c^3 K^2 \sum_{i=1}^{c} n_i^2}{n^2}$$

Now we will similarly consider $p$:

$$\mathbb{E}_{k,l} \| c e_k f_{k,l}(\theta) - \sum_{i=1}^{c} \left( \frac{c}{n} e_i \sum_{j=1}^{n_i} f_{i,j}(\theta) \right) \|^2 = \sum_{(k,l)} \frac{1}{n} \| \frac{c}{n} \sum_{i=1}^{c} \sum_{j=1}^{n_i} (e_k f_{k,l}(\theta) - e_i f_{i,j}(\theta)) \|^2 \leq$$

$$\leq \sum_{(k,l)} \frac{c^2}{n^3} \left( \sum_{i=1}^{c} \sum_{j=1}^{n_i} \| e_k f_{k,l}(\theta) - e_i f_{i,j}(\theta) \| \right)^2 =$$

$$= \sum_{(k,l)} \frac{c^2}{n^3} \left( \sum_{(i,j)} \| e_k f_{k,l}(\theta) - e_k f_{k,l}(\theta^*) + e_k f_{k,l}(\theta^*) - e_i f_{i,j}(\theta^*) + e_i f_{i,j}(\theta^*) - e_i f_{i,j}(\theta) \| \right)^2$$

where $\theta^* = \arg\min_{\theta \in \mathbb{R}^d} \max_{\pi \in \Delta_{c-1}} h(\theta, \pi)$. Thus:

$$\mathbb{E}_{k,l} \| c e_k f_{k,l}(\theta) - \sum_{i=1}^{c} \left( \frac{c}{n} e_i \sum_{j=1}^{n_i} f_{i,j}(\theta) \right) \|^2 \leq \sum_{(k,l)} \frac{c^2}{n^3} (\sum_{(i,j)} \| e_k f_{k,l}(\theta) - e_k f_{k,l}(\theta^*) \| + \| e_k f_{k,l}(\theta^*) \|$$

$$+ \| e_i f_{i,j}(\theta^*) \| + \| e_i f_{i,j}(\theta^*) - e_i f_{i,j}(\theta) \|)^2$$

Since $\| e_i f_{i,j}(\theta^*) - e_i f_{i,j}(\theta) \| = \| e_i (f_{i,j}(\theta^*) - f_{i,j}(\theta)) \| = | f_{i,j}(\theta^*) - f_{i,j}(\theta) | \leq K_{i,j} \| \theta^* - \theta \| \leq K \| \theta^* - \theta \|$ and $\| e_k f_{k,l}(\theta^*) \| = | f_{k,l}(\theta^*) | \leq \max_{k,l} | f_{k,l}(\theta^*) | =: G$:

$$\mathbb{E}_{k,l} \| c e_k f_{k,l}(\theta) - \sum_{i=1}^{c} \left( \frac{c}{n} e_i \sum_{j=1}^{n_i} f_{i,j}(\theta) \right) \|^2 \leq \sum_{(k,l)} \frac{c^2}{n^3} \left( \sum_{(i,j)} (2G + 2K \| \theta - \theta^* \|) \right)^2 =$$

$$= \sum_{(k,l)} \frac{c^2}{n^3} \left( n (2G + 2K \| \theta - \theta^* \|) \right)^2 = \frac{c^2}{n^3} n^3 (2G + 2K \| \theta - \theta^* \|)^2 =$$

$$= c^2 (2G + 2K \| \theta - \theta^* \|)^2 \leq 8 c^2 (G^2 + K^2 \| \theta - \theta^* \|)$$

Thus:

$$\sigma^2 \leq \max\left\{\frac{8c^3K^2\sum_{i=1}^c n_i^2}{n^2}, 8c^2(G^2 + K^2\|\theta - \theta^*\|)\right\} = \mathcal{O}(K^2)$$

**Two-Stage Group-Object Sampling**. An alternative approach involves a two-stage sampling process: first sample a group index with uniform probability $\frac{1}{c}$, then sample an object from this group with uniform probability $\frac{1}{n_i}$. This gives a probability $\frac{1}{cn_i}$ for selecting object $(i,j)$. To maintain unbiased gradient estimates, we modify the scaling in Lines 5, 7 as follows:

$$g = \frac{c^2 n_i}{n}\pi_i\nabla f_{i,j}(\theta)$$

$$p = \frac{c^2 n_i}{n}e_i \cdot f_{i,j}(\theta)$$

Then

$$\mathbb{E}g = \mathbb{E}\left(\frac{c^2 n_i}{n}\pi_i\nabla f_{i,j}(\theta)\right) = \sum_{i=1}^c\sum_{j=1}^{n_i}\frac{c^2 n_i}{n}\pi_i\nabla f_{i,j}(\theta)\frac{1}{cn_i} = \sum_{i=1}^c\pi_i\left(\frac{c}{n}\sum_{j=1}^{n_i}\nabla f_{i,j}(\theta)\right)$$

$$\mathbb{E}p = \mathbb{E}\left(\frac{c^2 n_i}{n}e_i \cdot f_{i,j}(\theta)\right) = \sum_{i=1}^c\sum_{j=1}^{n_i}\frac{c^2 n_i}{n}e_i \cdot f_{i,j}(\theta)\frac{1}{cn_i} = \sum_{i=1}^c e_i\left(\frac{c}{n}\sum_{j=1}^{n_i}\nabla f_{i,j}(\theta)\right)$$

**Probability-Weighted Group Sampling**. A third variant samples group $i$ according to its weight $\pi_i$, i.e. $i \sim Cat(\pi)$ followed by uniform sampling of $j$ with probability $\frac{1}{n_i}$. This gives a selection probability of $\frac{\pi_i}{n_i}$ for pair $(i,j)$. We adjust the scaling factors as:

$$g = \frac{cn_i}{n}\nabla f_{i,j}(\theta)$$

$$p = \frac{cn_i}{n\pi_i}e_i \cdot f_{i,j}(\theta)$$

Then

$$\mathbb{E}g = \mathbb{E}\left(\frac{cn_i}{n}\nabla f_{i,j}(\theta)\right) = \sum_{i=1}^c\sum_{j=1}^{n_i}\frac{cn_i}{n\pi_i}\pi_i\nabla f_{i,j}(\theta)\frac{\pi_i}{n_i} = \sum_{i=1}^c\pi_i\left(\frac{c}{n}\sum_{j=1}^{n_i}\nabla f_{i,j}(\theta)\right)$$

$$\mathbb{E}g = \mathbb{E}\left(\frac{cn_i}{n\pi_i}e_i \cdot f_{i,j}(\theta)\right) = \sum_{i=1}^c\sum_{j=1}^{n_i}\frac{cn_i}{n\pi_i}e_i \cdot f_{i,j}(\theta)\frac{\pi_i}{n_i} = \sum_{i=1}^c e_i\left(\frac{c}{n}\sum_{j=1}^{n_i}f_{i,j}(\theta)\right)$$

**Note.** We employ the third sampling technique in our Robust Training experiments (see Section 5.3). To see it let $n_i = k$ $\forall i$, where $k$ is the dataset length. In this scenario $n = c \cdot k$ is effective dataset size (i.e. attacked object can be considered as separate object). Since $j$ is independent of $i$ now we can reverse sampling order: first sample $j$, then sample $i$. This implementation – sample objects and then sample attacks for them – allows seamlessly integrate `ALSO` into a standard training procedure.

## B  PRACTICAL RECOMMENDATIONS

$\hat{\pi}$ **selection.** Our general recommendation for prior selection is as follows: a uniform prior is a safe default when domain knowledge for setting static weights is unavailable. For those already using `AdamW`, `ALSO` can be easily incorporated by initializing it with uniform weights. However, if there are established community practices for initializing static weights for a particular task (e.g., based on class frequencies), we recommend using such domain-informed priors, as they can further improve performance.

**Hyperparameters**. For practitioners, we recommend using the hyperparameters and search spaces provided in our work (see Appendix C) rather than deriving them directly from the theory, as we have shown these to yield strong empirical performance. Our goal with the theoretical analysis was to establish the formal soundness of `ALSO`. We aimed to prove that, unlike purely heuristic methods, our algorithm is guaranteed to converge to a stationary point in a standard non-convex stochastic setting. However, if one wants to use theory inspired batch size, we recommend to use gradient accumulations technique to fit into GPU memory.

## C    MISSING EXPERIMENT DETAILS

### C.1    BASELINES DESCRIPTION

Let us discuss described basic imbalance handling techniques. The first of these techniques is known as *upsampling* (Kahn & Marshall, 1953), the idea is to sample objects for gradient calculation at the current optimization step not uniformly, but proportionally to the class ratio of each object in the training dataset. For the $\hat{\pi}$ regularizer in the problem (4), we utilize this modified distribution instead of the vanilla uniform distribution $\mathcal{U}(\overline{1,n})$. This choice results in a significant improvement in the performance. The second technique is called *static weights* (He & Garcia, 2009). Its idea is similar to the previous method, however, instead of modifying the sampling distribution, objects are sampled uniformly. The class imbalance is then addressed by multiplying the loss function for each object by a weight equal to the inverse ratio of the number of objects belonging to that class in the training dataset. We also consider more advanced imbalance-aware losses. *Focal Loss* (Lin, 2017) down-weights well-classified examples and focuses the optimization on hard, typically minority, examples by modulating the standard cross-entropy with a factor that depends on the predicted probability. *Class-Balanced Loss* (Cui et al., 2019) instead re-weights each example using a factor inversely proportional to the "effective number" of samples in its class, which yields a more principled alternative to simple inverse-frequency weighting in long-tailed settings.

### C.2    UNBALANCED DATA DETAILS (SECTION 5.1)

**Data preprocessing.** For all optimizers the same preprocessing was used for fair comparison. We modified the images from CIFAR-10 train dataset with Normalizing and classical computer vision augmentations: Random Crop (Takahashi et al., 2019), Random Horizontally Flip.

**Training neural networks.** We use cross-entropy as the loss function. We do not apply learning rate schedules since we tune hyperparameters. We use a predefined batch size equal to $64$ and maximum number of epochs equal to $20$.

**Hyperparameter tuning.** Hyperparameter tuning is performed with the TPE sampler (200 iterations) with $5$ epoch from the Optuna package (Akiba et al., 2019). Hyperparameter tuning spaces for experiment are provided in Table 2.

| Parameter | Distribution |
|---|---|
| Learning rate | LogUniform$[1e\text{-}4, 1e\text{-}2]$ |
| Weight decay | LogUniform$[1e\text{-}6, 1e\text{-}2]$ |
| $\pi$-Learning rate ($\gamma_\pi$ from `ALSO`, used for `ALSO`, `DRAGO`) | LogUniform$[1e\text{-}5, 1e\text{-}3]$ |
| $\pi$-regularization ($\lambda$ from `ALSO`, used for `ALSO`, `DRAGO`, `RECOVER`, `Spectral Risk`) | LogUniform$[1e\text{-}3, 1]$ |

Table 2: The hyperparameter tuning space for unbalanced data experiment.

**Evaluation.** The tuned hyperparameters are evaluated under 20 random seeds. The mean test metric and its standard deviation over these random seeds are then used to compare algorithms as described in Section 5.1.

## C.3 Tabular Deep Learning Details (Section 5.2)

| Name | # Train | # Validation | # Test | # Num | # Bin | # Cat | Task type | Metric | Heterogeniety | Batch size |
|---|---|---|---|---|---|---|---|---|---|---|
| Sberbank Housing | 18 847 | 4 827 | 4 647 | 365 | 17 | 10 | Regression | RMSE | Heavy-tailed | 256 |
| Ecom Offers | 109 341 | 24 261 | 26 455 | 113 | 6 | 0 | Binclass | ROC AUC | Extreme shift | 1024 |
| Maps Routing | 160 019 | 59 975 | 59 951 | 984 | 0 | 2 | Regression | RMSE | - | 1024 |
| Homesite Insurance | 224 320 | 20 138 | 16 295 | 253 | 23 | 23 | Binclass | ROC AUC | Class imbalance | 1024 |
| Cooking Time | 227 087 | 51 251 | 41 648 | 186 | 3 | 3 | Regression | RMSE | Heavy-tailed | 1024 |
| Homecredit Default | 267 645 | 58 018 | 56 001 | 612 | 2 | 82 | Binclass | ROC AUC | High uncertainty | 1024 |
| Delivery ETA | 279 415 | 34 174 | 36 927 | 221 | 1 | 1 | Regression | RMSE | Non-symmetric | 1024 |
| Weather | 106 764 | 42 359 | 40 840 | 100 | 3 | 0 | Regression | RMSE | Non-symmetric | 1024 |
| Churn Modelling | 6 400 | 1 600 | 2 000 | 10 | 3 | 1 | Binclass | ROC AUC | Noisy data | 128 |
| California Housing | 13 209 | 3 303 | 4 128 | 8 | 0 | 0 | Regression | RMSE | Heavy-tailed | 256 |
| Adult | 26 048 | 6 513 | 16 281 | 6 | 1 | 8 | Binclass | ROC AUC | High uncertainty | 256 |
| Higgs Small | 62 751 | 15 688 | 19 610 | 28 | 0 | 0 | Binclass | ROC AUC | - | 512 |
| Black Friday | 106 764 | 26 692 | 33 365 | 4 | 1 | 4 | Regression | RMSE | Heavy-tailed | 512 |
| Microsoft | 723 412 | 235 259 | 241 521 | 131 | 5 | 0 | Regression | RMSE | - | 1024 |

Table 3: Properties of the datasets from (Gorishniy et al., 2024b; Rubachev et al., 2024). "# Num", "# Bin", and "# Cat" denote the number of numerical, binary, and categorical features, respectively

We mostly follow the experiment setup from (Gorishniy et al., 2024a). As such, most of the text below is copied from (Gorishniy et al., 2024a).

**Data preprocessing.** For each dataset, for all optimizers, the same preprocessing was used for fair comparison. For numerical features, by default, we used a slightly modified version of the quantile normalization from the Scikit-learn package (Pedregosa et al., 2011) (see the source code), with rare exceptions when it turned out to be detrimental (for such datasets, we used the standard normalization or no normalization). For categorical features, we used one-hot encoding. Binary features (i.e. the ones that take only two distinct values) are mapped to $\{0, 1\}$ without any further preprocessing.

**Training neural networks.** We use cross-entropy for classification problems and mean squared error for regression problems as loss function. We do not apply learning rate schedules. We do not use data augmentations. We apply global gradient clipping to $1.0$. For each dataset, we used a predefined dataset-specific batch size. We continue training until there are `patience` consecutive epochs without improvements on the validation set; we set `patience` $= 16$.

**Hyperparameter tuning.** In most cases, hyperparameter tuning is performed with the TPE sampler (100 iterations) from the Optuna package (Akiba et al., 2019). Hyperparameter tuning spaces for experiment are provided in Table 4.

**Evaluation.** On a given dataset, for a given model, the tuned hyperparameters are evaluated under multiple (in most cases, 15) random seeds. The mean test metric and its standard deviation over these random seeds are then used to compare algorithms as described in Table 3.

| Parameter | Distribution |
|---|---|
| # layers | UniformInt$[1, 5]$ |
| Width (hidden size) | UniformInt$[64, 1024]$ |
| Dropout rate | $\{0.0, \text{Uniform}[0.0, 0.5]\}$ |
| n_frequencies | UniformInt$[16, 96]$ |
| d_embedding | UniformInt$[16, 32]$ |
| frequency_init_scale | LogUniform$[1e\text{-}2, 1e1]$ |
| Learning rate | LogUniform$[3e\text{-}5, 1e\text{-}3]$ |
| Weight decay | $\{0, \text{LogUniform}[1e\text{-}4, 1e\text{-}1]\}$ |
| $\pi$-Learning rate ($\gamma_\pi$ from ALSO, used for ALSO, DRAGO) | LogUniform$[1e\text{-}5, 1e\text{-}3]$ |
| $\pi$-regularization ($\lambda$ from ALSO, used for ALSO, DRAGO, RECOVER, Spectral Risk) | LogUniform$[1e\text{-}3, 1]$ |
| Size (used for FastDRO) | Uniform$[0, 1]$ |
| n_draws (used for Spectral Risk) | LogUniform$[1e\text{-}3, 1]$ |

Table 4: The hyperparameter tuning space for tabular Deep Learning experiment.

## C.4 Robust Training to Adversarial Attacks (Section 5.3)

We provide a detailed description of the experimental pipeline employed in our study. Our approach is based on a modified version of the `ALSO` pipeline. Specifically, at each iteration, we sample an index $i$ from a categorical distribution parameterized by $\pi^k$, apply the corresponding attack, and then proceed with the `ALSO` step. For comparison, the baseline pipeline consists of standard optimization using the `AdamW` optimizer, where the index $i$ is sampled from a uniform distribution.

The general procedure for each algorithm can be summarized as follows:

1. Sample a mini-batch $(X_{\text{train}}^B, y_{\text{train}}^B)$ from the training set $X_{\text{train}}$.
2. Sample $i \sim \text{Categorical}(\pi^k)$ to select the attack for the batch, or sample $i$ from a uniform distribution in the baseline case.
3. Perform an optimizer step to update $\theta$ and $\pi$ (if required)

The hyperparameters used in our experiments are as follows: $\tau = 1$ (for DRO algorithms), $\gamma_\pi = 0.1$ for the `ALSO`, and a learning rate of $\text{lr} = 10^{-3}$ for all pipelines.

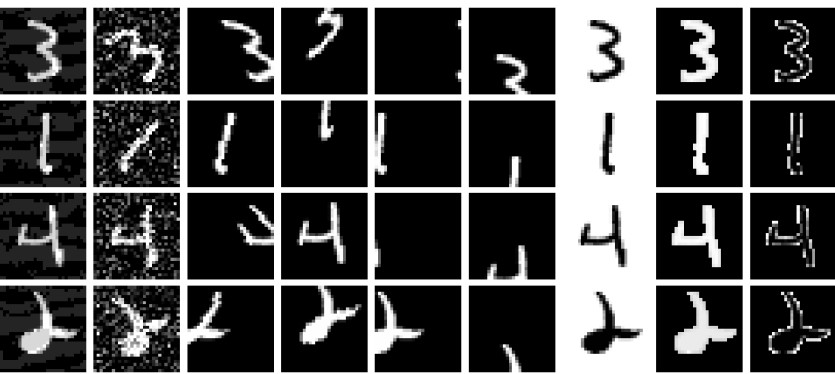

Figure 5: Examples of applied attacks to the test and train datasets.

## C.5 Distributed Training Details (Section 5.4)

**Data preprocessing.** For all optimizers the same preprocessing was used for fair comparison. We modified the images from CIFAR-10 train dataset with Normalizing and classical computer vision augmentations: Random Crop (Takahashi et al., 2019), Random Horizontally Flip.

**Parameter selection.** Different numbers of workers, different class distributions, and different class distributions among workers were considered during the experiments.

**Training neural networks.** We use cross-entropy as the loss function. We use a predefined batch size equal to 1024 and maximum number of epochs equal to 20.

## C.6 SPLIT LEARNING (SECTION 5.5)

**Split Learning Motivation.** The idea behind split learning is to train a shared encoder across multiple tasks distributed over different workers, while maintaining independent heads for each task's predictions (Thapa et al., 2021). This approach enables collaborative training without sharing raw data, enhancing privacy, and reduces computational and communication overhead, making it suitable for low-resource or budget-constrained settings (Kim et al., 2020).

**Experiment Details.** First, we train a ResNet-18 model on the Food101 dataset (Bossard et al., 2014) using the `AdamW` optimizer for 3 epochs. Next, we simulate the split learning process by introducing the Flowers102 dataset (Nilsback & Zisserman, 2008) into the training scheme. Training proceeds by alternating between datasets every epoch: one epoch on Food101, then one epoch on Flowers102, and so on. During each epoch, we train the shared encoder, while using separate linear heads for each dataset. For the DRO methods, we apply class weights for both datasets.

**Technical Details.** The default learning rate was set to $3 \times 10^{-4}$. Baseline hyperparameters were selected based on prior experiments and tuned over up to 5 iterations. The $\pi$-learning rate and $\pi$-decay parameters were kept at their default values of $1e$-$5$ and $1e$-$2$, respectively, without further tuning. All experiments were conducted on an NVIDIA Tesla V100 GPU.

**Comparison Details.** We compared the `ALSO` optimizer against baseline methods using the Accuracy@5 metric. Except for `DRAGO`, all DRO baselines improved accuracy on the newly introduced Flowers102 dataset at the expense of some accuracy loss on the original Food101 dataset, relative to `AdamW`. In contrast, `ALSO` outperformed `AdamW` on both datasets, demonstrating its ability to acquire out-of-domain knowledge without degrading performance on the initial task.

# D  ABLATION STUDY

## D.1  ALSO STEP TIME ANALYSIS

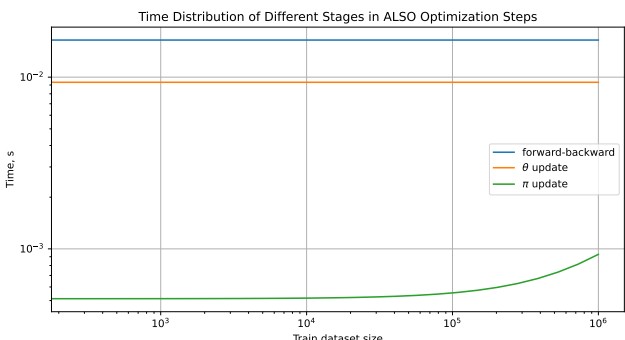

Figure 6: Time distribution over dataset size of three main parts of optimization process with ALSO: gradient computation (forward-backward), $\theta$ update and $\pi$ update. The trained model is ResNet-18 with batch size. Time of each part is averaged across $25$ training steps. We want to highlight, that gradient computations are required for all first order optimization methods, and this measurement is used only for comparison.

To analyze the time consumption of each component in the optimization process with ALSO, we conduct an experiment training ResNet-18 (He et al., 2016) with a fixed batch size of $64$ across various dataset sizes, measured time is averaged across $25$ iterations. This approach is chosen because while $\pi$ updates depend on dataset size, gradient computation and $\theta$ updates do not. We test dataset sizes up to 1 million samples, which exceeds our largest experimental dataset, which contains approximately $800000$ samples. The experiment was conducted on one NVIDIA GeForce RTX 2080 Ti GPU. We want to highlight, that gradient computations are required for all first order optimization methods, and this measurement is used only for comparison.

The results, presented in Figure 6, reveal a clear hierarchy in computational demands. Gradient computation (forward-backward passes) consistently requires significantly more time than both $\theta$ and $\pi$ updates across all dataset sizes, which is consistent with (Jiang et al., 2021). Furthermore, $\theta$ updates consistently demand more computational time than $\pi$ updates. This experiment leads to conclusion that the explicit weight vector update ($\pi$ update) is computationally negligible relative to the overall training step time.

## D.2  HYPERPARAMETERS SENSITIVITY

This ablation study examines ALSO's sensitivity to its $\pi$-specific hyperparameters: the $\pi$-learning rate ($\gamma_\pi$) and $\pi$-regularization ($\lambda$). We conducted full 2D sweeps for both parameters, fixing model weight learning rates and regularization to isolate their impact. Results from the imbalanced data setting (Section 5.1) show consistent performance across varying imbalance coefficients (Figure 7). Similarly, in Split Learning (Section 5.5), 2D sweeps confirm broad robustness on Food101 and Flowers102 datasets (Figure 8). Across all experiments, ALSO proves largely insensitive to $\gamma_\pi$ and $\lambda$ settings, suggesting strong performance is achievable without extensive tuning.

## D.3  DESIGN CHOICES

This section presents an empirical evaluation of key design choices in the proposed algorithm, focusing on the optimistic step and the non-adaptive update rule for the parameter $\pi$. We compare the performance of three algorithm variants:

1. Vanilla ALSO: The standard implementation of the proposed algorithm (Algorithm 1).

2. Descent-Ascent ALSO ($\alpha = 0$): A variant where the optimistic step is removed by setting the optimistic coefficient $\alpha$ to zero.

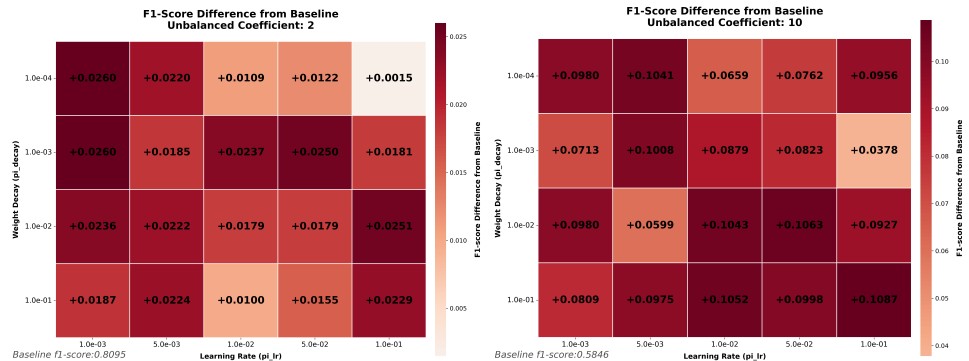

Figure 7: Robustness of `ALSO` to $\pi$-hyperparameters ($\Delta$F1-score vs. `AdamW` baseline). Each cell shows the F1-score difference between `ALSO` and `AdamW` with static weights (baseline), over a full 2D grid of $\pi$-learning rate ($\gamma_\pi$) and $\pi$-regularization ($\lambda$). All cells are red (positive $\Delta$F1), indicating that ALSO consistently outperforms the baseline across the entire grid and for different imbalance coefficients (2 and 10).

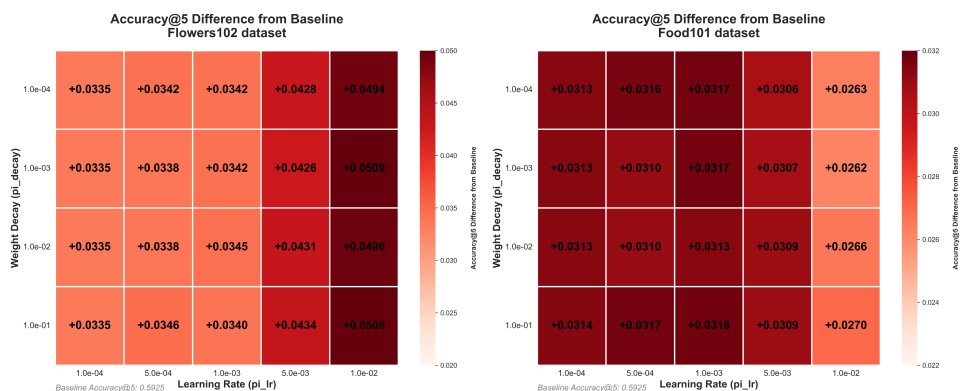

Figure 8: Robustness of `ALSO` to $\pi$-hyperparameters ($\Delta$F1-score vs. `AdamW` baseline). Each cell shows the F1-score difference between `ALSO` and `AdamW` with static weights (baseline), over a full 2D grid of $\pi$-learning rate ($\gamma_\pi$) and $\pi$-regularization ($\lambda$). All cells are red (positive $\Delta$F1), indicating that ALSO consistently outperforms the baseline across the entire grid and for both datasets from Split Learning section.

3. $\text{A}^\pi\text{LSO}$: A modified version of `ALSO` that employs the Adam optimizer for updating the weight vector $\pi$.

The algorithms were evaluated across three distinct experimental settings: Learning from Unbalanced Data (Section 5.1), Tabular Deep Learning (Section 5.2), and Split Learning (Section 5.5). The results are summarized in Figure 9, Table 5 and Figure 10.

The Descent-Ascent variant has a significantly lower performance compared to the other two algorithms, indicating the importance of the optimistic step. The $\text{A}^\pi\text{LSO}$ algorithm achieves comparable performance to vanilla `ALSO` in some scenarios (Table 5, Figure 10). However, in the Unbalanced Data experiment, $\text{A}^\pi\text{LSO}$ demonstrates degraded performance when the unbalanced coefficient is large ($\geq 10$).

Considering both performance and ease of implementation, we recommend vanilla `ALSO` as a robust baseline. While $\text{A}^\pi\text{LSO}$ can provide competitive results in certain settings, it introduces additional hyperparameters and computational overhead associated with the Adam optimizer for $\pi$. Therefore, $\text{A}^\pi\text{LSO}$ may be considered when sufficient computational resources are available for hyperparameter tuning and multiple experimental runs.

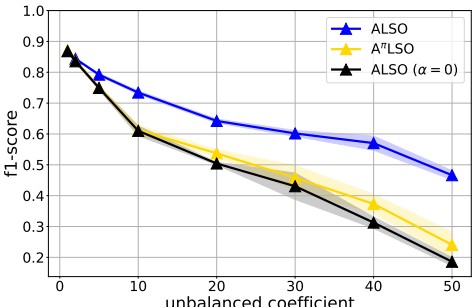

Figure 9: Performance comparison of `ALSO`, `ALSO` with $\alpha = 0$ (descent-ascent), and $\text{A}^\pi\text{LSO}$ (adaptive step over $\pi$) on the unbalanced CIFAR experiment from Section 5.1. Hyperparameter tuning is performed in the same manner as in the main experiment.

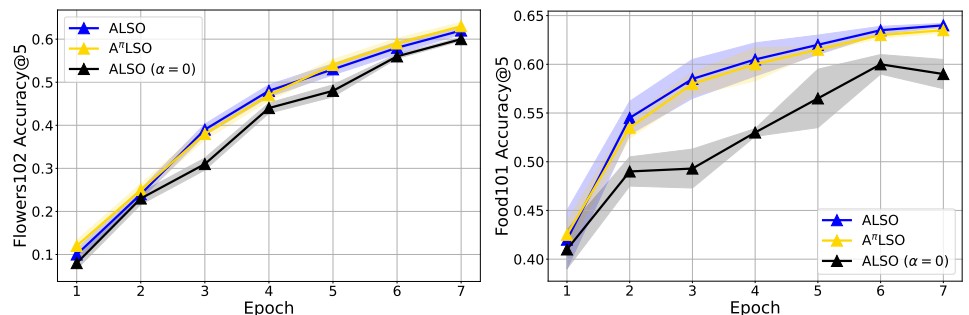

Figure 10: Performance comparison of `ALSO`, `ALSO` with $\alpha = 0$ (descent-ascent), and $\text{A}^\pi\text{LSO}$ (adaptive step over $\pi$) on the Split Learning experiment from Section 5.5

| Dataset | ALSO | ALSO $\alpha = 0$ | $\text{A}^\pi\text{LSO}$ |
|---|---|---|---|
| Weather (RMSE ↓) | **1.4928 ± 0.0042** | 1.5209 ± 0.0036 | 1.4967 ± 0.0066 |
| Ecom Offers (ROC-AUC ↑) | **0.5976 ± 0.0020** | 0.5975 ± 0.0020 | 0.5915 ± 0.0087 |
| Cooking Time (RMSE ↓) | **0.4806 ± 0.0003** | 0.4810 ± 0.0003 | **0.4806 ± 0.0004** |
| Maps Routing (RMSE ↓) | 0.1612 ± 0.0001 | 0.1613 ± 0.0002 | **0.1611 ± 0.0001** |
| Homesite Insurance (ROC-AUC ↑) | **0.9632 ± 0.0003** | 0.9630 ± 0.0004 | 0.9626 ± 0.0003 |
| Delivery ETA (RMSE ↓) | 0.5513 ± 0.0020 | 0.5536 ± 0.0030 | **0.5507 ± 0.0011** |
| Homecredit Default (ROC-AUC ↑) | **0.8587 ± 0.0012** | 0.8587 ± 0.0008 | 0.8587 ± 0.0011 |
| Sberbank Housing (RMSE ↓) | 0.2424 ± 0.0024 | 0.2457 ± 0.0044 | **0.2401 ± 0.0073** |
| Black Friday (RMSE ↓) | 0.6842 ± 0.0004 | 0.6843 ± 0.0013 | **0.6838 ± 0.0005** |
| Microsoft (RMSE ↓) | 0.7437 ± 0.0003 | 0.7435 ± 0.0003 | 0.7438 ± 0.0003 |
| California Housing (RMSE ↓) | 0.4495 ± 0.0046 | 0.4533 ± 0.0043 | **0.4455 ± 0.0032** |
| Churn Modeling (ROC-AUC ↑) | **0.8666 ± 0.0027** | 0.8597 ± 0.0076 | 0.8646 ± 0.0019 |
| Adult (ROC-AUC ↑) | **0.8699 ± 0.0001** | 0.8698 ± 0.0002 | 0.8698 ± 0.0014 |
| Higgs Small (ROC-AUC ↑) | 0.7280 ± 0.0009 | 0.7279 ± 0.0013 | **0.7288 ± 0.0012** |

Table 5: Performance comparison of `ALSO`, `ALSO` $\alpha = 0$ (descent-ascent) and $\text{A}^\pi\text{LSO}$ (adaptive step over $\pi$). The trained model is MLP-PLR (Gorishniy et al., 2022). Bold entries represent the best method on each dataset according to mean, underlined entries represent methods, which performance is best with standard deviations over 15 seeds taken into account. Metric is written near dataset name, ↑ means that higher values indicate better performance, ↓ means that lower values indicate better performance. Hyperparameter tuning is performed in the same manner as in the main experiment.

### D.4 TUNING COMPARISON

We evaluate `Adam` and `AdamW` with hyperparameters for `ALSO` to isolate effect of dynamic weights usage. The results are presented in Table 6. As we can see, the choice of hyperparameters, does not explain, why `ALSO` outperforms `Adam`.

| Dataset | Adam | AdamW | ALSO |
|---|---|---|---|
| Weather (RMSE ↓) | $1.5199 \pm 0.0034$ | $1.5199 \pm 0.0034$ | $\mathbf{1.4928 \pm 0.0042}$ |
| Ecom Offers (ROC-AUC ↑) | $\underline{0.5972 \pm 0.0020}$ | $0.5717 \pm 0.0020$ | $\mathbf{0.5976 \pm 0.0020}$ |
| Cooking Time (RMSE ↓) | $0.4810 \pm 0.0005$ | $0.4810 \pm 0.0005$ | $\mathbf{0.4806 \pm 0.0003}$ |
| Maps Routing (RMSE ↓) | $0.1617 \pm 0.0002$ | $0.1625 \pm 0.0002$ | $\mathbf{0.1612 \pm 0.0001}$ |
| Homesite Insurance (ROC-AUC ↑) | $0.9614 \pm 0.0003$ | $0.9593 \pm 0.0005$ | $\mathbf{0.9632 \pm 0.0003}$ |
| Delivery ETA (RMSE ↓) | $0.5550 \pm 0.0021$ | $0.5544 \pm 0.0014$ | $\mathbf{0.5513 \pm 0.0020}$ |
| Homecredit Default (ROC-AUC ↑) | $0.8581 \pm 0.0009$ | $0.8581 \pm 0.0009$ | $\mathbf{0.8585 \pm 0.0012}$ |
| Sberbank Housing (RMSE ↓) | $0.2457 \pm 0.0046$ | $0.2455 \pm 0.0047$ | $\mathbf{0.2424 \pm 0.0024}$ |
| Black Friday (RMSE ↓) | $\mathbf{0.6842 \pm 0.0006}$ | $0.6869 \pm 0.0006$ | $\mathbf{0.6842 \pm 0.0004}$ |
| Microsoft (RMSE ↓) | $\underline{0.7440 \pm 0.0002}$ | $0.7442 \pm 0.0003$ | $\mathbf{0.7437 \pm 0.0004}$ |
| California Housing (RMSE ↓) | $0.4554 \pm 0.0034$ | $0.4734 \pm 0.0038$ | $\mathbf{0.4495 \pm 0.0046}$ |
| Churn Modeling (ROC-AUC ↑) | $0.8620 \pm 0.0075$ | $0.8618 \pm 0.0038$ | $\mathbf{0.8666 \pm 0.0027}$ |
| Adult (ROC-AUC ↑) | $0.8693 \pm 0.0010$ | $0.8689 \pm 0.0009$ | $\mathbf{0.8699 \pm 0.0001}$ |
| Higgs Small (ROC-AUC ↑) | $\underline{0.7271 \pm 0.0013}$ | $0.7248 \pm 0.0013$ | $\mathbf{0.7280 \pm 0.0009}$ |

Table 6: Performance comparison of `Adam`, `AdamW` and `ALSO` with hyperparameters found for `ALSO`. The trained model is MLP-PLR (Gorishniy et al., 2022). Bold entries represent the best method on each dataset according to mean, underlined entries represent methods, which performance is best with standard deviations over 15 seeds taken into account. Metric is written near dataset name, ↑ means that higher values indicate better performance, ↓ means that lower values indicate better performance.

## D.5 WEIGHTS ANALYSIS

Here we perform analysis of $\pi$ vector behavior. In Figure 11 we can see that weights are changing during training process. For some tasks weights converge, while for other they are still changing. This effect can be explained that we use early stopping or stop training before converges.

More interesting is comparison of values of default loss and weighted. As we can see in Figure 12 weighted loss increases losses on some batches, and decreases on other. It means that intuition behind (4) is probably the same as we propose in Section 1.

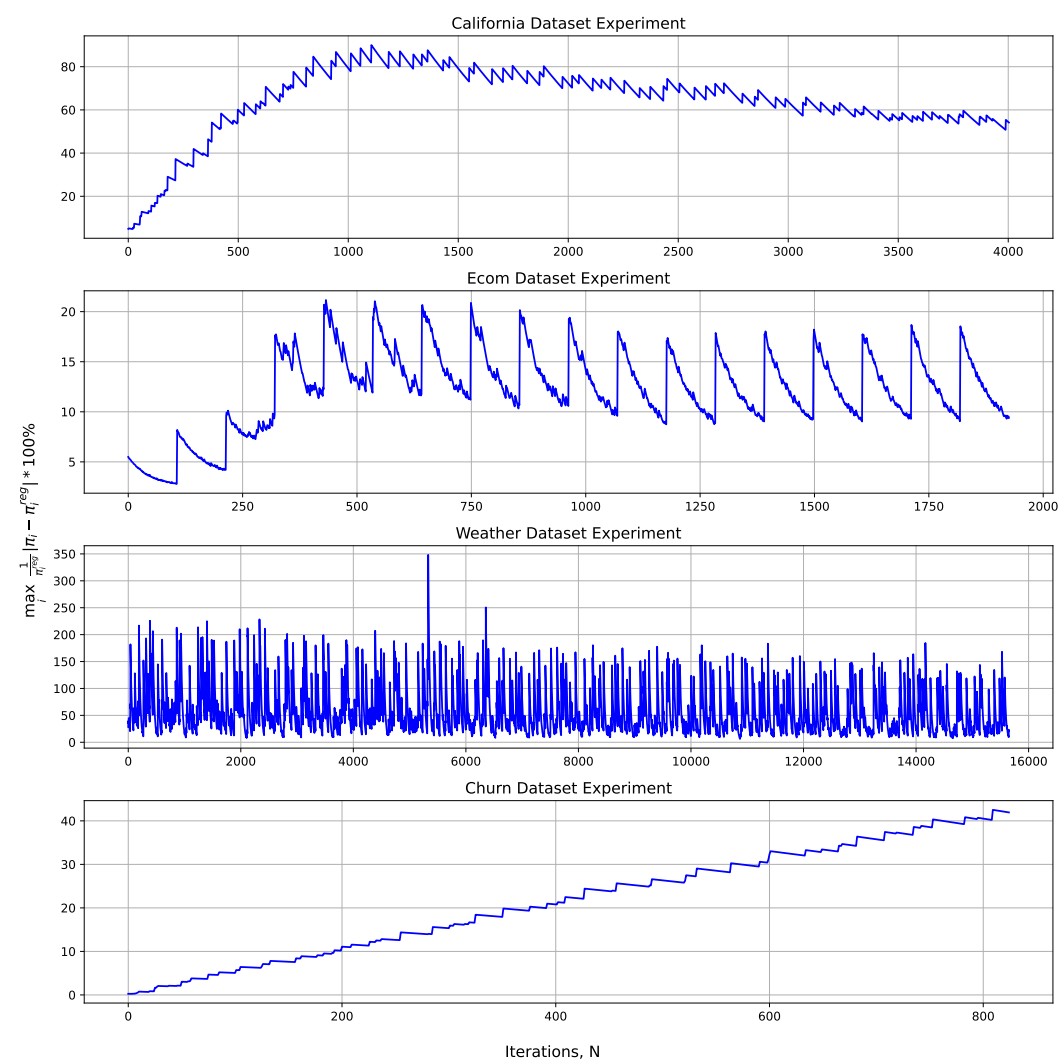

Figure 11: Maximum percentage difference between $\pi$ and $\hat{\pi}$ during training of several our experiments with `ALSO`.

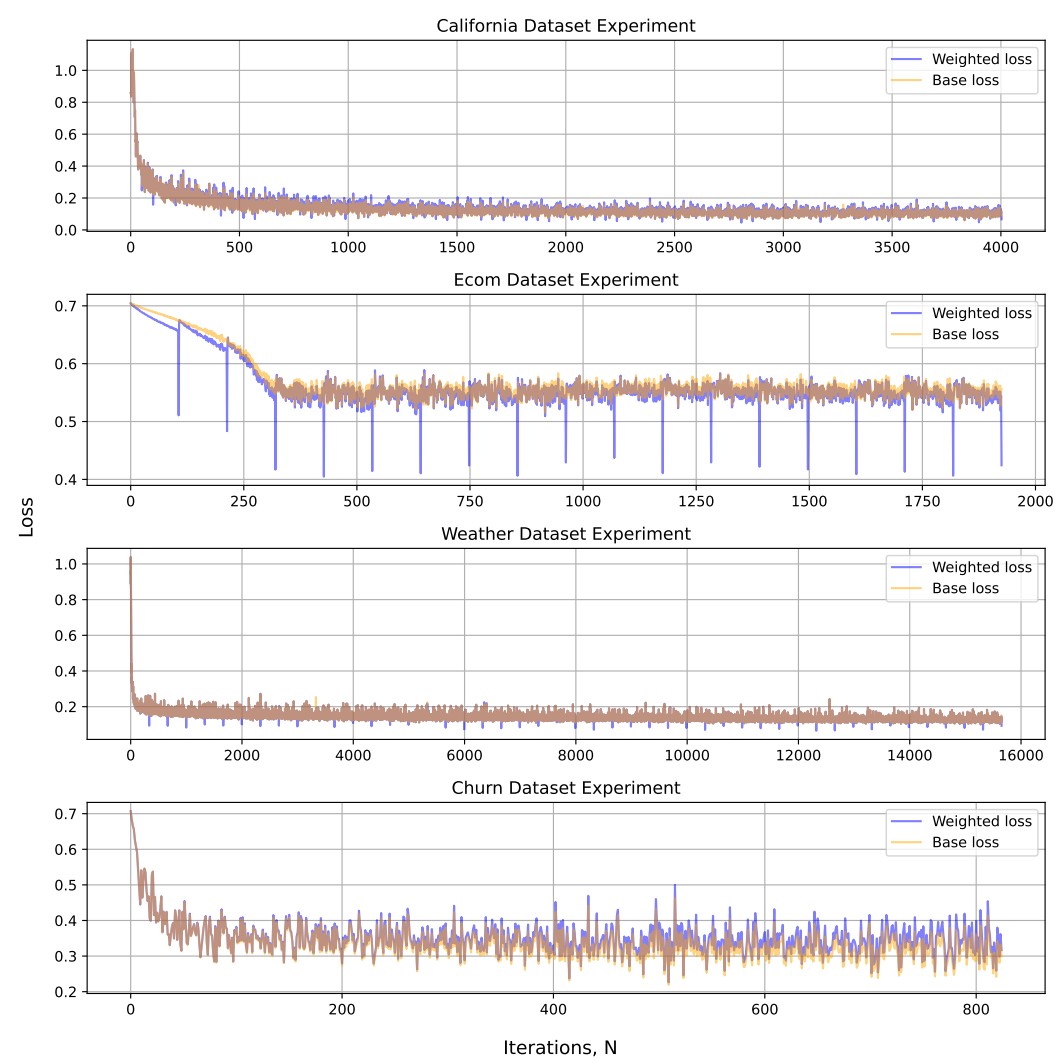

Figure 12: Comparison of weighted loss and non-weighted loss during f several our experiments with `ALSO`. At each iteration we report base loss on batch and weighted loss on batch.

# E  OPTIMISTIC MIRROR-PROX

## E.1  VARIATIONAL INEQUALITIES

It is widely accepted in the modern literature to study the saddle point problem, and, correspondingly, the main problem of the paper (4), within the more general framework of Variational Inequalities (VI) (Stampacchia, 1964; Beznosikov et al., 2020; Mokhtari et al., 2020; Hsieh et al., 2020; Gorbunov et al., 2022) with non-smooth regularization added, since we use the KL divergence in (4). The task is to find $z^* \in \mathcal{Z}$ such that for all $z \in \mathcal{Z}$ it holds that:

$$\langle F(z^*), z - z^* \rangle + \tau V(z, \hat{z}) - \tau V(z^*, \hat{z}) \geq 0, \tag{6}$$

where $\mathcal{Z}$ is some convex vector space and $F : \mathcal{Z} \to \mathbb{R}^d$ is an operator. The function $V(z_1, z_2)$ represents a Bregman divergence, which serves as a non-smooth regularizer (for example, the KL divergence in problem (4)). We now provide the formal definition of $V(z_1, z_2)$. Let $\omega(\cdot)$ be a proper differentiable and 1-strongly convex function with respect to $\|\cdot\|$ on $\mathcal{Z}$. Then for any $z_1, z_2 \in \mathcal{Z}$ we can define the Bregman divergence as

$$V(z_1, z_2) := \omega(z_1) - \omega(z_2) - \langle \nabla\omega(z_2), z_1 - z_2 \rangle. \tag{7}$$

The definition (7) is a generalization of the concept of norm for arbitrary convex sets. For example, the KL divergence from the equation (4) is a special case of the Bregman divergence on the simplex $\Delta_{n-1}$ with $\|\cdot\| = \|\cdot\|_1$ and a generating negative entropy function of the form $\omega_{\mathrm{KL}}(u) = \sum_{j=1}^{n} u_j \log(u_j)$.

In order to proceed from the problem (6) to saddle point, one should set $z = [x, y]^T$ and $F(z) = [\nabla_x g(x, y), -\nabla_y g(x, y)]^T$. It is common to consider methods for solving saddle-point problems together with the solution of VI (6).

**Application of Variational Inequalities.** Although VI were inspired by min-max problem, the formulation (6) has further numerous significant special cases, such as the classical minimization (Nesterov, 2005) and fixed point (Reich, 1983; Taiwo et al., 2021) problems. The setting (6) is applied in classical disciplines such as game theory, economics, equilibrium theory and convex analysis (Stampacchia, 1964; Browder, 1966; Rockafellar, 1969; Sibony, 1970; Luenberger et al., 1984). However, formulation (6) has gained the most popularity with the rise of machine learning and artificial intelligence models. VI problem arises in the GAN optimization (Arjovsky et al., 2017; Goodfellow et al., 2020; Aggarwal et al., 2021), in the reinforcement learning (Omidshafiei et al., 2017; Jin & Sidford, 2020) and adversarial training (Ben-Tal, 2009; Madry et al., 2017). , sparse matrix factorizations (Bach et al., 2008), unsupervised learning (Esser et al., 2010; Chambolle & Pock, 2011), non-smooth optimization (Nesterov, 2005) and discriminative clustering (Joachims, 2005).

Now we connect our problem (4) to (6). For simplicity, let $n = c$, $U = \Delta_{n-1}$, $n_i = 1$, $f_{i,1} := f_i$. Then:

**Proposition E.1.** *The formulation* (4) *is a special case of the VI problem* (6) *with*

$$z := [\theta, \pi]^T, \ \hat{z} := [\boldsymbol{0}, \hat{\pi}]^T, \ \mathcal{Z} = \mathbb{R}^d \times \Delta_{n-1},$$

$$V(z_1, z_2) := \frac{1}{2}\|\theta^1 - \theta^2\|_2^2 + KL\left[\pi^1 \parallel \pi^2\right],$$

$$F(z) := \left[\sum_{i=1}^{n} \pi_i \nabla f_i(\theta), \ -f_1(\theta), ..., -f_n(\theta)\right]^T.$$

We now introduce the common assumptions required for the analysis of solving (6).

**Assumption E.2.** The operator $F$ is $L_F$-Lipschitz continuous on $\mathcal{Z}$, i.e., for any $z_1, z_2 \in \mathcal{Z}$ the following inequality holds

$$\|F(z_1) - F(z_2)\|_* \leq L_F\|z_1 - z_2\|,$$

where $\|\cdot\|$ is the norm with respect to which the generating function $\omega(\cdot)$ of the Bregman divergence $V(\cdot, \cdot)$ form the problem (6) is 1-strongly convex.

**Assumption E.3.** The operator $F$ is monotone on $\mathcal{Z}$, i.e., for all $z_1, z_2 \in \mathcal{Z}$ the following inequality holds

$$\langle F(z_1) - F(z_2), z_1 - z_2 \rangle \geq 0.$$

Assumptions E.2 and E.3 are classical in the analysis of the problem (6) in the deterministic case (Korpelevich, 1976; Gidel et al., 2018; Tseng, 1995; Hsieh et al., 2019; Mokhtari et al., 2020).

### E.2 OPTIMISTIC MIRROR-PROX

This section introduces an optimistic Mirror-Prox algorithm (Popov, 1980) designed to solve problem (6). We derive a convergence rate for this algorithm and then establish its relationship to the problem (4). In this section we will use $f_{i,j}$ and $f_i$ as synonyms, since for simplicity and ease of notations we use $c = n$ during this section. Thus $j$ is always equal to 1.

---

**Algorithm 2** Optimistic Mirror-Prox

1: **Parameters:** stepsize $\gamma$, momentum $\alpha$, number of iterations $N$.
2: **Initialization:** choose $z^{-1} = z^0 \in \mathcal{Z}$.
3: **for** $k = 0, 1, 2, \ldots, N$ **do**
4:      $g^k = (1 + \alpha)F(z^k) - \alpha F(z^{k-1})$
5:      $z^{k+1} = \operatorname*{argmin}_{z \in \mathcal{Z}} \{\langle \gamma g^k, z \rangle + V(z, z^k) + \gamma\tau V(z, \hat{z})\}$
6: **end for**

---

We now provide proof of the convergence rate of Algorithm 2.

**Theorem E.4.** *Let Assumptions E.2 and E.3 be satisfied. Let the problem* (6) *be solved by Algorithm 2. Assume that the stepsize $\gamma$ is chosen such that $0 < \gamma \leq 1/(2L_F)$ and momentum $\alpha$ is chosen such that $\alpha = (1 + \gamma\tau)^{-1}$. Then, for all $k \geq 1$ it holds that*

$$V(z^*, z^k) = \mathcal{O}\left[\left(1 - \frac{\gamma\tau}{2}\right)^k V(z^*, z^0)\right].$$

*where $z^*$ is the solution of the problem* (6)*. In other words, if one takes $\gamma = 1/(2L_F)$, then to achieve $\varepsilon$-accuracy (in terms of $V(z^*, z^N) \leq \varepsilon$) one would need at most*

$$\mathcal{O}\left[\frac{L_F}{\tau} \cdot \log\left(\frac{V(z^*, z^0)}{\varepsilon}\right)\right] \quad \textit{iterations of Algorithm 2.}$$

Full proof of Theorem E.4 is provided in next Section E.3.

Since the main point of interest of this work is the specific problem, we need to adapt Algorithm 2 to the problem (4). Additionally, since we use Assumptions E.2, E.3 for convergence, we need to connect them with the problem (4). For this purpose we present one additional Assumption and two Propositions:

**Assumption E.5.** *For all $(i, j)$ functions $f_{i,j}$ from (4) are convex on $\Theta$, i.e., for any $\theta^1, \theta^2 \in \Theta$ the following inequality holds*

$$\langle \nabla f_{i,j}(\theta^1) - \nabla f_{i,j}(\theta^2), \theta^1 - \theta^2 \rangle \geq 0.$$

**Proposition E.6.** *Let Assumptions 4.2 and E.5 be satisfied. Then the target operator $F(\cdot)$ for the problem (4) from Proposition E.1 fits under Assumptions E.2 and E.3 with*

$$L_F^2 = \mathcal{O}\left[\max_{i \in \overline{1,n}}\{L_i^2\} + \max_{i \in \overline{1,n}}\{K_i^2\}\right].$$

**Proposition E.7.** *Consider the problem (4) and the step of Mirror-Prox like algorithm for solving it:*

$$z^{new} = \arg\min_{z \in \mathcal{Z}}\left\{\langle \gamma g, z \rangle + V(z, z^{old}) + \gamma\tau V(z, \hat{z})\right\}$$

*where $\hat{z} = (0, \hat{\pi})$ and $g = (g^\theta, g^\pi)$ is the target function from (4). Then the update rule is:*

$$\theta^{new} = \theta^{old} - \frac{\gamma}{1 + \gamma\tau}(g^\theta + \tau\theta^{old}),$$

$$\pi^{new} = SM\left[\log\pi^{old} - \frac{\gamma}{1 + \gamma\tau}\left(g^\pi + \tau\log\frac{\pi^{old}}{\hat{\pi}}\right)\right],$$

*where SM denotes softmax function.*

Full proof of Propositions E.6 and E.7 is given in Sections E.4 and E.5.

Combining the results from Theorem E.4 and Propositions E.6 and E.7 directly yields the convergence rate of the Optimistic Mirror-Prox for the problem (4).

### E.3 Proof of the convergence rate of Optimistic Mirror-Prox (Theorem E.4)

In the proof of Theorem E.4 we use technical lemma.

**Lemma E.8** (Bregman divergence properties). *For any Bregman divergence $V$ on the set $\mathcal{Z}$, for any $u \in \mathcal{Z}^*$, $z^1, \hat{z} \in \mathcal{Z}$ and $c \in \mathbb{R}$, if we define*

$$z^\dagger := \arg\min_{z \in \mathcal{Z}} \left\{ \langle u, z \rangle + V(z, z^1) + cV(z, \hat{z}) \right\}. \tag{8}$$

*Then, for all $z \in \mathcal{Z}$ it holds that*

$$(1+c)V(z, z^\dagger) \le V(z, z^1) - V(z^\dagger, z^1) - \langle u, z^\dagger - z \rangle + cV(z, \hat{z}) - cV(z^\dagger, \hat{z}).$$

*Proof.* Using optimality condition in the equation (8) we obtain that for all $z \in \mathcal{Z}$ it holds that:

$$\langle u + \nabla\omega(z^\dagger) - \nabla\omega(z^1) + c\nabla\omega(z^\dagger) - c\nabla\omega(\hat{z}), z^\dagger - z \rangle \le 0$$

Using the Law of cosines of the Bregman divergence we can obtain that:

$$\langle \nabla\omega(z_1) - \nabla\omega(z_2), z_1 - z_3 \rangle = V(z_3, z_1) + V(z_1, z_2) - V(z_3, z_2),$$

we can obtain that for all $z \in \mathcal{Z}$ it holds that:

$$\langle u, z^\dagger - z \rangle + V(z, z^\dagger) + V(z^\dagger, z^1) - V(z, z^1) + cV(z, z^\dagger) + cV(z^\dagger, \hat{z}) - cV(z, \hat{z}) \le 0$$

Re-arranging last inequality we obtain for all $z \in \mathcal{Z}$:

$$(1+c)V(z, z^\dagger) \le V(z, z^1) - V(z^\dagger, z^1) - \langle u, z^\dagger - z \rangle + cV(z, \hat{z}) - cV(z^\dagger, \hat{z}).$$

This finishes the proof. $\qquad\square$

*Proof of Theorem E.4.* Using Lemma E.8 with $u = \gamma[(1+\alpha)F(z^k) - \alpha F(z^{k-1})], z^1 = z^k$ and $c = \gamma\tau V(z, \hat{z})$, we can obtain that for all $z \in \mathcal{Z}$ it holds that

$$
\begin{aligned}
(1+\gamma\tau)\,V(z, z^{k+1}) \le\; & V(z, z^k) - V(z^{k+1}, z^k) + \gamma\tau V(z, \hat{z}) - \gamma\tau V(z^{k+1}, \hat{z}) \\
& - \gamma\langle(1+\alpha)F(z^k) - \alpha F(z^{k-1}), z^{k+1} - z \rangle.
\end{aligned} \tag{9}
$$

Consider the dot product in (9). By using straightforward algebra we can obtain that

$$- \gamma\langle(1+\alpha)F(z^k) - \alpha F(z^{k-1}), z^{k+1} - z \rangle = \underbrace{- \gamma\langle F(z^k) - F(z^{k+1}), z^{k+1} - z \rangle}_{①}$$

$$\underbrace{- \gamma\alpha\langle F(z^k) - F(z^{k-1}), z^k - z \rangle}_{②} \underbrace{- \gamma\alpha\langle F(z^k) - F(z^{k-1}), z^{k+1} - z^k \rangle}_{③}$$

$$\underbrace{- \gamma\langle F(z^{k+1}), z^{k+1} - z \rangle}_{④}.$$

Consider ③. Since Assumption E.2 is fulfilled, we can obtain that

$$
\begin{aligned}
-\gamma\alpha\langle F(z^k) - F(z^{k-1}), z^{k+1} - z^k \rangle &\le \gamma^2 L^2\alpha^2\|z^k - z^{k-1}\|^2 + \frac{1}{4}\|z^{k+1} - z^k\|^2 \\
&\le 2\gamma^2 L^2\alpha^2 V(z^k, z^{k-1}) + \frac{1}{2}V(z^{k+1}, z^k).
\end{aligned} \tag{10}
$$

Consider ④ $+ \gamma\tau V(z, \hat{z}) - \gamma\tau V(z^{k+1}, \hat{z})$. By using Assumption E.3 and the definition of the solution $z^* \in \mathcal{Z}$ of the problem (6) we can obtain that

$$
\begin{aligned}
-\gamma\langle F(z^{k+1}), z^{k+1} - z^* \rangle + \gamma\tau V(z, \hat{z}) - \gamma\tau V(z^{k+1}, \hat{z}) =& \\
-\gamma\langle F(z^{k+1}) - F(z^*), z^{k+1} - z^* \rangle& \\
-\gamma\langle F(z^*), z^{k+1} - z^* \rangle + \gamma\tau V(z^*, \hat{z}) - \gamma\tau V(z^{k+1}, \hat{z})& \\
\le -\gamma\left[\langle F(z^*), z^{k+1} - z^* \rangle - \tau V(z^*, \hat{z}) + \tau V(z^{k+1}, \hat{z})\right] \le 0.&
\end{aligned} \tag{11}
$$

Consider ②. For the moment, we simply introduce the notation $a_k := -② = -\gamma\alpha\langle F(z^k) - F(z^{k-1}), z^k - z\rangle$, and deal with it later in this proof. In this case ① is of the form $① = -\alpha^{-1}a_{k+1}$. Using this notation and the results of equations (10) and (11), expression (9) takes the form

$$(1+\gamma\tau)\,V(z^*, z^{k+1}) + \alpha^{-1}a_{k+1} \le V(z^*, z^k) + a_k + 2\gamma^2 L^2\alpha^2 V(z^k, z^{k-1}) - \frac{1}{2}V(z^{k+1}, z^k).$$

For convenience, let us introduce another notation: $\Phi_k := V(z^*, z^k) + a_k$, also set $\alpha = (1+\gamma\tau)^{-1}$, then we obtain result of the form

$$\Phi_{k+1} \le \alpha\Phi_k + \alpha\left[2\gamma^2 L^2\alpha^2 V(z^k, z^{k-1}) - \frac{1}{2}V(z^{k+1}, z^k)\right].$$

We now start to roll-out the recursion from step $k$ to the step $k-m$:

$$\Phi_{k+1} \le \alpha\Phi_k + \alpha\left[2\gamma^2 L^2\alpha^2 V(z^k, z^{k-1}) - \frac{1}{2}V(z^{k+1}, z^k)\right]$$

$$\le \alpha\left\{\alpha\Phi_{k-1} + \alpha\left[2\gamma^2 L^2\alpha^2 V(z^{k-1}, z^{k-2}) - \frac{1}{2}V(z^k, z^{k-1})\right]\right\}$$

$$+ \alpha\left[2\gamma^2 L^2\alpha^2 V(z^k, z^{k-1}) - \frac{1}{2}V(z^{k+1}, z^k)\right]$$

$$\le \alpha^2\left\{\alpha\Phi_{k-2} + \alpha\left[2\gamma^2 L^2\alpha^2 V(z^{k-2}, z^{k-3}) - \frac{1}{2}V(z^{k-1}, z^{k-2})\right]\right\}$$

$$+ \alpha^2\left[2\gamma^2 L^2\alpha^2 V(z^{k-1}, z^{k-2}) - \frac{1}{2}V(z^k, z^{k-1})\right]$$

$$+ \alpha\left[2\gamma^2 L^2\alpha^2 V(z^k, z^{k-1}) - \frac{1}{2}V(z^{k+1}, z^k)\right]$$

$$\dots$$

$$\le \alpha^{m+1}\Phi_{k-m} - \sum_{j=0}^{m-1}\alpha^{j+2}\left(\frac{1}{2} - 2\gamma^2\alpha L^2\right)V(z^{k-j}, z^{k-j-1})$$

$$- \frac{1}{2}\alpha V(z^{k+1}, z^k) + 2\gamma^2 L^2\alpha^{m+3}V(z^{k-m}, z^{k-m-1}). \tag{12}$$

If we consider $\gamma \le 1/(2L)$, then $1/2 - 2\gamma^2\alpha L^2 \ge 1/2 - 2\gamma^2 L^2 \ge 0$ and we can omit the sum in the equation (12). Taking $m = k$ in (12) we obtain:

$$\Phi_{k+1} \le \alpha^{k+1}\Phi_0 - \frac{1}{2}\alpha V(z^{k+1}, z^k) + 2\gamma^2 L^2\alpha^{k+2}V(z^0, z^{-1}).$$

Since we initialize $z^{-1} = z^0$ in the Algorithm 2 we get $V(z^0, z^{-1})$. Now we return all the notations back and get:

$$V(z^*, z^{k+1}) + \frac{1}{2}\alpha V(z^{k+1}, z^k) - \gamma\alpha\langle F(z^{k+1}) - F(z^k), z^{k+1} - z^*\rangle \le \alpha^k V(z^*, z^0). \tag{13}$$

By Using Fenchel-Young inequality we can obtain that:

$$V(z^*, z^{k+1}) + \frac{1}{2}\alpha V(z^{k+1}, z^k) - \gamma\alpha\langle F(z^{k+1}) - F(z^k), z^{k+1} - z^*\rangle \ge V(z^*, z^{k+1})$$

$$- \frac{1}{2}\alpha V(z^*, z^{k+1}) + \frac{1}{2}\alpha V(z^{k+1}, z^k) \tag{14}$$

$$- 2\gamma^2 L^2\alpha V(z^{k+1}, z^k) \ge \frac{1}{2}V(z^*, z^{k+1}).$$

Combining (13) and (14) we can obtain that:

$$V(z^*, z^{k+1}) \le 2\alpha^{k+1}V(z^*, z^0)$$

Subtracting $\alpha = (1+\gamma\tau)^{-1}$ finishes the proof. $\qquad\square$

E.4    PROOF OF PROPOSITION E.6

In the proof of Proposition E.6 we use several technical lemmas.

**Lemma E.9.** *If $V_{\mathcal{X}}$ and $V_{\mathcal{Y}}$ are Bregman divergences on normed vector spaces $(\mathcal{X}, \|\cdot\|_{\mathcal{X}})$ and $(\mathcal{Y}, \|\cdot\|_{\mathcal{Y}})$ respectively, then $V_{\mathcal{Z}}(\cdot) := V_{\mathcal{X}}(\cdot) + V_{\mathcal{Y}}(\cdot)$ is also a Bregman divergence on the normed vector space $(\mathcal{Z} := \mathcal{X} \times \mathcal{Y}, \|\cdot\|_{\mathcal{Z}} := \sqrt{\|\cdot\|_{\mathcal{X}}^2 + \|\cdot\|_{\mathcal{Y}}^2})$ with generating function $\omega_{\mathcal{Z}}(\cdot) = \omega_{\mathcal{X}}(\cdot) + \omega_{\mathcal{Y}}(\cdot)$. Moreover, for conjugate norm $\|\cdot\|_{\mathcal{Z}^*}$ it holds that $\|\cdot\|_{\mathcal{Z}^*}^2 \leq 2\|\cdot\|_{\mathcal{X}^*}^2 + 2\|\cdot\|_{\mathcal{Y}^*}^2$.*

*Proof.* Let us prove the first part of Lemma E.9. Since $\omega_{\mathcal{X}}(\cdot)$ and $\omega_{\mathcal{Y}}(\cdot)$ are 1-strongly convex on $(\mathcal{X}, \|\cdot\|_{\mathcal{X}})$ and $(\mathcal{Y}, \|\cdot\|_{\mathcal{Y}})$ respectively, for all $x_1, x_2 \in \mathcal{X}$ and $y_1, y_2 \in \mathcal{Y}$ it holds that

$$\omega_{\mathcal{X}}(x_2) \geq \omega_{\mathcal{X}}(x_1) + \langle \nabla \omega_{\mathcal{X}}(x_1), x_2 - x_1 \rangle + \frac{1}{2}\|x_1 - x_2\|_{\mathcal{X}}^2, \tag{15}$$

$$\omega_{\mathcal{Y}}(y_2) \geq \omega_{\mathcal{Y}}(y_1) + \langle \nabla \omega_{\mathcal{Y}}(y_1), y_2 - y_1 \rangle + \frac{1}{2}\|y_1 - y_2\|_{\mathcal{Y}}^2. \tag{16}$$

Now consider $\mathcal{Z} := \mathcal{X} \times \mathcal{Y}$, $\omega_{\mathcal{Z}}(z = (x, y)^T) = \omega_{\mathcal{X}}(x) + \omega_{\mathcal{Y}}(y)$, $\|\cdot\|_{\mathcal{Z}}^2 := \|\cdot\|_{\mathcal{X}}^2 + \|\cdot\|_{\mathcal{Y}}^2$ and $z_1 := (x_1, y_1)^T, z_2 := (x_2, y_2)^T \in \mathcal{Z}$. Summing up (15) and (16) we obtain that

$$\omega_{\mathcal{Z}}(z_2) \geq \omega_{\mathcal{Z}}(z_1) + \langle \nabla \omega_{\mathcal{Z}}(z_1), z_2 - z_1 \rangle + \frac{1}{2}\|z_1 - z_2\|_{\mathcal{Z}}^2, \tag{17}$$

since $\nabla_z \omega_{\mathcal{Z}}(z) = (\nabla_x \omega_{\mathcal{X}}(x), \nabla_y \omega_{\mathcal{Y}}(y))^T$. Inequality (17) means that $\omega_{\mathcal{Z}}(\cdot)$ is 1-strongly convex on $(\mathcal{Z}, \|\cdot\|_{\mathcal{Z}})$ by definition.

Function $\omega_{\mathcal{Z}}(\cdot)$ generates Bregman divergence of the form

$$\begin{aligned}
V_{\mathcal{Z}}(z_1, z_2) &= \omega_{\mathcal{Z}}(z_1) - \omega_{\mathcal{Z}}(z_2) - \langle \nabla_z \omega_{\mathcal{Z}}(z_2), z_1 - z_2 \rangle \\
&= \omega_{\mathcal{X}}(x_1) + \omega_{\mathcal{Y}}(y_1) - \omega_{\mathcal{X}}(x_2) - \omega_{\mathcal{Y}}(y_2) - \langle \nabla_x \omega_{\mathcal{X}}(x_2), x_1 - x_2 \rangle \\
&\quad - \langle \nabla_y \omega_{\mathcal{Y}}(y_2), y_1 - y_2 \rangle = V_{\mathcal{X}}(x_1, x_2) + V_{\mathcal{Y}}(y_1, y_2).
\end{aligned}$$

This finishes the first part of the proof. Consider the second statement. By definition of the conjugate norm for all $a := (a_x, a_y) \in \mathcal{Z}^*$ with $a_x \in \mathcal{X}^*$ and $a_y \in \mathcal{Y}^*$ we have

$$\begin{aligned}
\|a\|_{\mathcal{Z}^*} &\overset{\text{def}}{=} \sup_{z \in \mathcal{Z}: \|z\|_{\mathcal{Z}} \leq 1} \{\langle a, z \rangle\} = \sup_{(x,y)^T \in \mathcal{Z}: \|x\|_{\mathcal{X}}^2 + \|y\|_{\mathcal{Y}}^2 \leq 1} \{\langle a_x, x \rangle + \langle a_y, y \rangle\} \\
&\leq \sup_{x \in \mathcal{X}: \|x\|_{\mathcal{X}} \leq 1} \{\langle a_x, x \rangle\} + \sup_{y \in \mathcal{Y}: \|y\|_{\mathcal{Y}} \leq 1} \{\langle a_y, y \rangle\} = \|a_x\|_{\mathcal{X}^*} + \|a_y\|_{\mathcal{Y}^*}.
\end{aligned}$$

This means that $\|a\|_{\mathcal{Z}^*}^2 \leq (\|a_x\|_{\mathcal{X}^*} + \|a_y\|_{\mathcal{Y}^*})^2 \leq 2\|a_x\|_{\mathcal{X}^*}^2 + 2\|a_y\|_{\mathcal{Y}^*}^2$. This finishes the proof. $\square$

**Lemma E.10.** *If a function $g(x, y) : \mathcal{X} \times \mathcal{Y} \to \mathbb{R}$ is convex w.r.t. $x$ and concave w.r.t. $y$, then target operator $F$ for the min-max problem $\min_{x \in \mathcal{X}} \max_{y \in \mathcal{Y}} \{g(x, y)\}$ of the from*

$$F(z) := [\nabla_x g(x, y), \ -\nabla_y g(x, y)]^T,$$

*is monotone.*

*Proof.* Let us write down scalar product from the definition of the monotone operator from Assumption E.3:

$$\begin{aligned}
\langle F(z_1) - F(z_2), z_1 - z_2 \rangle &= \langle \nabla_x g(x_1, y_1) - \nabla_x g(x_2, y_2), x_1 - x_2 \rangle \\
&\quad - \langle \nabla_y g(x_1, y_1) - \nabla_y g(x_2, y_2), y_1 - y_2 \rangle \\
&= \langle \nabla_x g(x_1, y_1), x_1 - x_2 \rangle + \langle -\nabla_y g(x_1, y_1), y_1 - y_2 \rangle \\
&\quad + \langle \nabla_x g(x_2, y_2), x_2 - x_1 \rangle + \langle -\nabla_y g(x_2, y_2), y_2 - y_1 \rangle \\
&\geq g(x_1, y_1) - g(x_2, y_1) + g(x_1, y_2) - g(x_1, y_1) \\
&\quad + g(x_2, y_2) - g(x_1, y_2) + g(x_2, y_1) - g(x_2, y_2) = 0.
\end{aligned}$$

All inequalities hold since $g(x, y)$ is convex and concave w.r.t. $x$ and $y$ respectively. This finishes the proof. $\square$

*Proof of Proposition E.6.* We start from the fact, if $f_i$ from (1) fall under Assumption 4.2, then target operator

$$F(z = (\theta, \pi)^T) := \left[ \sum_{i=1}^n \pi_i \nabla \tilde{f}_i(\theta), \ -\tilde{f}_1(\theta), ..., -\tilde{f}_n(\theta) \right]^T,$$

from the equation (E.1) falls under Assumption E.2. Let us start from the definition of the Lipschitz continuous operators:

$$\|F(z_1) - F(z_2)\|_*^2 \le 2 \underbrace{\left\| \sum_{i=1}^n \pi_i^1 \nabla f_i(\theta^1) - \pi_i^2 \nabla f_i(\theta^2) \right\|_2^2}_{①}$$

$$+ 2 \underbrace{\left\| [f_1(\theta^1) - f_1(\theta^2), ..., f_n(\theta^1) - f_n(\theta^2)]^T \right\|_\infty^2}_{②}.$$

In this inequality we used Lemma E.9 with $(\mathcal{X}, \|\cdot\|_{\mathcal{X}}) = (\Theta, \|\cdot\|_2)$ and $(\mathcal{Y}, \|\cdot\|_{\mathcal{Y}}) = (\Delta_{n-1}, \|\cdot\|_1)$. Let us consider ①.

$$\left\| \sum_{i=1}^n \pi_i^1 \nabla f_i(\theta^1) - \pi_i^2 \nabla f_i(\theta^2) \right\|_2^2 = \left\| \sum_{i=1}^n \pi_i^1 \left[ \nabla f_i(\theta^1) - \nabla f_i(\theta^2) \right] - \sum_{i=1}^n \left[ \pi_i^2 - \pi_i^1 \right] \nabla f_i(\theta^2) \right\|_2^2$$

$$\le 2 \left\| \sum_{i=1}^n \pi_i^1 \left[ \nabla f_i(\theta^1) - \nabla f_i(\theta^2) \right] \right\|_2^2 + 2 \left\| \sum_{i=1}^n \left[ \pi_i^2 - \pi_i^1 \right] \nabla f_i(\theta^2) \right\|_2^2$$

$$\le 2 \sum_{i=1}^n \pi_i^1 \left\| \nabla f_i(\theta^1) - \nabla f_i(\theta^2) \right\|_2^2 + 2 \left( \sum_{i=1}^n \left| \pi_i^2 - \pi_i^1 \right| \cdot \left\| \nabla f_i(\theta^2) \right\|_2 \right)^2$$

$$\le 2 \sum_{i=1}^n \pi_i^1 L_i^2 \|\theta^1 - \theta^2\|_2^2 + 2 \left( \sum_{i=1}^n \left| \pi_i^2 - \pi_i^1 \right| \right)^2 \cdot G^2$$

$$\le 2 \max_{i \in \overline{1,n}} \{ L_i^2 \} \cdot \|\theta^1 - \theta^2\|_2^2 + 2G^2 \cdot \|\pi^1 - \pi^2\|_1^2. \tag{18}$$

Here we use a notation $G := \max_{i \in \overline{1,n}, \ \theta \in \Theta} \{ \|\nabla f_i(\theta)\|_2 \}$. Since $f_i(\cdot)$ are convex according to Assumption E.5, then $G = \max_{i \in \overline{1,n}} \{ K_i \}$.

Consider ②. By definition of $\|\cdot\|_\infty$ norm we can obtain:

$$\left\| [f_1(\theta^1) - f_1(\theta^2), ..., f_n(\theta^1) - f_n(\theta^2)]^T \right\|_\infty^2 = \left( \max_{i \in \overline{1,n}} \left\{ \left| f_i(\theta^1) - f_i(\theta^2) \right| \right\} \right)^2$$

$$= \max_{i \in \overline{1,n}} \left\{ \left| f_i(\theta^1) - f_i(\theta^2) \right|^2 \right\} \tag{19}$$

$$\le \max_{i \in \overline{1,n}} \left\{ K_i^2 \right\} \cdot \|\theta^1 - \theta^2\|_2^2.$$

Combing (18), (19) and the fact that $G = \max_{i \in \overline{1,n}} \{ K_i \}$, we can obtain that

$$\|F(z_1) - F(z_2)\|_*^2 \le \left( 4 \max_{i \in \overline{1,n}} \{ L_i^2 \} + 2 \max_{i \in \overline{1,n}} \left\{ K_i^2 \right\} \right) \cdot \|\theta^1 - \theta^2\|_2^2 + 4 \max_{i \in \overline{1,n}} \{ K_i^2 \} \cdot \|\pi^1 - \pi^2\|_1^2$$

$$\le 4 \left[ \max_{i \in \overline{1,n}} \{ L_i^2 \} + \max_{i \in \overline{1,n}} \left\{ K_i^2 \right\} \right] \cdot \left( \|\theta^1 - \theta^2\|_2^2 + \|\pi^1 - \pi^2\|_1^2 \right)$$

$$= 4 \left[ \max_{i \in \overline{1,n}} \{ L_i^2 \} + \max_{i \in \overline{1,n}} \left\{ K_i^2 \right\} \right] \cdot \|z_1 - z_2\|^2.$$

$$\tag{20}$$

The last equality holds because according to Lemma E.9 $\| \cdot \|_{\mathcal{Z}}^2 = \| \cdot \|_{\mathcal{X}}^2 + \| \cdot \|_{\mathcal{Y}}^2$. From (20) we can obtain that

$$L_F^2 \leq 4 \left[ \max_{i \in \overline{1,n}} \{L_i^2\} + \max_{i \in \overline{1,n}} \{K_i^2\} \right].$$

This finishes the first part of the proof.

Consider the second part of Proposition E.6. In this case $g(\theta, \pi) = \sum_{i=1}^n \pi_i f_i(\theta)$. This function is linear w.r.t. $\pi$, therefore it is concave w.r.t. $\pi$, according to the Assumption E.5 all functions $f_i(\cdot)$ are convex, therefore $g(\pi, \theta)$ is convex w.r.t. $\theta$. Now, using Lemma E.10, we can obtain that target operator for the problem (4) is monotone. This finishes the proof. $\qquad \square$

### E.5 PROOF OF PROPOSITION E.7

*Proof of Proposition E.7.* Consider the step of Mirror-Prox like algorithm:

$$z^{\text{new}} = \arg \min_{z \in \mathcal{Z}} \left\{ \langle \gamma g, z \rangle + V(z, z^{\text{old}}) + \gamma \tau V(z, \hat{z}) \right\} \tag{21}$$

According to structure of the problem (4) and definition of $z$, the problem (21) is equivalent to following problems:

$$\theta^{\text{new}} = \arg \min_{\theta \in \mathbb{R}^d} \left\{ \langle \gamma g^\theta, \theta \rangle + \frac{1}{2} \|\theta - \theta^{\text{old}}\|_2^2 + \frac{\gamma \tau}{2} \|\theta\|_2^2 \right\} \tag{22}$$

$$\pi^{\text{new}} = \arg \min_{\pi \in \Delta^{n-1}} \left\{ \langle \gamma g^\pi, \pi \rangle + \text{KL} \left[ \pi \, \| \, \pi^{\text{old}} \right] + \gamma \tau \text{KL} \left[ \pi \, \| \, \hat{\pi} \right] \right\} \tag{23}$$

We will start with (22). Using first order optimality condition for $\theta^{\text{new}}$ we can obtain that

$$\gamma g^\theta + (\theta^{\text{new}} - \theta^{\text{old}}) + \gamma \tau \theta^{\text{new}} = 0$$

Then

$$\theta^{\text{new}}(1 + \gamma \tau) = \theta^{\text{old}} - \gamma g^\theta$$

$$\theta^{\text{new}} = \frac{1 + \gamma \tau - \gamma \tau}{1 + \gamma \tau} \theta^{\text{old}} - \frac{\gamma}{1 + \gamma \tau} g^\theta$$

$$\theta^{\text{new}} = \theta^{\text{old}} - \frac{\gamma}{1 + \gamma \tau} \left( g^\theta + \tau \theta^{\text{old}} \right)$$

To deal with (23) we reformulate it as classical constrained optimization problem

$$\min_{\pi} \quad \langle \gamma g^\pi, \pi \rangle + \text{KL} \left[ \pi \, \| \, \pi^{\text{old}} \right] + \gamma \tau \text{KL} \left[ \pi \, \| \, \hat{\pi} \right] \quad s.t. \quad \sum_{i=1}^n \pi_i = 1, \ \pi_i \geq 0 \ \forall i = \overline{1...n} \tag{24}$$

We use Karush–Kuhn–Tucker conditions (Kuhn & Tucker, 1951) to solve problem (24). Let us write out a Lagrange function $L(\pi, \beta_1, \ldots, \beta_n, \lambda)$ for problem (24):

$$L(\pi, \beta_1, \ldots, \beta_n, \lambda) := \sum_{i=1}^n \left[ \gamma \pi_i g_i^\pi - \pi_i \log(\pi_i / \pi_i^{\text{old}}) - \gamma \tau \pi_i \log(\pi_i / \hat{\pi}_i) \right] - \sum_{i=1}^n \beta_i \pi_i + \lambda \sum_{i=1}^n \pi_i - \lambda,$$

where KKT multipliers $\beta_i \geq 0$ correspond to the inequalities $-\pi_i \leq 0$ and $\lambda \in \mathbb{R}$ stands for equality $\sum_{i=1}^n \pi_i - 1 = 0$.

Let us write out partial derivative $\partial L / \partial \pi_i$:

$$\frac{\partial L}{\partial \pi_i} = \gamma g_i^\pi + \log(\pi_i / \pi_i^{\text{old}}) + 1 + \gamma \tau \log(\pi_i / \hat{\pi}_i) + \gamma \tau - \beta_i + \lambda. \tag{25}$$

Since $L$ is convex with respect to $\pi$, we can set $\partial L/\partial \pi_i$ to zero. From (25) we can obtain that

$$\pi_i^* = \left(\pi_i^{\text{old}}(\hat{\pi}_i)^{\gamma\tau} \exp\left[-\gamma g_i^\pi\right] \cdot \exp\left[-\lambda - \gamma\tau - 1 + \beta_i\right]\right)^{\frac{1}{1+\gamma\tau}}$$

$$= \left(\pi_i^{\text{old}}(\hat{\pi}_i)^{\gamma\tau}\right)^{\frac{1}{1+\gamma\tau}} \exp\left[-\frac{1 + \gamma\tau + \gamma g_i^\pi}{1 + \gamma\tau}\right] \cdot \exp\left[\frac{\beta_i - \lambda}{1 + \gamma\tau}\right].$$

Now one can write dual problem and find out that $\lambda_i^* = 0$. Since $\sum_{i=1}^n \pi_i^* = 1$:

$$\exp\left[\frac{-\lambda}{1 + \gamma\tau}\right] \sum_{i=1}^n \left(\left(\pi_i^{\text{old}}(\hat{\pi}_i)^{\gamma\tau}\right)^{\frac{1}{1+\gamma\tau}} \exp\left[-\frac{1 + \gamma\tau + \gamma g_i^\pi}{1 + \gamma\tau}\right]\right) = 1$$

$$\Rightarrow \exp\left[\frac{-\lambda}{1 + \gamma\tau}\right] = \frac{1}{\sum_{i=1}^n \left(\left(\pi_i^{\text{old}}(\hat{\pi}_i)^{\gamma\tau}\right)^{\frac{1}{1+\gamma\tau}} \exp\left[-\frac{1+\gamma\tau+\gamma g_i^\pi}{1+\gamma\tau}\right]\right)}$$

then all conditions of KKT will be fulfilled and optimal $\pi^* = \pi^{\text{new}}$ takes form:

$$\pi_i^{\text{new}} = \frac{\left(\pi_i^{\text{old}}(\hat{\pi}_i)^{\gamma\tau}\right)^{\frac{1}{1+\gamma\tau}} \exp\left[-\frac{1+\gamma\tau+\gamma g_i^\pi}{1+\gamma\tau}\right]}{\sum_{j=1}^n \left(\left(\pi_j^{\text{old}}(\hat{\pi}_j)^{\gamma\tau}\right)^{\frac{1}{1+\gamma\tau}} \exp\left[-\frac{1+\gamma\tau+\gamma g_j^\pi}{1+\gamma\tau}\right]\right)}$$

$$= \frac{\left(\pi_i^{\text{old}}(\hat{\pi}_i)^{\gamma\tau}\right)^{\frac{1}{1+\gamma\tau}} \exp\left[-\frac{\gamma g_i^\pi}{1+\gamma\tau}\right]}{\sum_{j=1}^n \left(\left(\pi_j^{\text{old}}(\hat{\pi}_j)^{\gamma\tau}\right)^{\frac{1}{1+\gamma\tau}} \exp\left[-\frac{\gamma g_j^\pi}{1+\gamma\tau}\right]\right)}$$

Taking logarithm from both sides:

$$\log \pi_i^{\text{new}} = \frac{1}{1 + \gamma\tau} \log \pi_i^{\text{old}} + \frac{\gamma\tau}{1 + \gamma\tau} \log \hat{\pi}_i - \frac{\gamma g_i^\pi}{1 + \gamma\tau}$$

$$+ \log \sum_{j=1}^n \left(\left(\pi_j^{\text{old}}(\hat{\pi}_j)^{\gamma\tau}\right)^{\frac{1}{1+\gamma\tau}} \exp\left[-\frac{\gamma g_j^\pi}{1 + \gamma\tau}\right]\right)$$

$$= \log \pi_i^{\text{old}} - \frac{\gamma\tau}{1 + \gamma\tau} \log \pi_i^{\text{old}} + \frac{\gamma\tau}{1 + \gamma\tau} \log \hat{\pi}_i - \frac{\gamma g_i^\pi}{1 + \gamma\tau}$$

$$+ \log \sum_{j=1}^n \left(\left(\pi_j^{\text{old}}(\hat{\pi}_j)^{\gamma\tau}\right)^{\frac{1}{1+\gamma\tau}} \exp\left[-\frac{\gamma g_j^\pi}{1 + \gamma\tau}\right]\right)$$

$$= \log \pi_i^{\text{old}} - \frac{\gamma}{1 + \gamma\tau}(g_i^\pi + \tau \log \frac{\pi_i^{\text{old}}}{\hat{\pi}_i}) + \log \sum_{j=1}^n \left(\left(\pi_j^{\text{old}}(\hat{\pi}_j)^{\gamma\tau}\right)^{\frac{1}{1+\gamma\tau}} \exp\left[-\frac{\gamma g_j^\pi}{1 + \gamma\tau}\right]\right)$$

Then from softmax definition we can obtain that:

$$\log \pi_i^{\text{new}} = SM\left(\log \pi_i^{\text{old}} - \frac{\gamma}{1 + \gamma\tau}(g_i^\pi + \tau \log \frac{\pi_i^{\text{old}}}{\hat{\pi}_i})\right)$$

This finishes the proof.

$\square$

## F  Theory for ALSO

### F.1  Definitions

Let $h(\theta, \pi)$ be a differentiable function defined in 4. In our analysis, we will consider Assumptions 4.2, 4.3, and 4.1 to provide theoretical guarantees.

In fact, we apply 4.3 to estimate the norms of stochastic gradients and we add batch size $B$ to control the variance of noise that occurs due to stochastics in gradient oracle. Also in 4.2 we require the $K_{i,j}$-Lipschitz continuity of $f_{i,j}(\theta)$ and their $L_{i,j}$-smoothness. In the sequel, assumption F.2 is useful several times in calculations, but it has a different form, however, we can estimate this constant $L$ through our existing $L_{i,j}$ and $K_{i,j}$.

We use asssumption 4.1 with set $U$ because this notation is adopted in the related paper (Mehta et al., 2024). Namely, we define the domain for $\pi$ as the set $U \cap \Delta$, which is usually used to truncate corners of $\Delta$ to ensure that the KL divergence remains bounded on $\Delta \cap U$. However, in our theory we do not require that the simplex must be with truncated corners.

In this section, we consider a more general case of assumptions for our algorithm. So we now introduce several definitions and lemmas proven in Bylinkin et al. (2025), which will be used in the convergence analysis.

We consider more general problem than (4):

$$\min_{\theta \in \mathbb{R}^d} \max_{\pi \in S} \left[ \mathcal{L}(\theta, \pi) = \sum_{i=1}^{c} \pi_i \left( \frac{c}{n} \sum_{j=1}^{n_i} f_{i,j}(\theta) \right) + \frac{\tau}{2} \|\theta\|_2^2 - \lambda D_\psi(\pi \| \hat{\pi}) \right], \tag{26}$$

where we replace KL-divergence with general $D_\Psi$-divergence (Bregman divergence).

**Assumption F.1.** The domain $S \subseteq \mathbb{R}^c$ is nonempty, closed, convex, with $\hat{\pi} \in \mathrm{Int}(S)$.

**Assumption F.2.** The function $\mathcal{L}(\theta, \pi)$ is $L$-smooth, i.e. for all $(\theta_1, \pi_1), (\theta_2, \pi_2) \in \mathbb{R}^d \times S$ it satisfies

$$\|\nabla \mathcal{L}(\theta_1, \pi_1) - \nabla \mathcal{L}(\theta_2, \pi_2)\|^2 \leq L^2 \left( \|\theta_1 - \theta_2\|^2 + \|\pi_1 - \pi_2\|^2 \right).$$

**Lemma F.3.** *Under Assumptions 4.2, and F.1, the function $\mathcal{L}(\theta, \pi)$ in (26) is $L$-smooth (i.e. Assumption F.2), i.e. for all $(\theta^1, \pi^1), (\theta^2, \pi^2) \in \mathbb{R}^d \times S$ it holds*

$$\|\nabla \mathcal{L}(\theta^1, \pi^1) - \nabla \mathcal{L}(\theta^2, \pi^2)\|^2 \leq L^2 \left( \|\theta^1 - \theta^2\|^2 + \|\pi^1 - \pi^2\|^2 \right),$$

*where the Lipschitz constant $L$ can be chosen as*

$$L^2 = \left( \frac{c}{n} \max_{i \in [c]} \sum_{j=1}^{n_i} L_{i,j} + \tau + \frac{c}{n} \max_{i \in [c]} \sum_{j=1}^{n_i} K_{i,j} \right)^2 + (\lambda L_\psi)^2,$$

*with $L_{i,j}$ and $K_{i,j}$ being the smoothness and Lipschitz constants of $f_{i,j}$ from Assumption 4.2, and $L_\psi$ the Lipschitz constant of $\nabla_\pi D_\psi(\cdot \| \hat{\pi})$.*

*Proof.* We decompose the gradient into its $\theta$- and $\pi$-parts:

$$\nabla_\theta \mathcal{L}(\theta, \pi) = \sum_{i=1}^{c} \pi_i \left( \frac{c}{n} \sum_{j=1}^{n_i} \nabla f_{i,j}(\theta) \right) + \tau \theta, \quad \nabla_\pi \mathcal{L}(\theta, \pi) = \left( \frac{c}{n} \sum_{j=1}^{n_i} f_{i,j}(\theta) \right)_{i=1}^{c} - \lambda \nabla_\pi D_\psi(\pi \| \hat{\pi}).$$

For the $\theta$-part we obtain

$$\|\nabla_\theta \mathcal{L}(\theta^1, \pi^1) - \nabla_\theta \mathcal{L}(\theta^2, \pi^2)\|$$

$$\leq \sum_{i=1}^{c} |\pi_i^1 - \pi_i^2| \left( \frac{c}{n} \sum_{j=1}^{n_i} \|\nabla f_{i,j}(\theta^1)\| \right) + \frac{c}{n} \sum_{i=1}^{c} \pi_i^2 \sum_{j=1}^{n_i} \|\nabla f_{i,j}(\theta^1) - \nabla f_{i,j}(\theta^2)\| + \tau \|\theta^1 - \theta^2\|$$

$$\leq \frac{c}{n} \max_i \sum_{j=1}^{n_i} K_{i,j} \|\pi^1 - \pi^2\| + \left( \frac{c}{n} \max_i \sum_{j=1}^{n_i} L_{i,j} + \tau \right) \|\theta^1 - \theta^2\|.$$

For the $\pi$-part we analogously have

$$\|\nabla_\pi \mathcal{L}(\theta^1, \pi^1) - \nabla_\pi \mathcal{L}(\theta^2, \pi^2)\| \leq \frac{c}{n} \max_i \sum_{j=1}^{n_i} K_{i,j} \|\theta^1 - \theta^2\| + \lambda L_\psi \|\pi^1 - \pi^2\|.$$

Combining both estimates yields

$$\|\nabla \mathcal{L}(\theta^1, \pi^1) - \nabla \mathcal{L}(\theta^2, \pi^2)\|^2 \ \leq \ \Big(\frac{c}{n} \max_i \sum_j L_{i,j} + \tau + \frac{c}{n} \max_i \sum_j K_{i,j}\Big)^2 \|\theta^1 - \theta^2\|^2 + (\lambda L_\psi)^2 \|\pi^1 - \pi^2\|^2,$$

which completes the proof. $\qquad\square$

**Lemma F.4.** *Under Assumption 4.2, with $\tau = 0$, the function $\mathcal{L}(\theta, \pi)$ in (26) is $K$-lipschitz with respect to $\theta$, i.e. for all $\theta^1, \theta^2 \in \mathbb{R}^d$ and $\pi \in S$ it holds*

$$|\mathcal{L}(\theta^1, \pi) - \mathcal{L}(\theta^2, \pi)| \ \leq \ L\|\theta^1 - \theta^2\|,$$

*where the $K$ can be chosen as*

$$K \ = \ \frac{c}{n} \max_{i \in [c]} \sum_{j=1}^{n_i} K_{ij}$$

*with $K_{i,j}$ being Lipschitz constant of $f_{i,j}$ from Assumption 4.2.*

*Proof.*

$$|\mathcal{L}(\theta^1, \pi) - \mathcal{L}(\theta^2, \pi)| = |\sum_{i=1}^c \pi_i \frac{c}{n} \sum_{j=1}^{n_i} (f_{ij}(\theta^1) - f_{ij}(\theta^2))| \leq$$

$$\sum_{i=1}^c \pi_i \frac{c}{n} \sum_{i=1}^n |f_{ij}(\theta^1) - f_{ij}(\theta^2)| \leq \sum_{i=1}^c \pi_i \frac{c}{n} \sum_{j=1}^{n_i} K_{ij} \|\theta^1 - \theta^2\| \leq$$

$$\leq \frac{c}{n} \|\theta^1 - \theta^2\| \sum_{i=1}^c \pi_i \sum_{j=1}^{n_i} K_{ij} \leq \frac{c}{n} \max_{i \in [c]} \sum_{j=1}^{n_i} K_{ij}$$

The last inequality holds, since $\pi \in \Delta_{c-1}$. $\qquad\square$

**Assumption F.5.** The function $\psi$, which produce $D_\psi$, is **1-strongly convex**, i.e. for all $\pi_1, \pi_2 \in S$ it satisfies

$$\psi(\pi_1) \geqslant \psi(\pi_2) + \langle \nabla\psi(\pi_2), \pi_1 - \pi_2 \rangle + \frac{1}{2}\|\pi_2 - \pi_1\|^2.$$

Lets formulate lemma from (Bylinkin et al., 2025)

**Lemma F.6** ( Bylinkin et al. (2025))**.** *Consider the problem (26) under Assumption F.5. Then, for every $\theta \in \mathbb{R}^d$ the function $\mathcal{L}(\theta, \pi)$ is $\lambda$-**strongly concave**, i.e. for all $\pi_1, \pi_2 \in S$ it satisfies*

$$\mathcal{L}(\theta, \pi_1) \leq \mathcal{L}(\theta, \pi_2) + \langle \nabla_\psi \mathcal{L}(\theta, \pi_2), \pi_1 - \pi_2 \rangle - \frac{\lambda}{2}\left(D_\psi(\pi_1, \pi_2) + D_\psi(\pi_2, \pi_1)\right).$$

### F.2   AUXILIARY LEMMAS

*Notation* 1. For the saddle-point problem (26) and Algorithm 1, we use the following notation, aligned with Bylinkin et al. (2025):

$$g_\theta^t \ \equiv \ \frac{c}{B} \sum_{j=1}^B \pi_{c_j^t} \nabla_\theta f_{c_j^t, i_j^t}(\theta^t), \qquad\qquad \text{stochastic gradient w.r.t. } \theta,$$

$$g_\pi^t \ \equiv \ \frac{c}{B} \sum_{j=1}^B e_{c_j^t} f_{c_j^t, i_j^t}(\theta^t) \ - \ \lambda \nabla_\pi D_\psi(\pi^t \| \hat{\pi}), \qquad\qquad \text{stochastic gradient w.r.t. } \pi,$$

$$\gamma_\theta \text{ --- stepsize for } \theta, \qquad\qquad\qquad \gamma_\pi \text{ --- stepsize for } \pi,$$

$$\mathcal{L}(\theta, \pi) \;\equiv\; \sum_{i=1}^{c} \pi_i \Big( \tfrac{c}{n} \sum_{j=1}^{n_i} f_{i,j}(\theta) \Big) + \tfrac{\tau}{2}\|\theta\|_2^2 - \lambda D_\psi(\pi\|\hat\pi), \qquad S \text{ --- feasible set for } \pi.$$

Here $e_i$ denotes the $i$-th standard basis vector in $\mathbb{R}^c$, $\hat\pi$ is the reference distribution in the regularization term, and $\nabla_\pi D_\psi(\pi^t\|\hat\pi)$ denotes the gradient (or subgradient) of the divergence $D_\psi$ with respect to $\pi$.

According to the notation, Algorithm 1 can be formulated in a simpler form:

$$\theta^{t+1} = \theta^t \;-\; \gamma_\theta\, d_\theta^t,$$

$$\pi^{t+1} = \operatorname*{arg\,min}_{\pi \in S} \Big\{ \, \langle -\gamma_\pi g_\pi^t, \; \pi \rangle \;+\; D_\psi(\pi \,\|\, \pi^t) \Big\},$$

where $d_\theta^t$ is classsical Adam step.

We begin by noting that our convergence analysis is based on the Adam estimator. Let us introduce the main Adam Estimator process:

$$\theta^{t+1} = \theta^t - \gamma_\theta d_\theta^t \;=\; \theta^t - \gamma_\theta \frac{m_\theta^t}{b_t}, \tag{27}$$

$$\pi^{t+1} = \operatorname*{arg\,min}_{\pi \in S} \Big\{ \, \langle -\gamma_\pi g_\pi^t, \; \pi \rangle + D_\psi(\pi \,\|\, \pi^t) \Big\}. \tag{28}$$

We also introduce a copy of the main process, which behaves identically to the original algorithm but is used to generate the scaling constant $b_t$ for the main process:

$$\theta_{\text{copy}}^{t+1} = \theta_{\text{copy}}^t - \gamma_\theta \frac{m_{\theta,\text{copy}}^t}{b_t},$$

$$\pi_{\text{copy}}^{t+1} = \operatorname*{arg\,min}_{\pi \in S} \Big\{ \, \langle -\gamma_\pi \tilde g_\pi^t, \; \pi \rangle + D_\psi(\pi \,\|\, \pi_{\text{copy}}^t) \Big\}.$$

The update rules for the copy and main processes are:

$$m_{\theta,\text{copy}}^t = \beta_1 m_{\theta,\text{copy}}^{t-1} + (1-\beta_1)\tilde g_\theta^t,$$

$$b_t^2 = \beta_2 b_{t-1}^2 + (1-\beta_2)\|\tilde g_\theta^t\|^2,$$

$$m_\theta^t = \beta_1 m_\theta^{t-1} + (1-\beta_1)g_\theta^t,$$

where $g_\theta^t$ is the stochastic gradient with respect to $\theta$ at the point $(\theta^t, \pi^t)$, and $\tilde g_\theta^t$ is the stochastic gradient at the point $(\theta_{\text{copy}}^t, \pi_{\text{copy}}^t)$.

The first moment $m_\theta^t$ admits a closed-form expression:

$$m_\theta^t \;=\; (1-\beta_1)\sum_{k=0}^{t} \beta_1^{t-k}\, g_\theta^k.$$

We initialize

$$m_{\theta,\text{copy}}^{-1} = m_\theta^{-1} = 0, \qquad b_{-1}, b_0 > 0.$$

The purpose of introducing the copy process is to decouple the randomness of the estimator: in the original process, products of random variables inside expectations are dependent, while in the proposed estimator the corresponding quantities can be treated as independent, which allows us to move products under the expectation in the convergence analysis.

According to the above, the next lemma holds.

**Lemma F.7** ((Chezhegov et al., 2024), Lemma 13). *For a reference step $r \leq t$, and letting $\beta_2 = 1 - \frac{1}{K}$ for some $K \geq t - r$, the following lower bound holds:*

$$b_t^2 \geq \beta_2^{t-r} b_r^2 = \left(1 - \frac{1}{K}\right)^{t-r} b_r^2 \geq \left(1 - \frac{1}{K}\right)^K b_r^2 \geq c_m^2 b_r^2,$$

*where for our Adam-type estimator, we can choose $c_m = \frac{1}{2}$.*

Now let us formulate a technical lemma, which we will need in the future to evaluate the resulting sums:

**Lemma F.8.** *Let $a_t = -\langle \nabla\Phi(\theta^t), d_\theta^t \rangle$ and $\xi_t = -\langle \nabla\Phi(\theta^t), g_\theta^t \rangle$, where $d_\theta^t$ is the Adam estimator step and $g_\theta^t$ is the stochastic gradient used for the momentum term in the Adam estimator 27, and $\theta^t$ is the iterate of the main process at step $t$. Then, the following inequality holds:*

$$\sum_{t=0}^{T} a_t \leq \sum_{k=0}^{T} C_k \xi_k + 3\gamma_\theta \kappa L \sum_{k=0}^{T-1} A_k \|d_\theta^k\|^2,$$

*where*

$$C_k = (1 - \beta_1) \sum_{t=k}^{T} \frac{\beta_1^{t-k}}{b_t}, \qquad A_k = b_k \sum_{t=k+1}^{T} \frac{\beta_1^{t-k}}{b_t}.$$

*Proof.* According to the update rule, we have

$$a_t = \frac{1}{b_t} \left( (1 - \beta_1)\xi_t - \langle \nabla\Phi(\theta^t), \beta_1 m_\theta^{t-1} \rangle \right).$$

Hence, we get

$$a_t = \frac{1}{b_t} \left( (1 - \beta_1)\xi_t + \langle \nabla\Phi(\theta^{t-1}) - \nabla\Phi(\theta^t) - \nabla\Phi(\theta^{t-1}), \beta_1 m_\theta^{t-1} \rangle \right)$$

$$= \frac{1}{b_t} \left( (1 - \beta_1)\xi_t + \beta_1 b_{t-1} a_{t-1} + \langle \nabla\Phi(\theta^{t-1}) - \nabla\Phi(\theta^t), \beta_1 m_\theta^{t-1} \rangle \right).$$

Using $3\kappa L$-Lipschitzness of $\Phi$, the last term can be decomposed as follows:

$$\langle \nabla\Phi(\theta^{t-1}) - \nabla\Phi(\theta^t), \beta_1 m_\theta^{t-1} \rangle \leq 3\beta_1 \kappa L \|\theta^t - \theta^{t-1}\| \|m_\theta^{t-1}\|$$

$$\leq 3\gamma_\theta \kappa L \beta_1 b_{t-1} \|d_\theta^{t-1}\|^2,$$

where in the second inequality we apply the property of the proximal operator. Thus, one can obtain

$$a_t \leq \frac{1}{b_t}(1 - \beta_1)\xi_t + \beta_1 \frac{b_{t-1}}{b_t} a_{t-1} + 3\gamma_\theta \kappa L \beta_1 \frac{b_{t-1}}{b_t} \|d_\theta^{t-1}\|^2.$$

Running the recursion over $a_t$, we have

$$a_t \leq \frac{1}{b_t} \sum_{k=0}^{t} (1 - \beta_1) \beta_1^{t-k} \xi_k + 3\gamma_\theta \kappa L \sum_{k=0}^{t-1} \beta_1^{t-k} \frac{b_k}{b_t} \|d_\theta^k\|^2.$$

Summing over $t = 0$ to $T$, we get:

$$\sum_{t=0}^{T} a_t \leq \sum_{t=0}^{T} \frac{1}{b_t} \sum_{k=0}^{t} (1 - \beta_1)\beta_1^{t-k} \xi_k + 3\gamma_\theta \kappa L \sum_{t=0}^{T} \sum_{k=0}^{t-1} \frac{\beta_1^{t-k} b_k}{b_t} \|d_\theta^k\|^2.$$

Switching the order of sums in the second term leads to

$$\sum_{t=0}^{T} a_t = \sum_{t=0}^{T} \frac{1}{b_t} \sum_{k=0}^{t} (1 - \beta_1)\beta_1^{t-k} \xi_k + 3\gamma_\theta \kappa L \sum_{k=0}^{T-1} b_k \|d_\theta^k\|^2 \sum_{t=k+1}^{T} \frac{\beta_1^{t-k}}{b_t}.$$

Thus, the overall summed inequality becomes:

$$\sum_{t=0}^{T} a_t \leq \sum_{k=0}^{T} C_k \xi_k + 3\gamma_\theta \kappa L \sum_{k=0}^{T-1} A_k \|d_\theta^k\|^2,$$

where:

$$C_k = (1 - \beta_1) \sum_{t=k}^{T} \frac{\beta_1^{t-k}}{b_t}, \qquad A_k = b_k \sum_{t=k+1}^{T} \frac{\beta_1^{t-k}}{b_t}.$$

This finishes the proof. $\qquad\qquad\qquad\qquad\qquad\qquad\qquad\qquad\qquad\qquad\qquad\qquad\square$

The next lemma, that is useful for us, help us to upper bound distance between momentum and stochastic gradient:

**Lemma F.9.** *Let $g_t$ is stochastic gradient, and $m_t$ is momentum of the Adam estimator 27 then distance between them such as folowing:*

$$\|g_t - m_t\|^2 \leq \beta_1^2 \cdot G_t, \tag{29}$$

*where $\beta_1$ is parameter in Adam and $G_t = 2\left(\|g_t\|^2 + (1 - \beta_1)\sum_{k=0}^{t-1} \beta_1^{t-k}\|g_k\|^2\right)$.*

*Proof.*

$$\|g_t - m_t\|^2 = \|g_t - (1 - \beta_1)g_t - \beta_1 m_{t-1}\|^2 = \beta_1^2\|g_t - m_{t-1}\|^2$$
$$\leq 2\beta_1^2\left(\|g_t\|^2 + \|m_{t-1}\|^2\right)$$

We know that recursion on momentum $m_t$ is revealed in the following:

$$m_{t-1} = (1 - \beta_1)g_{t-1} + m_{t-2} = (1 - \beta_1)\sum_{k=0}^{t-1} \beta_1^{t-k} g_k$$

Using convexity of $\|\cdot\|^2$ we have:

$$\|m_{t-1}\|^2 = \|(1 - \beta_1)\sum_{k=0}^{t-1} \beta_1^{t-k} g_k\|^2 \leq (1 - \beta_1)^2 \frac{1}{1 - \beta_1^t} \sum_{k=0}^{t-1} \beta_1^{t-k} \|g_k\|^2$$
$$\leq (1 - \beta_1)\sum_{k=0}^{t-1} \beta_1^{t-k}\|g_k\|^2$$

$\qquad\qquad\qquad\qquad\qquad\qquad\qquad\qquad\qquad\qquad\qquad\qquad\qquad\qquad\qquad\qquad\qquad\square$

Now we can move on to the main theorem.

### F.3  MAIN LEMMAS AND THEOREM

#### F.3.1  MAIN LEMMA

**Lemma F.10** (Stochastic distance recursion)**.** *Consider the problem* (26) *under Assumptions F.2, F.5, and 4.3. Let $g_t = \nabla_\pi \mathcal{L}(\theta^t, \pi^t; \zeta_t)$ be the stochastic gradient computed using a mini-batch of size $B$, and let $\xi_t := g_t - \nabla_\pi \mathcal{L}(\theta^t, \pi^t)$ be the noise term. Then, Algorithm 27 with tuning*

$$\gamma_\pi = \frac{\lambda}{8L^2}, \qquad \gamma_\theta \leq \frac{c_m b_0}{1048\,L\,\kappa^4},$$

*produces a sequence $\{(\theta^t, \pi^t)\}_{t=1}^{T}$ such that*

$$\mathbb{E}[D_\psi(\pi^*(\theta^{t+1}), \pi^{t+1})] \leq \left(1 - \tfrac{1}{128\kappa^2}\right)\mathbb{E}[D_\psi(\pi^*(\theta^t), \pi^t)]$$
$$+ \gamma_\theta^2\, C_\Phi\, \mathbb{E}\|\nabla\Phi(\theta^t)\|^2 + \gamma_\theta^2\, C_B\, \tfrac{\sigma^2}{B} + \gamma_\theta^2\, \beta_1^2 C_\beta,$$

*where the constants are*

$$C_\Phi = \frac{2080\,\kappa^6}{c_m^2 b_0^2}, \qquad C_B = \frac{1040\,\kappa^6}{c_m^2 b_0^2} + \frac{\lambda^2}{32L^4}, \qquad C_\beta = \frac{8320\,\kappa^6}{c_m^2 b_0^2}\left(K^2 + \frac{\sigma^2}{B}\right).$$

*Proof.* To begin, we use three-point identity:

$$D_\psi(\pi^*(\theta^{t+1}), \pi^{t+1}) = D_\psi(\pi^*(\theta^{t+1}), \pi^*(\theta^t)) + D_\psi(\pi^*(\theta^t), \pi^{t+1})$$
$$+ \langle \nabla\psi(\pi^*(\theta^t)) - \nabla\psi(\pi^{t+1}), \pi^*(\theta^{t+1}) - \pi^*(\theta^t) \rangle. \quad (30)$$

Further, we write the optimality condition for the stochastic mirror-ascent step:

$$\langle -\gamma_\pi g_t + [\nabla\psi(\pi^{t+1}) - \nabla\psi(\pi^t)], \pi^*(\theta^t) - \pi^{t+1} \rangle \geq 0.$$

Applying (30), we obtain

$$-\gamma_\pi \langle g_t, \pi^*(\theta^t) - \pi^{t+1} \rangle + D_\psi(\pi^*(\theta^t), \pi^t) - D_\psi(\pi^*(\theta^t), \pi^{t+1}) - D_\psi(\pi^{t+1}, \pi^t) \geq 0.$$

Substituting $g_t = \nabla_\pi \mathcal{L}(\theta^t, \pi^t) + \xi_t$, we get:

$$-\gamma_\pi \langle \nabla_\pi\mathcal{L}(\theta^t, \pi^t), \pi^*(\theta^t) - \pi^{t+1} \rangle - \gamma_\pi \langle \xi_t, \pi^*(\theta^t) - \pi^{t+1} \rangle + D_\psi(\pi^*(\theta^t), \pi^t) - D_\psi(\pi^*(\theta^t), \pi^{t+1}) - D_\psi(\pi^{t+1}, \pi^t) \geq 0.$$

After re-arranging the terms, we get

$$D_\psi(\pi^*(\theta^t), \pi^{t+1}) \leq D_\psi(\pi^*(\theta^t), \pi^t) - D_\psi(\pi^{t+1}, \pi^t) - \gamma_\pi \langle \nabla_\pi\mathcal{L}(\theta^t, \pi^t), \pi^*(\theta^t) - \pi^{t+1} \rangle - \gamma_\pi \langle \xi_t, \pi^*(\theta^t) - \pi^{t+1} \rangle.$$
$$(31)$$

Since $\pi^*(\theta^t)$ is the exact maximum of $\mathcal{L}(\theta^t, \pi)$ in $\pi$, there is another optimility condition

$$\gamma_\pi \langle \nabla_\pi\mathcal{L}(\theta^t, \pi^*(\theta^t)), \pi^*(\theta^t) - \pi \rangle \geq 0.$$

Substituting $\pi = \pi^{t+1}$ and summing it with (31), we derive

$$D_\psi(\pi^*(\theta^t), \pi^{t+1}) \leq D_\psi(\pi^*(\theta^t), \pi^t) - D_\psi(\pi^{t+1}, \pi^t)$$
$$+ \gamma_\pi \langle \nabla_\pi\mathcal{L}(\theta^t, \pi^*(\theta^t)) - \nabla_\pi\mathcal{L}(\theta^t, \pi^t), \pi^*(\theta^t) - \pi^{t+1} \rangle - \gamma_\pi \langle \xi_t, \pi^*(\theta^t) - \pi^{t+1} \rangle$$
$$\leq D_\psi(\pi^*(\theta^t), \pi^t) - D_\psi(\pi^{t+1}, \pi^t)$$
$$+ \gamma_\pi \langle \nabla_\pi\mathcal{L}(\theta^t, \pi^*(\theta^t)) - \nabla_\pi\mathcal{L}(\theta^t, \pi^t), \pi^*(\theta^t) - \pi^t \rangle$$
$$+ \gamma_\pi \langle \nabla_\pi\mathcal{L}(\theta^t, \pi^*(\theta^t)) - \nabla_\pi\mathcal{L}(\theta^t, \pi^t), \pi^t - \pi^{t+1} \rangle - \gamma_\pi \langle \xi_t, \pi^*(\theta^t) - \pi^t \rangle - \gamma_\pi \langle \xi_t, \pi^t - \pi^{t+1} \rangle.$$

Now, we are going to utilize the strong concavity of $\mathcal{L}(\theta, \pi)$ in $\pi$:

$$\gamma_\pi \langle \nabla_\pi\mathcal{L}(\theta^t, \pi^*(\theta^t)) - \nabla_\pi\mathcal{L}(\theta^t, \pi^t), \pi^*(\theta^t) - \pi^t \rangle \leq \frac{-\gamma_\pi\lambda}{2} D_\psi(\pi^*(\theta^t), \pi^t).$$

Thus, we have

$$D_\psi(\pi^*(\theta^t), \pi^{t+1}) \leq \left(1 - \frac{\gamma_\pi\lambda}{2}\right) D_\psi(\pi^*(\theta^t), \pi^t) - D_\psi(\pi^{t+1}, \pi^t)$$
$$+ \gamma_\pi \langle \nabla_\pi\mathcal{L}(\theta^t, \pi^*(\theta^t)) - \nabla_\pi\mathcal{L}(\theta^t, \pi^t), \pi^t - \pi^{t+1} \rangle - \gamma_\pi \langle \xi_t, \pi^*(\theta^t) - \pi^{t+1} \rangle.$$

Next, we apply Cauchy-Schwartz inequality to the scalar product and obtain

$$D_\psi(\pi^*(\theta^t), \pi^{t+1}) \leq \left(1 - \frac{\gamma_\pi\lambda}{2}\right) D_\psi(\pi^*(\theta^t), \pi^t) - D_\psi(\pi^{t+1}, \pi^t)$$
$$+ \frac{\gamma_\pi\alpha}{2}\|\nabla_\pi\mathcal{L}(\theta^t, \pi^*(\theta^t)) - \nabla_\pi\mathcal{L}(\theta^t, \pi^t)\|^2 + \frac{\gamma_\pi}{2\alpha}\|\pi^t - \pi^{t+1}\|^2$$
$$- \gamma_\pi \langle \xi_t, \pi^*(\theta^t) - \pi^t \rangle - \gamma_\pi \langle \xi_t, \pi^t - \pi^{t+1} \rangle.$$

For the stochastic noise terms, we apply Young's inequality in Bregman geometry:

$$-\gamma_\pi \langle \xi_t, \pi^t - \pi^{t+1} \rangle \leq \gamma_\pi^2\|\xi_t\|_*^2 + \tfrac{1}{2}D_\psi(\pi^{t+1}, \pi^t).$$

Using $L$-smoothness of $\mathcal{L}$ (see Assumption F.2) and $\psi$ is 1-strongly convex (see Assumption F.5), we obtain

$$D_\psi(\pi^*(\theta^t), \pi^{t+1}) \leq \left(1 - \frac{\gamma_\pi\lambda}{2}\right) D_\psi(\pi^*(\theta^t), \pi^t) - D_\psi(\pi^{t+1}, \pi^t) + \frac{1}{2}D_\psi(\pi^{t+1}, \pi^t)$$
$$+ \gamma_\pi\alpha L^2 D_\psi(\pi^*(\theta^t), \pi^t) + \frac{\gamma_\pi}{\alpha}D_\psi(\pi^{t+1}, \pi^t)$$

$$- \gamma_\pi \langle \xi_t, \pi^*(\theta^t) - \pi^t \rangle \ + \ \gamma_\pi^2 \|\xi_t\|_*^2.$$

Choose $\alpha = 2\gamma_\pi$. Substituting this into the previous inequality and reducing terms $D_\psi(\pi^{t+1}, \pi^t)$, we get

$$\begin{aligned} D_\psi(\pi^*(\theta^t), \pi^{t+1}) \leq \ & \left( 1 - \frac{\gamma_\pi \lambda}{2} \right) D_\psi(\pi^*(\theta^t), \pi^t) \\ & + 2\gamma_\pi^2 L^2 D_\psi(\pi^*(\theta^t), \pi^t) \\ & - \gamma_\pi \langle \xi_t, \pi^*(\theta^t) - \pi^t \rangle \ + \ \gamma_\pi^2 \|\xi_t\|_*^2. \end{aligned}$$

Taking conditional expectation $\mathbb{E}[\cdot \mid \mathcal{F}_t]$ and using $\mathbb{E}[\langle \xi_t, \pi^*(\theta^t) - \pi^t \rangle \mid \mathcal{F}_t] = 0$, we obtain

$$\mathbb{E}\big[D_\psi(\pi^*(\theta^t), \pi^{t+1}) \mid \mathcal{F}_t\big] \leq \left( 1 - \frac{\gamma_\pi \lambda}{2} + 2\gamma_\pi^2 L^2 \right) D_\psi(\pi^*(\theta^t), \pi^t) \ + \ \gamma_\pi^2 \frac{\sigma^2}{B}. \tag{32}$$

The stepsize that minimizes the quadratic factor is

$$\gamma_\pi = \frac{\lambda}{8L^2}.$$

Substituting this choice and applying full expectation yields

$$\mathbb{E}\big[D_\psi(\pi^*(\theta^t), \pi^{t+1})\big] \leq \left( 1 - \frac{1}{32\kappa^2} \right) \mathbb{E}\big[D_\psi(\pi^*(\theta^t), \pi^t)\big] + \frac{\lambda^2}{64L^4} \frac{\sigma^2}{B}, \tag{33}$$

where $\kappa = \frac{L}{\lambda}$ is the condition number.

Let us return to (30). Note that

$$\nabla\psi(\pi^*(\theta^t)) - \nabla\psi(\pi^{t+1}) = \frac{1}{\lambda} \left( \nabla_\pi \mathcal{L}(\theta^t, \pi^{t+1}) - \nabla_\pi \mathcal{L}(\theta^t, \pi^*(\theta^t)) \right).$$

Thus, there is

$$\begin{aligned} D_\psi(\pi^*(\theta^{t+1}), \pi^{t+1}) = & D_\psi(\pi^*(\theta^{t+1}), \pi^*(\theta^t)) + D_\psi(\pi^*(\theta^t), \pi^{t+1}) \\ & + \frac{1}{\lambda} \langle \nabla_\pi \mathcal{L}(\theta^t, \pi^{t+1}) - \nabla_\pi \mathcal{L}(\theta^t, \pi^*(\theta^t)), \pi^*(\theta^{t+1}) - \pi^*(\theta^t) \rangle \\ \leq & D_\psi(\pi^*(\theta^{t+1}), \pi^*(\theta^t)) + D_\psi(\pi^*(\theta^t), \pi^{t+1}) \\ & + \frac{\alpha L^2}{\lambda} D_\psi(\pi^*(\theta^t), \pi^{t+1}) + \frac{1}{\lambda\alpha} D_\psi(\pi^*(\theta^{t+1}), \pi^*(\theta^t)). \end{aligned}$$

Let us choose $\alpha = \lambda^3/64L^4$. With such a choice and using fact that $\kappa \geq 1$, we have

$$D_\psi(\pi^*(\theta^{t+1}), \pi^{t+1}) \leq 65\kappa^4 D_\psi(\pi^*(\theta^{t+1}), \pi^*(\theta^t)) + \left( 1 + \frac{1}{64\kappa^2} \right) D_\psi(\pi^*(\theta^t), \pi^{t+1}).$$

To deal with $D_\psi(\pi^*(\theta^t), \pi^{t+1})$, we utilize (33). Using $(1 + \frac{1}{64\kappa^2})(1 - \frac{1}{32\kappa^2}) \leq 1 - \frac{1}{64\kappa^2}$ and $1 + \frac{1}{64\kappa^2} \leq 2$ we obtain

$$\mathbb{E}\big[D_\psi(\pi^*(\theta^{t+1}), \pi^{t+1})\big] \leq 65\kappa^4 \mathbb{E}\big[D_\psi(\pi^*(\theta^{t+1}), \pi^*(\theta^t))\big] + \left( 1 - \frac{1}{64\kappa^2} \right) \mathbb{E}\big[D_\psi(\pi^*(\theta^t), \pi^t)\big] + \frac{\lambda^2}{32L^4} \frac{\sigma^2}{B}. \tag{34}$$

The remaining task is to prove that the descent step does not dramatically change the distance between the optimal values of weights. Let us write down two optimality conditions:

$$\langle \nabla_\pi \mathcal{L}(\theta^t, \pi^*(\theta^t)), \pi - \pi^*(\theta^t) \rangle \leq 0,$$
$$\langle \nabla_\pi \mathcal{L}(\theta^{t+1}, \pi^*(\theta^{t+1})), \pi - \pi^*(\theta^{t+1}) \rangle \leq 0.$$

Let us substitute $\pi = \pi^*(\theta^{t+1})$ into the first inequality and $\pi = \pi^*(\theta^t)$ into the second one. When summing them up, we have

$$\langle \nabla_\pi \mathcal{L}(\theta^t, \pi^*(\theta^t)) - \nabla_\pi \mathcal{L}(\theta^{t+1}, \pi^*(\theta^{t+1})), \pi^*(\theta^{t+1}) - \pi^*(\theta^t) \rangle \leq 0. \tag{35}$$

On the other hand, we can take advantage of the strong concavity of the objective (see Lemma F.6) and write

$$\langle \nabla_\pi \mathcal{L}(\theta^t, \pi^*(\theta^{t+1})) - \nabla_\pi \mathcal{L}(\theta^t, \pi^*(\theta^t)), \pi^*(\theta^{t+1}) - \pi^*(\theta^t) \rangle \tag{36}$$

$$\leq -\frac{\lambda}{2} \left[ D_\psi(\pi^*(\theta^t), \pi^*(\theta^{t+1})) + D_\psi(\pi^*(\theta^{t+1}), \pi^*(\theta^t)) \right]. \tag{37}$$

Combining (35) and (36), we obtain

$$\frac{\lambda^2}{4} \left[ D_\psi(\pi^*(\theta^t), \pi^*(\theta^{t+1})) + D_\psi(\pi^*(\theta^{t+1}), \pi^*(\theta^t)) \right]^2 \leq L^2 \|\pi^*(\theta^{t+1}) - \pi^*(\theta^t)\|^2 \|\theta^{t+1} - \theta^t\|^2.$$

Re-arranging the terms and substituting Adam estimator step, we derive

$$\left[ D_\psi(\pi^*(\theta^t), \pi^*(\theta^{t+1})) + D_\psi(\pi^*(\theta^{t+1}), \pi^*(\theta^t)) \right] \leq 4\kappa^2 \|\theta^{t+1} - \theta^t\|^2 \equiv 4\gamma_\theta^2 \kappa^2 \left\| d_\theta^t \right\|^2.$$

After simplifying, we have

$$D_\psi(\pi^*(\theta^{t+1}), \pi^*(\theta^t)) \leq 4\gamma_\theta^2 \kappa^2 \left\| d_\theta^t \right\|^2.$$

Using lemma F.9 and lemma F.7:

$$\|d_\theta^t\|^2 = \|\frac{m_\theta^t}{b_t}\|^2 \leq \frac{1}{c_m^2 b_0^2} \|m_\theta^t\|^2 \leq \frac{4}{c_m^2 b_0^2} \left( \|g_\theta^t - m_\theta^t\|^2 + \|\nabla_\theta \mathcal{L}(\theta^t, \pi^t)\|^2 + \|\xi_t\|^2 \right) \tag{38}$$

$$\leq \frac{4}{c_m^2 b_0^2} \left( \beta_1^2 \cdot G_t + \|\nabla_\theta \mathcal{L}(\theta^t, \pi^t)\|^2 + \|\xi_t\|^2 \right), \tag{39}$$

where $\xi_t = \nabla_\theta \mathcal{L}(\theta^t, \pi^t) - g_\theta^t$ is the stochastic gradient noise, $G_t = 2 \left( \|g_\theta^t\|^2 + (1 - \beta_1) \sum_{k=0}^{t-1} \beta_1^{t-k} \|g_\theta^k\|^2 \right)$.

Using $L$-smoothness of $\mathcal{L}$ (see Assumption F.2) and $\psi$ is 1-strongly convex (see Assumption F.5), we obtain

$$\|\nabla_\theta \mathcal{L}(\theta^t, \pi^t)\|^2 \leq 2 \left( \|\nabla \Phi(\theta^t)\|^2 + \|\nabla_\theta \mathcal{L}(\theta^t, \pi^t) - \nabla \Phi(\theta^t)\|^2 \right)$$

$$\leq 2\|\nabla \Phi(\theta^t)\|^2 + 4L^2 D_\psi(\pi^*(\theta^t), \pi^t)$$

Applying expectation and using assumption 4.3 we have:

$$\mathbb{E}\|d_\theta^t\|^2 \leq \frac{4}{c_m^2 b_0^2} \left( \beta_1^2 \cdot \mathbb{E}[G_t] + 2\mathbb{E}\|\nabla \Phi(\theta^t)\|^2 + 4L^2 \mathbb{E}\left[ D_\psi(\pi^*(\theta^t), \pi^t) \right] + \frac{\sigma^2}{B} \right). \tag{40}$$

Setting $\tau = 0$ and using $K$-Lipschitzness F.4 of $\mathcal{L}$ and boundess of variance 4.3, we have

$$\|g_\theta^k\|^2 \leq 2K^2 + \frac{2\sigma^2}{B} \quad \Rightarrow \quad \mathbb{E}[G_t] \leq 8K^2 + \frac{8\sigma^2}{B}. \tag{41}$$

After substituting inequality 41 into 40 we obtain

$$\mathbb{E}\|d_\theta^t\|^2 = \frac{4}{c_m^2 b_0^2} \left( \beta_1^2 \cdot 8(K^2 + \frac{\sigma^2}{B}) + 2\mathbb{E}\|\nabla \Phi(\theta^t)\|^2 + 4L^2 \mathbb{E}[D_\psi(\pi^*(\theta^t), \pi^t)] + \frac{\sigma^2}{B} \right). \tag{42}$$

Let us take an expectation and derive

$$\mathbb{E} D_\psi(\pi^*(\theta^{t+1}), \pi^*(\theta^t)) \leq \frac{16\gamma_\theta^2 \kappa^2}{c_m^2 b_0^2} \left( 8\beta_1^2 \left( K^2 + \frac{\sigma^2}{B} \right) + 2\mathbb{E}\|\nabla \Phi(\theta^t)\|^2 + 4L^2 \mathbb{E}[D_\psi(\pi^*(\theta^t), \pi^t)] + \frac{\sigma^2}{B} \right).$$

Substituting this into (34) we have

$$\mathbb{E}[D_\psi(\pi^*(\theta^{t+1}), \pi^{t+1})] \leq \frac{1040 \gamma_\theta^2 \kappa^6}{c_m^2 b_0^2} \left( 8\beta_1^2(K^2 + \frac{\sigma^2}{B}) + 2\mathbb{E}\|\nabla \Phi(\theta^t)\|^2 + 4L^2 \mathbb{E}[D_\psi(\pi^*(\theta^t), \pi^t)] + \frac{\sigma^2}{B} \right)$$

$$+ \left( 1 - \frac{1}{64\kappa^2} \right) \mathbb{E}[D_\psi(\pi^*(\theta^t), \pi^t)] + \frac{\lambda^2}{32L^4} \frac{\sigma^2}{B}.$$

Using $\gamma_\theta \leq \frac{c_m b_0}{1048\, L\, \kappa^4}$ and substituting (42) into (34), we have

$$\mathbb{E}\big[D_\psi(\pi^*(\theta^{t+1}), \pi^{t+1})\big] \leq \Big(1 - \tfrac{1}{128\kappa^2}\Big) \mathbb{E}\big[D_\psi(\pi^*(\theta^t), \pi^t)\big]$$
$$+ \frac{1040\,\gamma_\theta^2\,\kappa^6}{c_m^2\, b_0^2} \Big(8\beta_1^2\big(K^2 + \tfrac{\sigma^2}{B}\big) + 2\,\mathbb{E}\|\nabla\Phi(\theta^t)\|^2 + \tfrac{\sigma^2}{B}\Big)$$
$$+ \tfrac{\lambda^2}{32L^4}\,\tfrac{\sigma^2}{B}.$$

Collecting terms, we obtain

$$\mathbb{E}[D_\psi(\pi^*(\theta^{t+1}), \pi^{t+1})] \leq \Big(1 - \tfrac{1}{128\kappa^2}\Big) \mathbb{E}[D_\psi(\pi^*(\theta^t), \pi^t)]$$
$$+ \gamma_\theta^2\, C_\Phi\, \mathbb{E}\|\nabla\Phi(\theta^t)\|^2 + \gamma_\theta^2\, C_B\, \tfrac{\sigma^2}{B} + \gamma_\theta^2\, \beta_1^2 C_\beta,$$

where the constants are

$$C_\Phi = \frac{2080\,\kappa^6}{c_m^2 b_0^2}, \qquad C_B = \frac{1040\,\kappa^6}{c_m^2 b_0^2} + \frac{\lambda^2}{32L^4}, \qquad C_\beta = \frac{8320\,\kappa^6}{c_m^2 b_0^2}\Big(K^2 + \tfrac{\sigma^2}{B}\Big).$$

This completes the proof of the stochastic version of the main lemma. $\qquad\square$

### F.3.2  Main theorem

Now let us proceed to the convergence proof for Algorithm 1.

*Proof.* 4.5 One can note that $\Phi$ is $3\kappa L$-smooth. Indeed,

$$\|\nabla\Phi(\theta_1) - \nabla\Phi(\theta_2)\|^2 = \|\nabla_\theta \mathcal{L}(\theta_1, \pi^*(\theta_1)) - \nabla_\theta \mathcal{L}(\theta_2, \pi^*(\theta_2))\|^2$$
$$\leq L^2 \big[\|\theta_1 - \theta_2\|^2 + 2D_\psi(\pi^*(\theta_1), \pi^*(\theta_2))\big] \leq L^2\big(1 + 4\kappa^2\big)\|\theta_1 - \theta_2\|^2$$
$$\leq 9\kappa^2 L^2 \|\theta_1 - \theta_2\|^2.$$

Thus, we can write

$$\Phi(\theta^{t+1}) \leq \Phi(\theta^t) + \langle\nabla\Phi(\theta^t), \theta^{t+1} - \theta^t\rangle + 3\kappa L\|\theta^{t+1} - \theta^t\|^2$$
$$= \Phi(\theta^t) - \gamma_\theta \langle\nabla\Phi(\theta^t), d_\theta^t\rangle + 3\gamma_\theta^2 \kappa L \big\|d_\theta^t\big\|^2$$

Summing from $t = 0$ to $T$ yields

$$\Phi(\theta^{T+1}) \leq \Phi(\theta^0) - \gamma_\theta \sum_{t=0}^{T} \langle\nabla\Phi(\theta^t), d_\theta^t\rangle + 3\gamma_\theta^2 \kappa L \sum_{t=0}^{T} \|d_\theta^t\|^2.$$

Applying lemma F.8 with $a_t = -\langle\nabla\Phi(\theta^t), d_\theta^t\rangle$ we have:

$$\Phi(\theta^{T+1}) \leq \Phi(\theta^0) + \gamma_\theta \sum_{k=0}^{T} C_k \xi_k + 3\gamma_\theta^2 \kappa L \sum_{k=0}^{T} (1 + A_k)\|d_\theta^k\|^2,$$

where $\xi_k = -\langle\nabla\Phi(\theta^k), g_\theta^k\rangle$ and $g_\theta^k$ is the stochastic gradient in the Adam estimator 27.

By decomposing the stochastic gradient into the true gradient and the noise $g_\theta^k = \nabla_\theta \mathcal{L}(\theta^k, \pi^k) + \eta_k$, we have

$$\Phi(\theta^{T+1}) \leq \Phi(\theta^0) - \gamma_\theta \sum_{k=0}^{T} C_k \big\langle\nabla\Phi(\theta^k), \nabla_\theta \mathcal{L}(\theta^k, \pi^k)\big\rangle$$
$$- \gamma_\theta \sum_{k=0}^{T} C_k \big\langle\nabla\Phi(\theta^k), \eta_k\big\rangle + 3\gamma_\theta^2 \kappa L \sum_{k=0}^{T} (1 + A_k)\|d_\theta^k\|^2.$$

Rearranging the terms and dividing by $\gamma_\theta$ yields

$$\sum_{k=0}^{T} C_k \left\langle \nabla\Phi(\theta^k), \nabla_\theta\mathcal{L}(\theta^k, \pi^k) \right\rangle \le \frac{\Phi(\theta^0) - \Phi(\theta^{T+1})}{\gamma_\theta} - \sum_{k=0}^{T} C_k \left\langle \nabla\Phi(\theta^k), \eta_k \right\rangle + 3\gamma_\theta\kappa L \sum_{k=0}^{T-1}(1+A_k)\|d_\theta^k\|^2.$$

(43)

Applying Young's inequality to the scalar product:

$$\left\langle \nabla\Phi(\theta^k), \nabla_\theta\mathcal{L}(\theta^k, \pi^k) \right\rangle \ge \frac{1}{2}\left\|\nabla\Phi(\theta^k)\right\|^2 - \frac{1}{2}\left\|\nabla_\theta\mathcal{L}(\theta^k, \pi^k) - \nabla\Phi(\theta^k)\right\|^2.$$

$$\frac{1}{2}\sum_{k=0}^{T} C_k\|\nabla\Phi(\theta^k)\|^2 - \frac{1}{2}\sum_{k=0}^{T} C_k\|\nabla_\theta\mathcal{L}(\theta^k, \pi^k) - \nabla\Phi(\theta^k)\|^2 \le \frac{\Phi(\theta^0) - \Phi(\theta^{T+1})}{\gamma_\theta}$$

$$- \sum_{k=0}^{T} C_k\langle\nabla\Phi(\theta^k), \eta_k\rangle + 3\gamma_\theta\kappa L \sum_{k=0}^{T-1}(1+A_k)\|d_\theta^k\|^2. \quad (44)$$

Let $\mathcal{F}_k$ denote the history of the main process up to time $k$, and let the coefficients $C_k = (1-\beta_1)\sum_{j=k}^{T}\beta_1^{j-k}/b_j$ be generated by an auxiliary (copy) sequence $\{b_j\}_{j\ge0}$. Since $C_k$ depends only on future $\{b_j\}_{j\ge k}$ from the copy process, while $r_k := \langle\nabla\Phi(\theta^k), \eta_k\rangle$ is generated by the main process at time $k$, we have the conditional independence of $C_k$ and $r_k$ with respect to $(\mathcal{F}_k, \text{copy})$. Using the unbiasedness $\mathbb{E}[\eta_k \mid \mathcal{F}_k] = 0$, the tower property gives

$$\mathbb{E}[C_k\, r_k] = \mathbb{E}[\,\mathbb{E}[C_k\, r_k \mid \mathcal{F}_k, \text{copy}]] = \mathbb{E}[\,\mathbb{E}[C_k \mid \mathcal{F}_k, \text{copy}]\,\mathbb{E}[r_k \mid \mathcal{F}_k]] = 0.$$

Taking conditional expectation of (43) and then applying the tower property, we obtain

$$\frac{1}{2}\sum_{k=0}^{T}\mathbb{E}\big[C_k\|\nabla\Phi(\theta^k)\|^2\big] - \frac{1}{2}\sum_{k=0}^{T}\mathbb{E}\big[C_k\|\nabla_\theta\mathcal{L}(\theta^k, \pi^k) - \nabla\Phi(\theta^k)\|^2\big] \le \frac{\Phi(\theta^0) - \mathbb{E}\,\Phi(\theta^{T+1})}{\gamma_\theta}$$

$$+ 3\gamma_\theta\kappa L \sum_{k=0}^{T-1}\mathbb{E}\big[(1+A_k)\|d_\theta^k\|^2\big]. \quad (45)$$

To separate the factors on the left, use conditional independence as above:

$$\mathbb{E}\big[C_k\|\nabla\Phi(\theta^k)\|^2 \mid \mathcal{F}_k, \text{copy}\big] = \mathbb{E}[C_k \mid \text{copy}] \cdot \|\nabla\Phi(\theta^k)\|^2.$$

Hence

$$\mathbb{E}\big[C_k\|\nabla\Phi(\theta^k)\|^2\big] = \mathbb{E}\big[\mathbb{E}[C_k \mid \text{copy}]\,\|\nabla\Phi(\theta^k)\|^2\big].$$

Let us get the bound of the scaling parameter $b_t$ in the Adam estimator 27:

$$\mathbb{E}\big[\|g_\theta^t\|^2 \mid \theta_{\text{copy}}^k, \pi_{\text{copy}}^k\big] \le 2\,(K^2 + \frac{\sigma^2}{B}), \quad (46)$$

$$\mathbb{E}\big[b_i \mid \theta_{\text{copy}}^k, \pi_{\text{copy}}^k\big] \le \mathbb{E}\left[\sqrt{\beta_2 b_{i-1}^2 + (1-\beta_2)\|\tilde{g}_\theta^t\|^2} \,\Big|\, \theta_{\text{copy}}^k, \pi_{\text{copy}}^k\right]$$

$$\le \mathbb{E}\big[\max\{b_{i-1}, \|\tilde{g}_\theta^t\|\} \mid \theta_{\text{copy}}^k, \pi_{\text{copy}}^k\big]$$

$$\le \max_i \sqrt{2K^2 + 2\frac{\sigma^2}{B}} = \sqrt{2K^2 + 2\frac{\sigma^2}{B}}. \quad (47)$$

Using F.7 we have

$$\mathbb{E}[C_k \mid \theta_{\text{copy}}^k, \pi_{\text{copy}}^k] = (1-\beta_1)\sum_{j=k}^{T}\frac{\beta_1^{j-k}}{\mathbb{E}[b_j \mid \theta_{\text{copy}}^k, \pi_{\text{copy}}^k]} \ge (1-\beta_1)\min_{j\in\{0,\dots,T\}}\frac{1}{\mathbb{E}[b_j \mid \theta_{\text{copy}}^k, \pi_{\text{copy}}^k]} \ge \frac{1-\beta_1}{\sqrt{2K^2 + 2\frac{\sigma^2}{B}}}$$

and

$$\mathbb{E}[C_k \mid \theta_{\text{copy}}^k, \pi_{\text{copy}}^k] \leq \frac{1}{c_m b_0}.$$

Therefore,

$$\sum_{k=0}^{T} \mathbb{E}\big[C_k \|\nabla\Phi(\theta^k)\|^2\big] \geq \frac{1-\beta_1}{\sqrt{2K^2 + 2\frac{\sigma^2}{B}}} \sum_{k=0}^{T} \mathbb{E}\big[\|\nabla\Phi(\theta^k)\|^2\big] \tag{48}$$

and

$$\sum_{k=0}^{T} \mathbb{E}\Big[C_k \big\|\nabla_\theta \mathcal{L}(\theta^k, \pi^k) - \nabla\Phi(\theta^k)\big\|^2\Big] \leq \frac{1}{c_m b_0} \sum_{k=0}^{T} \mathbb{E}\Big[\big\|\nabla_\theta \mathcal{L}(\theta^k, \pi^k) - \nabla\Phi(\theta^k)\big\|^2\Big]. \tag{49}$$

Combining (45) and (48), (49), we arrive at

$$\frac{1-\beta_1}{2\sqrt{2K^2 + 2\frac{\sigma^2}{B}}} \sum_{k=0}^{T} \frac{1}{2}\mathbb{E}\big[\|\nabla\Phi(\theta^k)\|^2\big] - \frac{1}{c_m b_0} \sum_{k=0}^{T} \frac{1}{2}\mathbb{E}\Big[\big\|\nabla_\theta \mathcal{L}(\theta^k, \pi^k) - \nabla\Phi(\theta^k)\big\|^2\Big] \leq \frac{\Phi(\theta^0) - \mathbb{E}\,\Phi(\theta^{T+1})}{\gamma_\theta}$$

$$+ 3\gamma_\theta \kappa L \sum_{k=0}^{T-1} \mathbb{E}\big[(1+A_k)\|d_\theta^k\|^2\big]. \tag{50}$$

Using 42 we have:

$$\mathbb{E}\|d_\theta^t\|^2 = \frac{4}{c_m^2 b_0^2}\Big(\beta_1^2 \cdot 8(K^2 + \frac{\sigma^2}{B}) + 2\,\mathbb{E}\|\nabla\Phi(\theta^t)\|^2 + 4L^2\,\mathbb{E}[D_\psi(\pi^*(\theta^t), \pi^t)] + \frac{\sigma^2}{B}\Big). \tag{51}$$

By definition of $A_k$:

$$\mathbb{E}A_t \leq \frac{\beta_1}{c_m b_0(1-\beta_1)}\sqrt{2K^2 + 2\frac{\sigma^2}{B}},$$

$$\mathbb{E}\big[(1+A_t)\|d_\theta^t\|^2\big] \leq \left(1 + \frac{\beta_1}{c_m b_0(1-\beta_1)}\sqrt{2K^2 + 2\frac{\sigma^2}{B}}\right)$$

$$\cdot \frac{4}{c_m^2 b_0^2}\Big(\beta_1^2 \cdot 8(K^2 + \frac{\sigma^2}{B}) + 2\,\mathbb{E}\|\nabla\Phi(\theta^t)\|^2 + 4L^2\,\mathbb{E}[D_\psi(\pi^*(\theta^t), \pi^t)] + \frac{\sigma^2}{B}\Big).$$

$$C_A := \frac{\beta_1}{c_m b_0(1-\beta_1)}\sqrt{2K^2 + 2\frac{\sigma^2}{B}}, \qquad C_D := \frac{4}{c_m^2 b_0^2}.$$

Then the auxiliary bounds read

$$\mathbb{E}A_t \leq C_A,$$

$$\mathbb{E}\big[(1+A_t)\|d_\theta^t\|^2\big] \leq (1+C_A)\,C_D\Big(\beta_1^2 \cdot 8\Big(K^2 + \frac{\sigma^2}{B}\Big) + 2\,\mathbb{E}\|\nabla\Phi(\theta^t)\|^2 + 4L^2\,\mathbb{E}[D_\psi(\pi^*(\theta^t), \pi^t)] + \frac{\sigma^2}{B}\Big).$$

Substituting these inequalities into the main relation yields

$$\frac{1-\beta_1}{2\sqrt{2K^2 + 2\frac{\sigma^2}{B}}} \sum_{k=0}^{T} \frac{1}{2}\mathbb{E}\big[\|\nabla\Phi(\theta^k)\|^2\big] - \frac{1}{c_m b_0} \sum_{k=0}^{T} \frac{1}{2}\mathbb{E}\Big[\big\|\nabla_\theta \mathcal{L}(\theta^k, \pi^k) - \nabla\Phi(\theta^k)\big\|^2\Big] \leq \frac{\Phi(\theta^0) - \mathbb{E}\,\Phi(\theta^{T+1})}{\gamma_\theta}$$

$$+ 3\gamma_\theta \kappa L \sum_{k=0}^{T-1} (1+C_A)\,C_D\Big(\beta_1^2 \cdot 8\Big(K^2 + \frac{\sigma^2}{B}\Big) + 2\,\mathbb{E}\|\nabla\Phi(\theta^k)\|^2 + 4L^2\,\mathbb{E}[D_\psi(\pi^*(\theta^k), \pi^k)] + \frac{\sigma^2}{B}\Big).$$

Using smoothness of $\mathcal{L}$ and the definition of $\pi^*(\theta^k)$:

$$\frac{1-\beta_1}{2\sqrt{2K^2+2\frac{\sigma^2}{B}}}\sum_{k=0}^{T}\frac{1}{2}\mathbb{E}\big[\|\nabla\Phi(\theta^k)\|^2\big] - \frac{1}{c_m b_0}\sum_{k=0}^{T}L^2\mathbb{E}\big[D_\psi(\pi^*(\theta^k),\pi^k)\big] \leq \frac{\Phi(\theta^0)-\mathbb{E}\,\Phi(\theta^{T+1})}{\gamma_\theta}$$

$$+ 3\gamma_\theta\kappa L\sum_{k=0}^{T-1}(1+C_A)\,C_D\Big(\beta_1^2\cdot 8\Big(K^2+\frac{\sigma^2}{B}\Big) + 2\,\mathbb{E}\|\nabla\Phi(\theta^k)\|^2 + 4L^2\,\mathbb{E}[D_\psi(\pi^*(\theta^k),\pi^k)] + \frac{\sigma^2}{B}\Big).$$

Using

$$\gamma_\theta \leq \frac{1-\beta_1}{72\,\kappa L(1+C_A)C_D\,\sqrt{2K^2+2\sigma^2/B}},$$

we have

$$\frac{1-\beta_1}{2\sqrt{2K^2+2\frac{\sigma^2}{B}}}\sum_{k=0}^{T}\frac{1}{3}\,\mathbb{E}\big[\|\nabla\Phi(\theta^k)\|^2\big] \leq \left[\frac{7(1-\beta_1)}{2\sqrt{2K^2+2\frac{\sigma^2}{B}}} + \frac{1}{c_m b_0}\right]L^2\sum_{k=0}^{T}\mathbb{E}[D_\psi(\pi^*(\theta^k),\pi^k)]$$

$$+ \frac{\Phi(\theta^0)-\mathbb{E}\,\Phi(\theta^{T+1})}{\gamma_\theta} + 3\gamma_\theta\kappa L\sum_{k=0}^{T-1}(1+C_A)\,C_D\Big(\beta_1^2\cdot 8\Big(K^2+\frac{\sigma^2}{B}\Big) + \frac{\sigma^2}{B}\Big). \tag{52}$$

Simplifying our inequality we obtain:

$$\frac{1}{T+1}\sum_{k=0}^{T}\mathbb{E}\big[\|\nabla\Phi(\theta^k)\|^2\big] \leq M_1\,\frac{1}{T+1}\sum_{k=0}^{T}\mathbb{E}\big[D_\psi(\pi^*(\theta^k),\pi^k)\big] + M_2\,\frac{\Phi(\theta^0)-\mathbb{E}\,\Phi(\theta^{T+1})}{(T+1)\gamma_\theta} + M_3\,\gamma_\theta,$$

where

$$M_1 = \left[21 + \frac{6\sqrt{2K^2+2\sigma^2/B}}{(1-\beta_1)}\,\frac{1}{c_m b_0}\right]L^2,$$

$$M_2 = \frac{6\sqrt{2K^2+2\sigma^2/B}}{(1-\beta_1)},$$

$$M_3 = \frac{18\,\kappa L\sqrt{2K^2+2\sigma^2/B}}{(1-\beta_1)}\,(1+C_A)\,C_D\Big(8\beta_1^2(K^2+\tfrac{\sigma^2}{B}) + \tfrac{\sigma^2}{B}\Big).$$

Let us denote $\delta = 1 - 1/128\kappa^2$. Lemma F.10 transforms into

$$\mathbb{E}[D_\psi(\pi^*(\theta^{t+1}),\pi^{t+1})] \leq \Big(1-\tfrac{1}{128\kappa^2}\Big)\mathbb{E}[D_\psi(\pi^*(\theta^t),\pi^t)]$$

$$+ \gamma_\theta^2\,C_\Phi\,\mathbb{E}\|\nabla\Phi(\theta^t)\|^2 + \gamma_\theta^2\,C_B\,\frac{\sigma^2}{B} + \gamma_\theta^2\,\beta_1^2 C_\beta,$$

where the constants are

$$C_\Phi = \frac{2080\,\kappa^6}{c_m^2 b_0^2}, \qquad C_B = \frac{1040\,\kappa^6}{c_m^2 b_0^2} + \frac{\lambda^2}{32L^4}, \qquad C_\beta = \frac{8320\,\kappa^6}{c_m^2 b_0^2}\Big(K^2+\tfrac{\sigma^2}{B}\Big).$$

Hence, by unrolling the recursion, we obtain

$$\frac{1}{T+1}\sum_{t=0}^{T}\mathbb{E}\,D_\psi(\pi^*(\theta^t),\pi^t) \leq \frac{1}{T+1}\cdot\frac{1}{1-\delta}\,D_\psi(\pi^*(\theta^0),\pi^0)$$

$$+ \frac{1}{1-\delta}\left(\gamma_\theta^2\,C_\Phi\,\frac{1}{T+1}\sum_{t=0}^{T}\mathbb{E}\|\nabla\Phi(\theta^t)\|^2 + \gamma_\theta^2\,C_B\,\frac{\sigma^2}{B} + \gamma_\theta^2\,\beta_1^2 C_\beta\right).$$

Substituting the bound on the divergence into the main inequality, we obtain:

$$\frac{1}{T+1}\sum_{k=0}^{T}\mathbb{E}\big[\|\nabla\Phi(\theta^k)\|^2\big] \leq M_1\left[\frac{1}{T+1}\cdot\frac{1}{1-\delta}\,D_\psi(\pi^*(\theta^0),\pi^0)\right.$$

$$+ \frac{1}{1-\delta}\Big(\gamma_\theta^2 C_\Phi \frac{1}{T+1}\sum_{t=0}^{T}\mathbb{E}\|\nabla\Phi(\theta^t)\|^2 + \gamma_\theta^2 C_B \tfrac{\sigma^2}{B} + \gamma_\theta^2 \beta_1^2 C_\beta\Big)\Bigg]$$

$$+ M_2 \frac{\Phi(\theta^0) - \mathbb{E}\,\Phi(\theta^{T+1})}{(T+1)\gamma_\theta} + M_3\,\gamma_\theta.$$

Using $\gamma_\theta \le \sqrt{\frac{(1-\delta)}{2M_1 C_\Phi}}$ we obtain

$$\frac{1}{T+1}\sum_{k=0}^{T}\mathbb{E}\big[\|\nabla\Phi(\theta^k)\|^2\big] \le 2M_1\Bigg[\frac{1}{T+1}\cdot\frac{1}{1-\delta}\,D_\psi(\pi^*(\theta^0),\pi^0)$$

$$+ \frac{1}{1-\delta}\Big(\gamma_\theta^2 C_B \tfrac{\sigma^2}{B} + \gamma_\theta^2 \beta_1^2 C_\beta\Big)\Bigg]$$

$$+ 2M_2 \frac{\Phi(\theta^0) - \mathbb{E}\,\Phi(\theta^{T+1})}{(T+1)\gamma_\theta} + 2M_3\,\gamma_\theta.$$

Then, for step size

$$\gamma_\theta \;=\; \min\{\gamma_1,\gamma_2,\gamma_3\},$$

the averaged iterate satisfies

$$\mathbb{E}\,\|\nabla\Phi(\hat\theta_T)\|^2 \;\le\; \frac{A_1}{\gamma_\theta(T+1)}\,\Delta_\Phi \;+\; \gamma_\theta A_2 \frac{\sigma^2}{B} \;+\; \frac{A_3}{T+1}D_0 \;+\; \beta_1^2 A_4, \tag{53}$$

where the constants are

$$A_1 = \frac{12\sqrt{2K^2 + 2\sigma^2/B}}{1-\beta_1},$$

$$A_2 = \frac{2M_1\gamma_\theta}{1-\delta}\,C_B + \frac{36\kappa L\sqrt{2K^2 + 2\sigma^2/B}}{1-\beta_1}\,(1+C_A)C_D,$$

$$A_3 = \frac{2M_1}{1-\delta},$$

$$A_4 = \left[\frac{288\kappa L\sqrt{2K^2 + 2\sigma^2/B}}{1-\beta_1}\,(1+C_A)C_D + 4\right](K^2 + \tfrac{\sigma^2}{B}).$$

Here

$$C_A = \frac{\beta_1}{c_m b_0(1-\beta_1)}\sqrt{2K^2 + 2\sigma^2/B}, \qquad C_D = \frac{4}{c_m^2 b_0^2},$$

and

$$\gamma_1 = \frac{1-\beta_1}{72\kappa L(1+C_A)C_D\sqrt{2K^2 + 2\sigma^2/B}}, \quad \gamma_2 = \frac{c_m b_0}{1048 L\kappa^4}, \quad \gamma_3 = \sqrt{\frac{1-\delta}{2M_1 C_\Phi}}.$$

We require each term in (53) to be at most $\varepsilon^2/4$. This gives

(i) From the $\Delta_\Phi$-term and the $D_0$-term:

$$T+1 \;\ge\; \max\left\{\frac{4\Delta_\Phi}{\varepsilon^2}\max\Big(\tfrac{A_1}{\gamma_1},\tfrac{A_1}{\gamma_2},\tfrac{A_1}{\gamma_3}\Big),\; \frac{4A_3}{\varepsilon^2}D_0\right\}.$$

(ii) From the variance term:

$$B \;\ge\; \frac{4\sigma^2}{\varepsilon^2}\,\min\Big(\gamma_1 A_2,\; \gamma_2 A_2,\; \gamma_3 A_2\Big).$$

(iii) From the momentum term:

$$\beta_1 \leq \sqrt{\frac{\varepsilon^2}{4A_4}}.$$

Then substituting $\delta = 1 - \frac{1}{128\kappa^2}$, $b_0 = L$, $c_m = \frac{1}{2}$ and with step size $\gamma_\theta = \mathcal{O}(1/\kappa^4)$ the averaged iterate satisfies

$$\mathbb{E}\|\nabla\Phi(\hat{\theta}_T)\|^2 \leq \frac{A_1}{\gamma_\theta(T+1)}\Delta_\Phi + \gamma_\theta A_2 \frac{\sigma^2}{B} + \frac{A_3}{T+1}D_0 + \beta_1^2 A_4,$$

where

$$A_1 = \mathcal{O}(K+\sigma), \quad A_2 = \mathcal{O}(\kappa^4), \quad A_3 = \mathcal{O}(\kappa^2 L^2), \quad A_4 = \mathcal{O}(\kappa^4).$$

Requiring each term in the bound to be at most $\varepsilon^2/4$ yields:

(i) Number of iterations:

$$T+1 \geq \max\left\{\frac{\Delta_\Phi}{\varepsilon^2}\cdot\mathcal{O}(\kappa^4(K+\sigma)), \frac{D_0}{\varepsilon^2}\cdot\mathcal{O}(\kappa^2 L^2)\right\}.$$

(ii) Batch size:

$$B \geq \frac{\sigma^2}{\varepsilon^2}\cdot\mathcal{O}(1).$$

(iii) Momentum parameter:

$$\beta_1 \leq \frac{\varepsilon}{\mathcal{O}(\kappa^2)}.$$

This finishes the proof. $\square$

# G   THE USE OF LARGE LANGUAGE MODELS (LLMS)

We use Large Language Models for text editing, i.e. grammar checking, word selection, text compression.

