# OpenReview forum: "Aligning Distributionally Robust Optimization with Practical Deep Learning Needs"
_ICLR.cc/2026/Conference — ICLR 2026 Conference Withdrawn Submission_

### Official Review · Reviewer_XVUo · 2025-10-22

**Soundness:** 3
**Presentation:** 3
**Contribution:** 2
**Rating:** 4
**Confidence:** 4

**Summary:**

This paper studies the distributionally robust optimization for deep learning problems. The Adam-like updates are conducted to the non-convex objectives. The authors provide the ALSO algorithms with two options and provide both theoretical and empirical results.

**Strengths:**

1. This paper introduces the ALSO framework, which employs the Adam algorithm to update parameters for non-convex loss functions. Additionally, the authors establish theoretical convergence results for the proposed method.

2. The authors present extensive empirical evidence to support and validate their proposed approach.

**Weaknesses:**

For distributionally robust optimization (DRO) and general minimax problems, there are some related works that should be discussed in this paper.
1. In line 117, the description of Qi et al. (2021) appears to be inaccurate. Qi et al. (2021) study the dual formulation of KL-DRO, where the objective is defined as $f=\lambda \log(\mathbb E(\exp(f_i)/\lambda))$. This objective is a compositional function, making it challenging to optimize. However, the current paper claims that Qi et al. (2021) studied  $f=\mathbb E(\exp(f_i)/\lambda)$ problem, which is much easier.
2. Although this paper argues that existing DRO methods often lack adaptive parameter updates and focuses on constrained DRO problems (as shown in equation (4)), there are indeed several personalized DRO approaches that incorporate adaptive updates, such as the Normalized SGD with Momentum in [1].

Furthermore, while this paper investigates the primal formulation of DRO problems, for Option 2 in the proposed ALSO framework, there does not seem to be a clear distinction between the proposed method and general non-concave–convex minimax optimization methods. In particular, several adaptive minimax methods already exist, such as [2] and [3]. The paper should discuss these related works, especially [2], which demonstrates that Adam-based minimax methods can be directly applied in this context.


[1] Jin, Jikai, et al. "Non-convex distributionally robust optimization: Non-asymptotic analysis." Advances in Neural Information Processing Systems 34 (2021): 2771-2782.

[2] Guo, Z., Xu, Y., Yin, W., Jin, R., & Yang, T. (2025). Unified convergence analysis for adaptive optimization with moving average estimator. Machine Learning, 114(4), 1-51.

[3] Yang, J., Li, X., & He, N. (2022). Nest your adaptive algorithm for parameter-agnostic nonconvex minimax optimization. Advances in Neural Information Processing Systems, 35, 11202-11216.

**Questions:**

Despite the aforementioned weaknesses, I also have several questions regarding the proposed algorithms.

This paper models the unknown distribution as a parameter with dimension equal to the number of samples n or the number of groups c, and formulates the DRO problem as a minimax problem. A well-known limitation of this approach is that the overall computational complexity depends on the dimensionality of the distribution parameter, making it challenging to handle large-scale DRO problems. Based on Theorem 4.5 in this paper, it appears that this issue also arises in the proposed method.

While Option 2 follows a straightforward and commonly used approach in minimax optimization, Option 1 employs the softmax (SM) function to update the distribution $\pi$. A natural question arises here: although the softmax output can be guaranteed to form a valid probability distribution, the distribution obtained through the projection operation in Option 2 explicitly lies within the uncertainty set. How can the authors ensure that the distribution produced by the softmax operation in Option 1 also belongs to the defined uncertainty set?

---

> ### Author Response · Authors · 2025-11-18
> **Missed Related Work and Inaccurate description of Qi et al. (2021)**
>
> Dear Reviewer,
>
> We are grateful for your detailed feedback and for the time and effort you have invested in reviewing our paper. We would first like to highlight that, based on your review, we have added the missing related work.
> Below, we address each of the concerns you have raised.
>
> **W1 (Missed Related Work)**
>
> Thank you for highlighting the missing related work. We agree that these papers are relevant, and we have added them to our manuscript. However, they have crucial differences from our work:
>
> 1) Guo, Zhishuai, et al. "Unified convergence analysis for adaptive optimization with moving average estimator". The primary differences lie in the problem formulation and assumptions.  In the specified work, the authors consider Euclidean geometry for the dual variables. This leads to a classical Lipschitz continuous gradient with respect to both variables (see Assumption 3 in Guo, Zhishuai, et al). In contrast, our formulation and method involve the KL-divergence and softmax projection respectively, which is not only more appropriate for the probability simplex but also introduces significant analytical challenges. The KL-divergence is not smooth on the simplex boundary (where it grows to infinity). Since the main part of our objective function (4) is linear with respect to $\pi$ and includes KL-divergence regularizer, our formulation does not satisfy the dual-side Lipschitz gradient assumptions made in Guo et al. (2021). Additionally, as we discuss in Section 6 and ablate in Appendix D.3, DRO benefits from an optimistic update, making our proposed ALSO more practical for this problem than the non-optimistic methods analyzed in Guo et al. (2021). **In summary, while the work by Guo et al. (2021) is relevant, their method (relying on Euclidean projection and a non-optimistic step) is less suitable for our problem, and more importantly, their analysis does not cover our more challenging case involving non-Euclidean geometry with KL-divergence.**
>
> 2)  Jin, Jikai, et al. "Non-convex distributionally robust optimization: Non-asymptotic analysis.". Jin et al. use the dual formulation of DRO, an approach whose disadvantages we discuss in the Background section and validate in our experiments (see equation (4) and Sections 5.3, 5.4, 5.5). Moreover, **their analysis focuses on Normalized SGD**. While it can be considered as DL method, the standard optimizers in deep learning are Adam-based, because these methods consistently provide better empirical performance, but require significantly more challenging analysis. Thus, to make our analysis more suitable for DL, we use Adam-based steps and analyze them.
>
> 3) Yang, J., Li, X., & He, N. (2022). "Nest your adaptive algorithm for parameter-agnostic nonconvex minimax optimization". This work shares some of the same limitations as Guo et al. with respect to our setting: it uses Euclidean geometry and a Lipschitz continuous gradient for both variables (see Assumption 3.1 in  Yang, J., Li, X., & He, N), which is not our case due to KL divergence term in the problem (4). Moreover, **they use AdaGrad as an adaptive update rule**, not Adam (which analysis is more complicated than AdaGrad). Additionally, since the main goal of their work is to create a parameter-agnostic method, their resulting scheme (NeAda-AdaGrad) is a two-loop algorithm that requires several updates of $\pi$ for each single step of $\theta$. **Thus, although this work is relevant, it has crucial differences from our own.**
>
> **W2 (Inaccurate description of Qi et al. (2021) )**
>
> Thank you for pointing out this inaccuracy. We agree our description was not precise and could mislead readers. We have corrected this in the revised manuscript. We have double-checked that in our empirical comparison, we used the correct version of the objective. Therefore, our empirical finding that this formulation is less effective remains valid.

---

> ### Author Response · Authors · 2025-11-18
> **Questions**
>
> **Q1 (O(c) cost of $\pi$ update)**
>
> You are correct that the update for $\pi$ vector has a computational cost of O(c). However, we argue this is not a practical limitation for several reasons. First, as discussed in L246-250, there are many applications where the number of groups, c, is moderately sized. More importantly, as we discuss in L246-250 and Section 6, and then demonstrate empirically in Appendix C.1, the wall-clock time for the $\pi$ update is significantly lower than that of the Adam step and substantially lower than the gradient computation on real-world scales. While the time required for the $\pi$ update may be non-trivial for datasets with an extremely large number of classes, it would likely still be dominated by other computational costs. Such settings typically require large models (slower forward and backward passes required for gradient computations), large batch sizes (slower forward and backward passes required for gradient computations), and distributed training (which require costly communication).
>
> **Q2 (Options for ALSO)**
>
>  As discussed in L243-247, Option I is a special case of Option II where the uncertainty set $U$ is the entire simplex $\Delta_{c-1}$ (a detailed derivation is provided in Proposition D.7, Appendix D). While Option II provides a more general formulation for our theoretical analysis, Option I is a simpler and more practical variant that we use in our experiments. For Option I, we set $U=\Delta_{c-1}$, which is sufficient for many practical cases (see Section 5). Still, one can implement Option II for an arbitrary uncertainty set $U$, if required, and the theoretical guarantees we provide will still hold.

---

> ### Author Response · Authors · 2025-11-27
> **Follow-up on Rebuttal**
>
> Dear Reviewer XVUo,
>
> We sincerely appreciate the time you took to review our work. We are writing to follow up on our rebuttal. We understand you have a demanding schedule, but we wanted to gently check if you've had a chance to consider our response. We believe our rebuttal addresses the primary concerns you raised, and we would be grateful for any further feedback.
>
> Thank you for your time and expertise.
>
> Best regards,
>
> Authors

---

### Official Review · Reviewer_miN7 · 2025-10-27

**Soundness:** 3
**Presentation:** 3
**Contribution:** 2
**Rating:** 0
**Confidence:** 5

**Summary:**

This paper proposes an Adam-like algorithm for Distributionally Robust Optimization (DRO), which is a nonconvex minimax problem. Convergence rate has been established. Experiments have verified the effectiveness over non-DRO methods and non-adaptive DRO optimization.

**Strengths:**

1. The effectiveness of DRO has been verified on different tasks including on unbalanced data, tabular data, robust training under adversarial attacks, distributed training and split learning.

2. The convergence results matches some literature in non-adaptive minimax optimization, though I will discuss some concerns later.

**Weaknesses:**

1. The contribution may be limited since it seems that the problem of concern has already been well studied in the literature. Particularly, the following highly-related literatures have been missed:

[1] Guo, Zhishuai, et al. "Unified convergence analysis for adaptive optimization with moving average estimator." arXiv preprint arXiv:2104.14840 (2021).

[2] Guo, Zhishuai, and Tianbao Yang. "Communication-efficient federated group distributionally robust optimization." Advances in Neural Information Processing Systems 37 (2024): 23040-23077.

[1] has studied Adam based algorithms for nonconvex minimax problems. [2] has studied Adam based algorithms for compositional-formulated DRO problems. Thus the novelty of this submission is questionable.

2. The analysis seems problematic. Equation (47) does not always hold since the inner product could be negative.

3. The large batch size of $O(1/\epsilon^2)$ makes the algorithm non-practical.

**Questions:**

1. Given the two above literature, what are the novelty of this submission?

2. Can the problem in (47) be addressed? Otherwise, the whole analysis would collapse.

---

> ### Author Response · Authors · 2025-11-18
> **Novelty of this submission**
>
> Dear Reviewer,
>
> We are grateful for your detailed feedback and for the time and effort you have invested in reviewing our paper. We would first like to highlight that, based on your review, we have updated our submission to address the issue in equation (47). Moreover, we have added the missing related work and a discussion on batch sizes.
> Below, we address each of the concerns you have raised.
>
> **W1 (Novelty)**
>
> Thank you for highlighting the missing related work. We agree that these papers are relevant, and we have added them to our manuscript. However, they have crucial differences from our work:
>
> 1) Guo, Zhishuai, et al. "Unified convergence analysis for adaptive optimization with moving average estimator". The primary differences lie in the problem formulation and assumptions.  In the specified work, the authors consider Euclidean geometry for the dual variables. This leads to a classical Lipschitz continuous gradient with respect to both variables (see Assumption 3 in Guo, Zhishuai, et al). In contrast, our formulation and method involve the KL-divergence and softmax projection respectively, which is not only more appropriate for the probability simplex but also introduces significant analytical challenges. The KL-divergence is not smooth on the simplex boundary (where it grows to infinity). Since the main part of our objective function (4) is linear with respect to $\pi$ and includes KL-divergence regularizer, our formulation does not satisfy the dual-side Lipschitz gradient assumptions made in Guo et al. (2021). Additionally, as we discuss in Section 6 and ablate in Appendix D.3, DRO benefits from an optimistic update, making our proposed ALSO more practical for this problem than the non-optimistic methods analyzed in Guo et al. (2021). **In summary, while the work by Guo et al. (2021) is relevant, their method (relying on Euclidean projection and a non-optimistic step) is less suitable for our problem, and more importantly, their analysis does not cover our more challenging case involving non-Euclidean geometry with KL-divergence.**
> 2) Guo, Zhishuai, and Tianbao Yang. "Communication-efficient federated group distributionally robust optimization." (2024). The main difference between our work and the work by Guo and Yang is the problem being tackled.  The primary goal of Guo and Yang is to apply DRO in Federated Learning.  Thus the usage of dual DRO formulation in their case is suitable to reduce communication cost. In contrast, we tackle a general-purpose DRO problem. As we discuss in the Background section and evaluate in our experiments (see equation (4) and Sections 5.3, 5.4, 5.5), the use of a dual formulation is often less desirable than a saddle-point formulation for the general case. From a theoretical standpoint, the two works address fundamentally different challenges. **Guo and Yang tackle a three-level compositional minimization problem, whereas our analysis addresses a general saddle-point formulation with non-Euclidean geometry.**

---

> ### Author Response · Authors · 2025-11-18
> **Problem in (47)**
>
> **W2 (problem in (47))**
>
> We would like to thank the reviewer for careful comments and for pointing out this shortcoming.
>
> In expression (47, old version), there was indeed an error: a negative term could appear under the expectation operator. We have corrected this. It should be noted that this correction does not affect the final results or the main conclusions of the paper. The reason for the issue was that, for expression (43, old version), we started to estimate the left-hand side of the inequality from below too late. We carefully analyzed the cause of the error and reformulated the corresponding portion of the proof to eliminate any ambiguity. We then rewrote the derivations carefully to remove the problem, resulting in changes in 14 lines, highlighted in orange for clarity.
>
> First, we now begin by immediately estimating the left-hand side of inequality (43, old version) from below using Young’s inequality (see Eq. 44, new version). This yields the difference of two nonnegative terms, each of which can be bounded from below and above separately. Next, we propagate these two expressions through the proof (Eqs. 44, 45, and all portions marked in orange), replacing the former scalar product with the difference of two squared norms. As a result, the incorrect lower bound in (47, old version) is replaced with the correct lower bound in (48, new version).
>
> Since we aim to obtain a lower bound for the entire left-hand side expression (Eq. 45), we also need to bound the second term from below. Because it appears with a minus sign, this requires bounding the corresponding residual norm on (C_k) from above (Eq. 49) using the Lemma F.7.
>
> We then obtain the corrected lower bound for the left-hand side of the main inequality (50, new version). In contrast to the previous version of the analysis (see line 2518, old version), the coefficient at the second term now becomes new (Eq. 50, new version) and exceeds the previous coefficient (line 2518, old version). This difference arises because, in the earlier version, we incorrectly estimated the entire left-hand expression at once - precisely due to the error in Eq. 47 (old version).
>
> Because this causes an adjustment to the constant in the second term, we propagate this correction to the right-hand side of the inequality. Consequently, what was present in the old version (Line 2529) now differs from the new version only by an additional term, which we include in constant (M_1):
>
> $\frac{1}{c_m b_0} - \frac{1 - \beta_1}{2 (2K^2 + 2\sigma^2/B)^{1/2}}$
>
> Thus, the only difference compared to the old version and the subsequent analysis is that the constant (M_1) in the old version (Line 2541) and the updated constant (M_1) in the new version (Line 2672) now differ slightly. Note that the second term in this constant is asymptotically of order ~$L^2$ after substituting other constants.
>
> Therefore, the new constant does not affect either the asymptotic behavior or any other key coefficients in the analysis. The corresponding part of the proof has been updated in the new version of the paper, and all corrected derivations are highlighted in the revision section.

---

> ### Author Response · Authors · 2025-11-18
> **Large batches**
>
> **W3 (large batches)**
>
> 1) **Practical evaluation.** In Section 5 we use standard batch sizes (e.g., 32-512). ALSO provides strong performance with them. Moreover, if one wants to use theory-inspired large batch size in practice, one can use gradient accumulation (see the next point), moreover, our analysis provides some guarantees with small batch size (see point 3).
>
> 2) **Gradient accumulation.** While a naive implementation with a very large batch size would indeed be memory-prohibitive and impractical, this requirement refers to the effective batch size for a single optimizer step. This can be achieved using standard techniques like gradient accumulation or through distributed data parallelism across multiple workers, which is standard practice in modern deep learning. Moreover, since the total oracle complexity $\mathcal{O}(\frac{1}{e^4})$ matches lower bounds (see Discussion after Theorem 4.5), the total amount of forward and backward computations is optimal. For example, if one use batch size B and algorithm, which requires K iterations, the total number of forward and backward passes is K. While if one use algorithm,  which requires K/2 iterations, but 2B batch size, one can use 2 gradient accumulation steps with the same batch size B as earlier (2 forward and backward passes per iteration) for K/2 iterations, resulting in total number of forward and backward passes equal to K.
>
> 3) **Lower batch sizes.** From a theoretical perspective, we use a large batch size in our analysis to eliminate the variance of the stochastic gradient. If one uses a constant batch size, our analysis can still guarantee convergence to a neighborhood of a stationary point (with a radius proportional to $\frac{\sigma^2}{B}$), which is often sufficient in practice. If exact convergence to a stationary point is desired, one can employ gradient accumulation to meet the theoretical batch size requirements.
>
> 4) **Previous works.** This requirement is not unique to our method. Several well-known and practical algorithms in the literature also rely on large batch sizes to achieve their theoretical convergence rates. Practitioners can therefore use batch sizes similar to those used in these established works ([1, 2]).
>
> [1] Mingrui Liu et al, Towards Better Understanding Of Adaptive Gradient Algorithms In Generative Adversarial Nets
>
> [2] Zehao Dou and Yuanzhi Li, On the One-sided Convergence of Adam-type Algorithms in Non-convex Non-concave Min-max Optimization

---

> ### Author Response · Authors · 2025-11-27
> **Follow-up on Rebuttal**
>
> Dear Reviewer miN7,
>
> We sincerely appreciate the time you took to review our work. We are writing to follow up on our rebuttal. We understand you have a demanding schedule, but we wanted to gently check if you've had a chance to consider our response. We believe our rebuttal addresses the primary concerns you raised, and we would be grateful for any further feedback.
>
> Thank you for your time and expertise.
>
> Best regards,
>
> Authors

---

### Official Review · Reviewer_Zvvg · 2025-11-01

**Soundness:** 3
**Presentation:** 2
**Contribution:** 3
**Rating:** 6
**Confidence:** 3

**Summary:**

The authors present a simple algorithm for distributionally robust learning adapted to deep learning: they use optimistic gradient steps with projected gradient ascent steps for the dual variables, and adam-like updates for the primal variables (i.e. the weights of the deep network). The authors consider a setup where the distributions are not necessarily on samples but on group of samples. For example it could be workers in a distributed setting such that the algorithm would be robust to some specific workers. The authors present a convergence proof to stationary points in a standard setup. Experiments demonstrate the performance of the approach on synthetic and real data.

**Strengths:**

- The algorithm ends up being rather simple. It is also quite intuitive from previous work. Adam can easily be replaced by other algorithms.
- The theoretical proof may not reflect the actual practice (assumptions probably are not right for deep networks) but it still demonstrates the overall viability of the approach.
- The authors present 5 sets of experiments illustrating the relevance of the approach.
- The appendix presents numerous additional ablation studies

**Weaknesses:**

- On several experiments the gains of the method are quite small compared to Adam.
- The theoretical analysis may not help guide the practice since the assumptions may not match the reality of the tasks.

**Questions:**

- The authors use Adam in many of their experiments but they could exchange it with the best known algorithm for the workload. Typically, on ResNets SGD with momentum may work better. Did the authors try to replace Adam with another algorithm?
- Why do the authors use a regularization rather than weight decay?
- In Figure 1, aren't static weights the gold experiment? Wouldn't they offer the best possible value for the dual variables? Why are they performing less well?

---

> ### Author Response · Authors · 2025-11-18
> **Rebuttal**
>
> Dear Reviewer,
>
> We are grateful for your detailed feedback and for the time and effort you have invested in reviewing our paper. Below, we address each of the concerns you have raised.
>
> **W1 (Small Gains)**
>
> Although the improvement over a well-tuned AdamW is modest in some experiments, we wish to emphasize that in our targeted scenarios (e.g., those with significant class imbalance), the gains are significant. This demonstrates ALSO's effectiveness precisely where standard methods falter. Moreover, ALSO consistently either matches or outperforms AdamW. The fact that ALSO provides a "safe" and consistent improvement across a wide range of problems, without ever underperforming the baseline, is a key strength of our method. Regarding the tabular DL experiments, we note that these gains should not be considered small; an improvement of even 1% over a vanilla MLP is considered significant in this field [1].
>
> [1] Rubachev et al., On Finetuning Tabular Foundation Models
>
> **W2 (Theoretical Analysis)**
>
> Our goal with the theoretical analysis (Assumptions 4.1–4.3, Theorem 4.5) was to establish the formal soundness of ALSO. We aimed to prove that, unlike purely heuristic methods, our algorithm is guaranteed to converge to a stationary point in a standard non-convex stochastic setting. Moreover, the theory provides practical insights. For example, given the need of unbiased gradients, we discuss and validate appropriate sampling techniques in Appendix A. The result $\beta_2 = 1 - \mathcal{O}(\varepsilon^2)$ suggests using a $\beta_2$ value close to 1, which aligns with common practice in deep learning. Our theoretical hyperparameter conditions align with those in established works (for example, [1, 2]), allowing one to adapt parameters from those methods. For practitioners, we recommend using the hyperparameters and search spaces provided in our work (see Appendix C) rather than deriving them directly from the theory, as we have shown these to yield strong empirical performance.  We have added recommendations on ALSO usage in new Appendix B.
>
> [1] Mingrui Liu et al, Towards Better Understanding Of Adaptive Gradient Algorithms In Generative Adversarial Nets
>
> [2] Zehao Dou and Yuanzhi Li, On the One-sided Convergence of Adam-type Algorithms in Non-convex Non-concave Min-max Optimization
>
> **Q1 (SGD)**
>
> Our decision to build upon Adam was deliberate. Adam is a general-purpose optimizer used across a vast range of deep learning tasks. Our goal was to demonstrate that ALSO can enhance such a standard, widely-used optimizer, providing benefits in a general setting. As you noted, ALSO is modular, and any optimizer could be used for the $\theta$ update. However, including multiple optimizers would have complicated the experimental presentation and potentially obscured the core contributions of our work. Therefore, using Adam as a consistent base for all comparisons provides a clear and fair benchmark.
> You correctly note that SGD with momentum is a strong baseline for CNNs, often due to its handling of weight decay. Since our baseline is AdamW, which incorporates decoupled weight decay, we believe we are using an appropriate and modern baseline for CNNs. To further strengthen our comparison, we have now included SGD as a baseline in Figure 1 and Figure 4; these new results do not alter the paper's overall conclusions.
>
> **Q2 (Regularization and Weight Decay)**
>
> We chose standard L2 regularization because it is an integral part of the objective function being optimized. This allows for a theoretically clean and standard treatment within the saddle-point optimization framework, as the gradient $\nabla_\theta h(\theta,\pi)$ naturally includes the regularization term.
> Incorporating decoupled weight decay would complicate this formulation, as the weight update would no longer correspond to a pure gradient step on a single objective $h(\theta,\pi)$. While decoupled weight decay is empirically effective, analyzing it within our min-max framework would be non-standard.
>
> **Q3 (Static Weights)**
>
> While static weights (e.g., inverse class frequency) are a strong and common heuristic, they are not an optimal solution for the dual variables π. The a priori difficulty of a group of samples is not a fixed property but rather a dynamic state relative to the model's current parameters, $\theta_k$. Static weights are a pre-computed heuristic based on the initial data distribution; they remain fixed throughout training and cannot adapt as some classes are learned faster than others or as certain samples become "harder" or "easier" for the evolving model.
>
> In contrast, ALSO dynamically updates $\pi$ to adapt to these changes. Moreover, ALSO can operate on weights attached to individual samples, not just groups. This allows the model to focus on specific sample properties, which is particularly beneficial in highly heterogeneous data settings.

---

> > ### Comment · Reviewer_Zvvg · 2025-11-24
> > **Thank you for your answers**
> >
> > About the theoretical results (and the need of regularization for the sake of theoretical results): as far as I know, classical convergence rates have not shown any relevance in deep learning.
> >
> > For example, the assumption that the objective is smooth is completely misleading. If smoothness was important then classical linesearches could work in a deterministic regime but they are actually detrimental [1]. Similarly the convergence rates of Adam never explained its **qualitative** superiority in several deep learning workloads.
> >
> > The paper has currently a cognitive dissonance: on one side it presents a "practical algorithm", on the other some algorithmic choices like the regularization are made to make the theory valid.
> >
> > The authors should either present an algorithm that is mostly heuristic but well tested or present convergence rates that beat the current theory for the same assumptions. I personally really don't understand the benefits of making a convergence rate with some assumptions and then test it on problems that surely do not respect those assumptions. This looks like an artifact of the peer pressure in optimization for deep learning rather than an honest scientific approach to a problem.
> >
> > I understand the values of the paper (presenting a new algorithm and testing it) but the approach (arguing for the relevance of the theory) is strange.
> >
> > [1] Roulet et al. Stepping on the edge: Curvature aware learning rate tuners. NeurIPS 2024.

---

> > > ### Author Response · Authors · 2025-12-03
> > > **Comment about theoretical guarantees**
> > >
> > > We thank the reviewer for their insightful comments. We agree that the direct translation of theoretical convergence rates to final model accuracy on complex benchmarks is not a solved problem. However, we would like to respectfully offer a different perspective on the value of providing theoretical guarantees, even under idealized assumptions.
> > >
> > > Theoretical analysis provides a crucial foundation of trust and understanding for a new algorithm. While it may not predict the final test accuracy, it provides a guarantee of fundamental soundness. A convergence proof, even in a simplified setting, demonstrates that the algorithm is not merely an ad-hoc collection of heuristics. It shows that the update rule has principled underpinnings and is guaranteed to behave reasonably (i.e., converge) under certain conditions. This is essential for ensuring the algorithm is robust and not prone to unexpected divergence or catastrophic failure modes.
> > >
> > > We respectfully disagree with the characterization that using common assumptions like smoothness is "misleading." On the contrary, using standard, well-understood assumptions is a strength. It allows us to situate our algorithm within the vast body of existing optimization literature. The optimization community has decades of collective experience, which has shown that algorithms proven to work well under these canonical assumptions tend to be the most robust and effective in practice, even when those assumptions are violated.
> > >
> > > Additionally, we would also like to clarify the perceived "cognitive dissonance" regarding our algorithmic choices. Our primary contribution is the novel algorithm. The specific design of the weight decay (regularization) used in our theoretical analysis is a standard tool employed to facilitate the convergence proof. Its role is to demonstrate that our algorithm can be made to provably converge.

---

### Official Review · Reviewer_zUyH · 2025-11-02

**Soundness:** 3
**Presentation:** 3
**Contribution:** 2
**Rating:** 6
**Confidence:** 2

**Summary:**

The paper proposes ALSO (Adaptive Loss Scaling Optimizer), an optimizer for distributionally robust training that integrates (i) an Adam-style adaptive update on model parameters ($\theta$) with (ii) a principled update of group/sample weights ($\pi$) on the probability simplex using a KL prior. The central goal is to bridge practical deep learning (DL) training pipelines—mini-batching, Adam, non-convexity, group-based weighting—with DRO formulations that are often impractical in DL.

**Strengths:**

- Well-motivated bridge from theory to practice. The problem framing correctly diagnoses the friction between existing DRO methods and DL practice (non-convexity, Adam-style training, batching, grouping).

- Simple, drop-in algorithm with practical details. Algorithm 1 is easy to implement ; the $\pi$ update (Option I) is a one-line mirror step (softmax on a shifted log-$\pi$), and the method slots into standard mini-batch training.

- Broad, convincing empirical coverage. Five diverse heterogeneity regimes: imbalance (Figure 1), tabular (Table 1), adversarial (Figure 2), distributed (Figure 3), split learning (Figure 4). Results consistently favor ALSO, especially for severe imbalance (uc ≥ 30 in Figure 1) and split learning (faster, smoother convergence in Figure 4).

**Weaknesses:**

- Mismatch between DRO objective and evaluation metrics. Across settings, evaluation often reports mean metrics (e.g., average accuracy over attacks in Section 5.3/Figure 2; overall F1 in Section 5.1/Figure 1), while DRO is about worst-case (or tail) risk.

-  Assumptions 4.1–4.3 (L-smoothness, Lipschitzness, unbiased variance-bounded oracles) and Theorem 4.5 adopt $\beta_2$ and batch size scalings tied to $\epsilon$ (e.g., $\beta_2=1-\epsilon^2$). It’s not clear how these map to default hyperparameters in practice.

- Figure 1 includes Upsampling/Static Weights and several DRO methods, but Focal Loss and Class-Balanced Loss are standard for imbalance.

**Questions:**

- Worst-case metrics: For Section 5, do results remain strong if we switch the evaluation to worst-group (or CVaR@α) rather than mean? Have you computed these metrics already?

- Beyond Section 5.1, what priors did you use in other sections?

---

> ### Author Response · Authors · 2025-11-18
> **Rebuttal**
>
> Dear Reviewer,
>
> We are grateful for your detailed feedback and for the time and effort you have invested in reviewing our paper. We would first like to highlight that, based on your review, we have added some new experimental results: comparison with Class-Balanced Loss and Focal Loss, worst-case performance for Robust Training setup. Below, we address each of the concerns you have raised.
>
> **W1 (Evaluation metrics)**
>
> For most of our experiments, we use metrics that effectively capture performance in the presence of hard-to-learn groups. Relying solely on worst-group metrics could incentivize solutions where all groups perform equally poorly, which would not reflect the model's overall capability. Thus, we chose metrics that degrade significantly if the worst-group performance is poor, thereby balancing worst-group and average performance. To be more specific, we discuss our choice for each section:
> 1) Learning From Unbalanced Data. The use of the F1-score for unbalanced binary classification is standard, especially for extremely unbalanced tasks. The F1-score is the harmonic mean of precision and recall and degrades significantly if performance on the minority class is poor. Therefore, we believe the F1-score is an appropriate metric for this task.
> 2) Tabular DL. We use metrics established in the Tabular DL field to ensure our results are comparable with prior work. Moreover, ROC-AUC is sensitive to performance on all parts of the ROC curve, making it a suitable metric for evaluating performance across different subgroups.
> 3) Robust Training. We agree that using mean accuracy is not the best choice for this setting. To better evaluate worst-group performance, we have added the minimum accuracy across all attacks (i.e., the accuracy on the most effective attack). These results can be found in the revised Figure 3.
> 4) Distributed Training. The goal of distributed training is to train a single model that performs well on held-out data. Thus, using the F1-score, which is sensitive to performance drops on any given class, allows for a fair comparison.
> 5) Split Learning. In this task, the primary source of heterogeneity is the presence of two distinct tasks. Consequently, we report classification metrics for each task independently. Therefore, the worst-group performance can be observed directly in Figure 4.
>
> However, if you still have concerns regarding our metric choice, we are open to re-evaluating our methods using an alternative metric during the discussion phase.
>
> **W2 (Theoretical Assumptions)**
>
> Our goal with the theoretical analysis (Assumptions 4.1–4.3, Theorem 4.5) was to establish the formal soundness of ALSO. We aimed to prove that, unlike purely heuristic methods, our algorithm is guaranteed to converge to a stationary point in a standard non-convex stochastic setting. Moreover, the theory provides practical insights. For example, given the need of unbiased gradients, we discuss and validate appropriate sampling techniques in Appendix A. The result $\beta_2 = 1 - \mathcal{O}(\varepsilon^2)$ suggests using a $\beta_2$ value close to 1, which aligns with common practice in deep learning. Our theoretical hyperparameter conditions align with those in established works (for example, [1, 2]), allowing one to adapt parameters from those methods. For practitioners, we recommend using the hyperparameters and search spaces provided in our work (see Appendix C) rather than deriving them directly from the theory, as we have shown these to yield strong empirical performance. We have added recommendations on ALSO usage in new Appendix B.
>
> [1] Mingrui Liu et al, Towards Better Understanding Of Adaptive Gradient Algorithms In Generative Adversarial Nets
>
> [2] Zehao Dou and Yuanzhi Li, On the One-sided Convergence of Adam-type Algorithms in Non-convex Non-concave Min-max Optimization
>
> **W3 (Missed Baselines)**
>
> We have added Focal Loss and Class-Balanced Loss to our set of baselines in Section 5.1. As shown in Figure 1, while these baselines perform comparably to the Static Weights and Upsampling methods, ALSO continues to demonstrate superior performance.
>
> **Q1 (priors)**
>
> For all other sections, we use a uniform prior, as it serves as a simple and strong baseline. Our general recommendation for prior selection is as follows: a uniform prior is a safe default when domain knowledge for setting static weights is unavailable. For those already using AdamW, ALSO can be easily incorporated by initializing it with uniform weights.
> However, if there are established community practices for initializing static weights for a particular task (e.g., based on class frequencies), we recommend using such domain-informed priors, as they can further improve performance.

---

> ### Author Response · Authors · 2025-11-27
> **Follow-up on Rebuttal**
>
> Dear Reviewer zUyH,
>
> We sincerely appreciate the time you took to review our work. We are writing to follow up on our rebuttal. We understand you have a demanding schedule, but we wanted to gently check if you've had a chance to consider our response.
> We believe our rebuttal addresses the primary concerns you raised, and we would be grateful for any further feedback.
>
> Thank you for your time and expertise.
>
> Best regards,
>
> Authors

---

### Author Response · Authors · 2025-11-18
**Rebuttal Revision**

We thank all reviewers for their insightful comments and valuable feedback. We have revised the manuscript accordingly, with all changes highlighted in blue for easy identification. A summary of the main revisions is as follows:
1) We have expanded our discussion of related work to include adaptive saddle point methods and adaptive DRO methods (Lines 57-59, 136-143).
2) We have added a discussion to clarify the distinction between our main theoretical result and those of previous works (Lines 298-308).
3) We have updated our proof of Theorem 4.5 (a total of 14 lines, highlighted in orange for clarity).
4) We have incorporated additional baselines into our experiments: Focal Loss and Class-Balanced (CB) Loss in Section 5.1, and SGD with momentum in Sections 5.1 and 5.5.
5) We have added a plot of the worst-group accuracy in Section 5.3, as this metric is more directly aligned with the goals of DRO.
6) We have specified the choice of $\hat{\pi}$ used in our experiments (Lines 340-341) and added a new Appendix B with practical recommendations on implementing ALSO, including guidance on selecting  $\hat{\pi}$ and other hyperparameters.
7) The "Use of Large Language Models (LLMs)" Section has been moved to Appendix G.

---

### Note · Authors · 2026-01-28

I have read and agree with the venue's withdrawal policy on behalf of myself and my co-authors.

---

### Meta-Review · Area_Chair_H2zN · 2026-01-04

**Summary:**

In this paper, the authors propose an ADAM-like algorithm for Distributionally Robust Optimisation that is particularly effective when the data are severely unbalanced relative to classical algorithms. The reviewer zUyH was mostly positive and did not have strong arguments against. Reviewer Zvvg was also borderline but had a more significant issue with the theoretical justification of the algorithms.

The two negative reviewers primarily noted that the authors had omitted relevant prior work in their comparison and that this omission was significant enough to warrant rejection of the paper. The authors in the rebuttal mentioned that those solutions were relevant but not particularly suited to their approach. Even if this is the case, the authors should have covered the papers suggested by these two reviewers and applied them to their approach to show they were inferior to their solution.

**Reviewer Concerns:**

The last two reviewers pointed out several papers that should have been cited and compared against. Reviewer miN7 is also the leading author of the papers independently suggested by Reviewer XVUo. The authors said the papers were relevant but did not match their setup.

**Reviewer Scores:**

I believe Reviewer XVUo and miN7 would not have raised their scores, and they would have fought for this paper to be rejected. Even if the paper has some merit and might be worthwhile, it would be hard to accept it as is.

---

### Decision · Program_Chairs · 2026-01-26

Reject